# KorB switching from DNA-sliding clamp to repressor mediates long-range gene silencing in a multi-drug resistance plasmid

Thomas C. McLean [1,8] ✉, Francisco Balaguer-Pérez[2,8], Joshua Chandanani [3,8], Christopher M. Thomas [4], Clara Aicart-Ramos[2], Sophia Burick [3], Paul Dominic B. Olinares[5], Giulia Gobbato[1], Julia E. A. Mundy[6], Brian T. Chait[5], David M. Lawson [6], Seth A. Darst [3] ✉, Elizabeth A. Campbell [7] ✉, Fernando Moreno-Herrero [2] ✉ & Tung B. K. Le [1] ✉

Examples of long-range gene regulation in bacteria are rare and generally thought to involve DNA looping. Here, using a combination of biophysical approaches including X-ray crystallography and single-molecule analysis for the KorB–KorA system in *Escherichia coli*, we show that long-range gene silencing on the plasmid RK2, a source of multi-drug resistance across diverse Gram-negative bacteria, is achieved cooperatively by a DNA-sliding clamp, KorB, and a clamp-locking protein, KorA. We show that KorB is a CTPase clamp that can entrap and slide along DNA to reach distal target promoters up to 1.5 kb away. We resolved the tripartite crystal structure of a KorB–KorA–DNA co-complex, revealing that KorA latches KorB into a closed clamp state. DNA-bound KorA thus stimulates repression by stalling KorB sliding at target promoters to occlude RNA polymerase holoenzymes. Together, our findings explain the mechanistic basis for KorB role switching from a DNA-sliding clamp to a co-repressor and provide an alternative mechanism for long-range regulation of gene expression in bacteria.

Spatiotemporal gene expression in eukaryotes is commonly coordinated by transcriptional enhancers/silencers that loop over long genomic distances to regulate target-gene promoters[1]. By contrast, gene regulation over kilobase distances is rare in bacteria, and it remains to be determined whether other mechanisms, beyond DNA looping and DNA supercoiling[2–4], are involved.

Long-range gene regulation by the DNA-binding protein KorB is important for stable vertical inheritance and horizontal transmission of the broad-host-range multidrug resistance plasmid RK2, as well as for the fitness of its host bacterium[5–10]. KorB is a member of the same family of proteins as ParB. ParB is a self-loading cytidine triphosphate (CTP)-dependent DNA clamp that loads and accumulates in the vicinity of a bacterial centromere-like *parS* sequence on bacterial chromosomes or plasmids[11–15]. ParB is crucial for plasmid and bacterial chromosome segregation[15–18]. However, in contrast to ParB, KorB also functions as a long-range silencer, repressing the expression of plasmid genes that promote replication and conjugative transfer[19]. The best understood KorB-mediated transcriptional repression requires a 16 bp silencer sequence called *OB* (operator of KorB) and the small DNA-binding protein KorA that binds site specifically to a

[1]Department of Molecular Microbiology, John Innes Centre, Norwich, UK. [2]Department of Macromolecular Structures, Centro Nacional de Biotecnología, Consejo Superior de Investigaciones Científicas, Madrid, Spain. [3]Laboratory of Molecular Biophysics, The Rockefeller University, New York, NY, USA. [4]School of Biosciences, University of Birmingham, Birmingham, UK. [5]Laboratory of Mass Spectrometry and Gaseous Ion Chemistry, The Rockefeller University, New York, NY, USA. [6]Department of Biochemistry and Metabolism, John Innes Centre, Norwich, UK. [7]Laboratory of Molecular Pathogenesis, The Rockefeller University, New York, NY, USA. [8]These authors contributed equally: Thomas C. McLean, Francisco Balaguer-Pérez, Joshua Chandanani. ✉e-mail: thomas.mclean@jic.ac.uk; darst@rockefeller.edu; campbee@rockefeller.edu; fernando.moreno@cnb.csic.es; tung.le@jic.ac.uk

12 bp *OA* sequence (operator of KorA). *OB* is positioned either 4–10 bp (*OB* proximal) or 45 bp to >1,000 bp (*OB* distal) upstream of target promoters[19–23]. KorB-mediated long-range gene silencing has been investigated for three decades[19,20,24–26] yet is not fully understood. Conflicting models propose that KorB either polymerizes on DNA to reach the target promoter from a distal *OB* site or promotes DNA looping over long distances to connect *OB* and the target promoter or a combination thereof[19]. In this Article, we investigate KorAB–CTP interaction and provide a unifying model for KorAB-mediated long-range gene silencing. We find that KorB binds CTP to form a protein clamp that can entrap and slide along DNA to mediate long-range transcriptional repression, likely by allowing KorB to reach target promoters from distal *OB* sites. We resolved the tripartite crystal structure of a KorB–KorA–DNA co-complex, finding that the DNA-binding KorA is also a clamp-locking protein that docks underneath the DNA-binding domain (DBD) of KorB to latch the closed-clamp state. Our data suggest that the KorA–KorB interaction stalls KorB sliding at target core promoter elements and exploits an inherently unstable open RNA polymerase (RNAP)–promoter complex to exclude RNAP holoenzymes from the promoters. KorA–KorB interaction also increases the residence time of KorAB on DNA, enhancing repression. Overall, we demonstrate how a DNA-binding and clamp-locking protein KorA allows KorB to switch functions between sliding and stalling on DNA to act as an effective repressor. Our findings, therefore, provide unanticipated insights into long-range transcriptional repression mechanisms in bacteria.

## Results

### KorB is a DNA-stimulated CTPase

As KorB harbours a widely distributed ParB N-terminal domain (NTD) (Fig. 1a and Extended Data Fig. 1a), we sought to determine whether it binds and hydrolyses CTP. Isothermal titration calorimetry (ITC) showed that KorB binds CTP with moderate affinity ($K_D = 4.5 ± 0.5 \mu M$) (Fig. 1b) and approximately tenfold more tightly to a non-hydrolysable analogue cytidine-5′-(3-thiotriphosphate) (CTPγS) ($K_D = 0.4 ± 0.1 \mu M$) (Extended Data Fig. 1b) but qualitatively much weaker to cytidine diphosphate (CDP) (Extended Data Fig. 1b). We then solved a 2.3 Å resolution co-crystal structure of CTPγS with a KorBΔN30ΔCTD variant that lacks the N-terminal peptide (N30) and the C-terminal domain (CTD) (Fig. 1c and Extended Data Fig. 1c) to locate CTP-contacting residues (Fig. 1d). Structure-guided mutagenesis and ITC showed that arginine 117 to alanine (R117A) substitution abolished CTP binding while asparagine 146 to alanine (N146A) did not (Fig. 1b).

The phosphate-binding motif (GERRxR) (Extended Data Fig. 1a) is highly conserved in ParB, with the glutamate residue being crucial for CTPase activity but not for CTP binding[13,14,27]. KorB, despite having an alanine at the equivalent position on its phosphate-binding motif (GARRYR), hydrolysed ~60–90 CTP molecules per KorB dimer per hour in the presence of CTP and a cognate DNA-binding site *OB* (Fig. 1e and Extended Data Fig. 1d), a rate that is comparable to that of the *Bacillus subtilis* ParB-like protein Noc and approximately sixfold faster than *Caulobacter crescentus* ParB (Extended Data Fig. 1d)[14,28]. KorB did not noticeably hydrolyse other nucleoside triphosphates (NTPs) nor CTP if *OB* DNA was removed or a scrambled *OB* DNA was included (Fig. 1e). Furthermore, CTP hydrolysis was abolished in the CTP-binding mutant R117A and also in the CTP-binding proficient N146A (Fig. 1e). Altogether, our data show that KorB is an *OB* DNA-stimulated CTPase.

### CTP and *OB* DNA promote KorB NTD engagement

In the presence of CTP, canonical ParB self-engages at the NTD to create a clamp-like molecule[11–14]. We similarly observed dimerization of NTDs from opposing KorB subunits (with a substantial interface area of ~2,097 Å²) (Fig. 1c), supporting a potential clamping mechanism. To detect KorB NTD dimerization in solution, we used cysteine-specific crosslinking of a purified KorB variant with bismaleimidoethane (BMOE) (Fig. 2a). Based on the KorBΔN30ΔCTD–CTPγS co-crystal

structure and sequence alignment between KorB and *C. crescentus* ParB (Extended Data Fig. 1a), residue S47 at the NTD was substituted by cysteine on an otherwise cysteine-less KorB background (Extended Data Fig. 2a). KorB (S47C) crosslinked at ~20% efficiency in the absence of CTP (lane 2) or in the presence of 24 bp *OB* DNA or scrambled *OB* DNA alone (lanes 3 and 4) (Fig. 2a). Crosslinking efficiency reduced to ~5–10% when CTP alone was included (lane 6) or together with scrambled DNA (lane 5) (Fig. 2a), suggesting that in the absence of the *OB* DNA, CTP inhibits KorB NTD engagement. However, the crosslinking efficiency increased to ~80% when both CTP and cognate *OB* DNA were added together (lane 7) (Fig. 2a), indicating that both CTP and *OB* DNA are required to promote KorB NTD engagement. Consistent with this, a CTP-binding mutant KorB (S47C R117A) did not crosslink beyond ~20% (Fig. 2a). Furthermore, KorB (S47C N146A), which can bind but not hydrolyse CTP, did not crosslink beyond ~40% (Fig. 2a), suggesting that the N146A substitution also impairs NTD engagement. However, this is unlikely due to the lack of CTP hydrolysis as a non-hydrolysable analogue CTPγS could readily promote crosslinking of KorB (S47C), even without DNA or with a scrambled DNA (lanes 8 and 9) (Fig. 2a).

### CTP enables KorB diffusion on an *OB* DNA substrate

To investigate the impact of CTP on KorB–DNA interaction, we used dual optical tweezers combined with confocal fluorescence microscopy[29]. Here, individual DNA molecules were immobilized between two polystyrene beads and extended almost to their contour length under force (Fig. 2b). We did not observe fluorescence signal from Alexa Fluor 488 (AF488)–KorB incubated with DNA devoid of *OB* sites (Extended Data Fig. 3a), consistent with the requirement of *OB* for KorB binding. Next, we used DNA with eight *OB* sites that were present in either one or two clusters on the DNA, to increase the probability of observing KorB–DNA binding (Fig. 2b; see figure legend and Methods for more details on the construction of these DNA molecules). Kymographs showed two stable regions of high fluorescence, corresponding to the position of the two *OB* clusters (Fig. 2b) when AF488–KorB alone was used, consistent with the initial binding of KorB at *OB* being CTP independent. In the presence of CTP or CTPγS, fluorescence signals were identified outside of the *OB* clusters (Fig. 2b), suggesting that KorB–CTP was now distributed non-specifically along the DNA. We determined that AF488–KorB diffuses on DNA with a diffusion constant ($D = ~1.6 ± 0.1 \mu m^2 s^{-1}$) (Extended Data Fig. 3b), which is approximately fourfold higher than that of *B. subtilis* ParB–CTP diffusion on DNA[29]. We also occasionally observed KorB–*OB* stable binding events under both CTP and CTPγS conditions (Fig. 2b). We interpret these events as either new KorB proteins being loaded at 8×*OB* or existing KorB proteins diffusing and occasionally re-binding to *OB* when they re-encounter this cluster.

### KorB–CTP does not condense *OB* DNA in vitro

Canonical ParB–CTP was previously reported to bridge and condense DNA in vitro[29–32]. To test whether KorB induces DNA condensation, we used magnetic tweezers and DNA molecules containing a cluster of 16 *OB* sites (16×*OB*) and an *OA* site (1×*OA*) (Extended Data Fig. 4). We observed no difference in the force–extension curves in the presence or absence of KorB + 2 mM CTP (Extended Data Fig. 4), indicating that KorB–CTP does not condense DNA under the tested conditions. In the case of *B. subtilis Bs*ParB control, we observed unspecific condensation at high concentrations (1–2 μM). As expected, *Bs*ParB was not able to condense at a lower concentration (500 nM) because *Bs*ParB does not recognize *OB* sites on our DNA substrate.

### CTP-dependent KorB NTD engagement is essential for long-range transcriptional repression

KorB was previously shown to mediate long-range transcription repression[19]. To investigate whether CTP binding and DNA sliding influence this activity, we constructed two promoter–*xylE* transcriptional fusion reporters and assayed for catechol dioxygenase activity in the presence

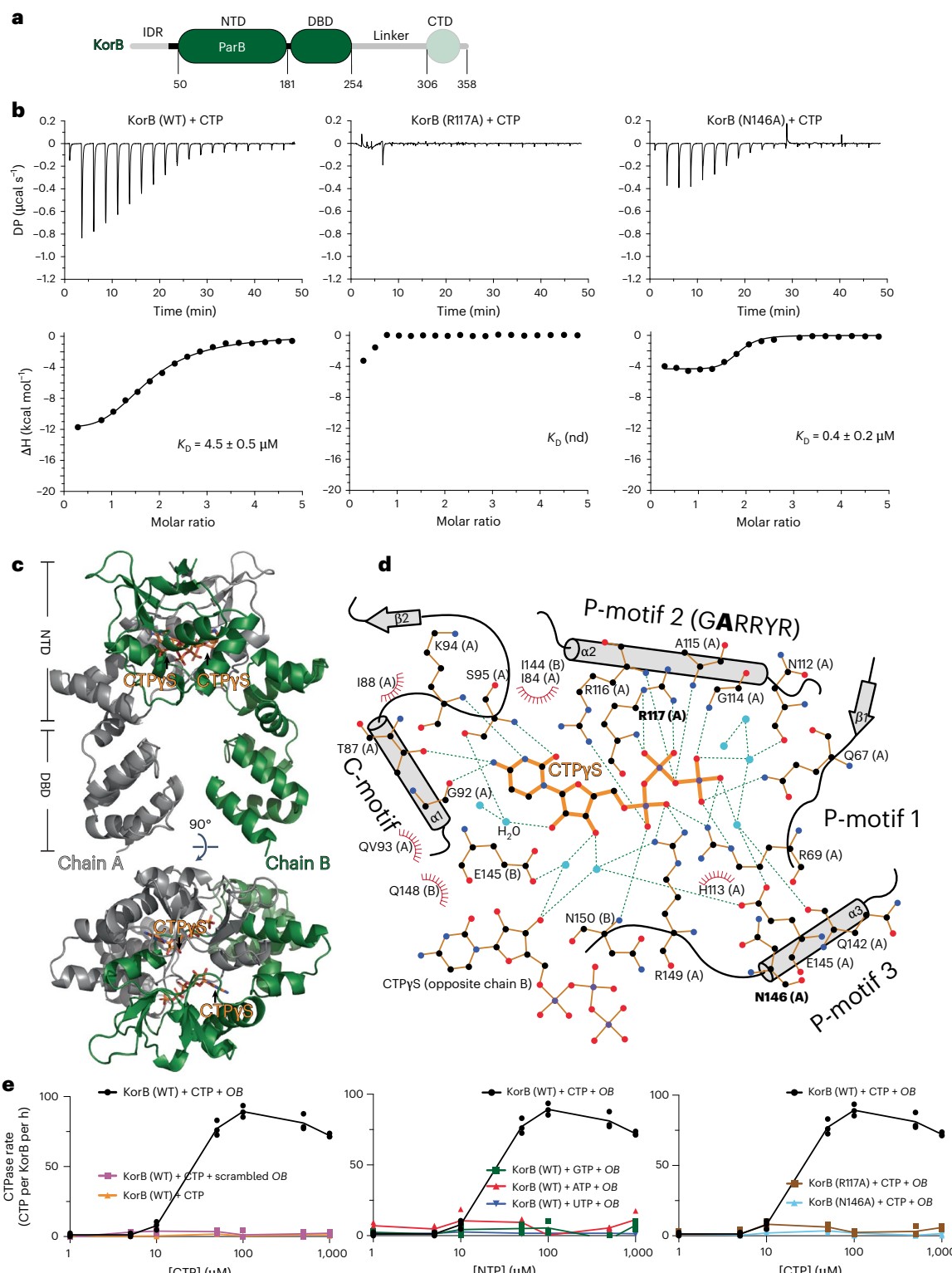

**Fig. 1 | KorB binds and hydrolyses CTP in the presence of *OB* DNA. a**, The domain architecture of KorB: an intrinsically disordered region (IDR) IncC (ParA)-interacting peptide, the NTD, a central *OB* DBD, a predicted flexible 53-amino acid linker and a CTD. The KorBΔN30ΔCTD variant that was used for crystallization lacks the 30 N-terminal amino acids, the linker and the CTD (faded green). **b**, Analysis of the interaction of KorB (WT and mutants) with CTP by ITC. Each experiment was duplicated. The y-axes show a measured power differential (DP) between the reference and sample cells to maintain a zero temperature between the cells, and enthalpy (ΔH) of binding. **c**, Co-crystal structure of a KorBΔN30ΔCTD–CTPγS complex reveals a CTP-binding pocket and a closed conformation at the NTD of KorB. Top: the front view of the co-crystal structure

of KorBΔN30ΔCTD (dark green and grey) bound to a non-hydrolysable analogue CTPγS (orange). Bottom: the top view of the KorBΔN30ΔCTD–CTPγS co-crystal structure. **d**, The protein–ligand interaction map of CTPγS bound to KorBΔN30ΔCTD. Hydrogen bonds are shown as dashed green lines and hydrophobic interactions as red semi-circles. We did not observe electron density for $Mg^{2+}$ in the CTP-binding pocket. N146 does not make contact with CTPγS; however, mutations at the equivalent residue in ParB were previously reported to disrupt CTP hydrolysis[11,28], and thus N146 was also selected for mutagenesis in this study. **e**, NTP hydrolysis rates of KorB (WT and variants) were measured at increasing concentrations of NTP. Experiments were triplicated.

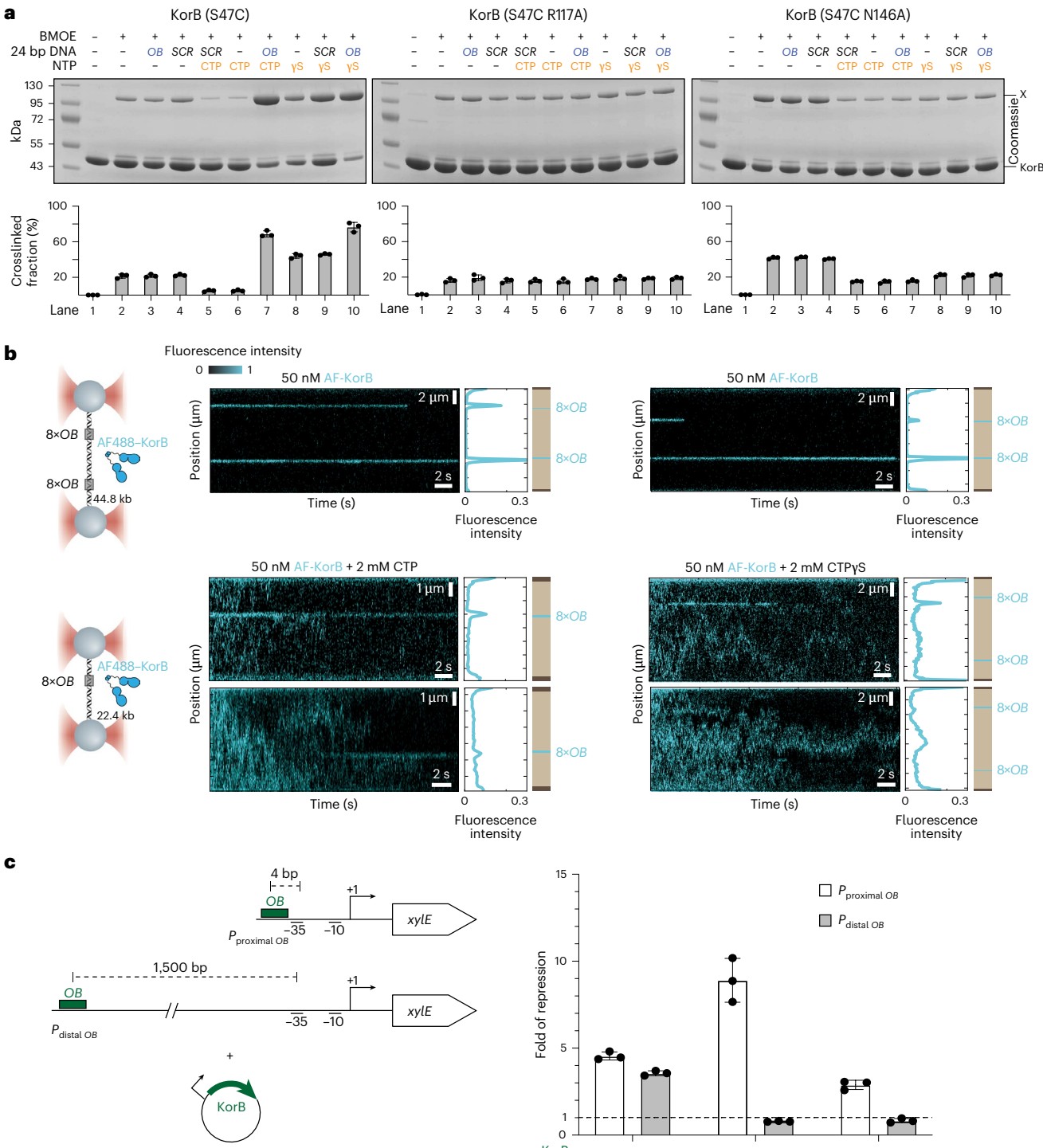

**Fig. 2 | CTP and *OB* DNA promote the engagement of the NTD of KorB in vitro and are essential for KorB to repress transcription from a distance.**
**a**, SDS–PAGE analysis of BMOE crosslinking products of 8 µM of KorB (S47C) dimer (and variants) ± 1 µM 24 bp *OB*/scrambled (SCR) DNA ± 1 mM CTP. X indicates a crosslinked form of KorB. Sub-stoichiometric concentration of *OB* DNA was sufficient to promote efficient crosslinking of KorB (S47C) (Extended Data Fig. 2b). The S47C substitution did not impact the *OB*-mediated repression function of KorB (Extended Data Fig. 2c). Quantification of the crosslinked fraction is shown below each representative image. Data are represented as mean values ± s.d. from three replicates. **b**, CTP binding promotes the diffusion of KorB on DNA containing *OB* sites. Left: schematic of the C-trap optical tweezers experiment where DNA containing one or two clusters of 8×*OB* sites were tethered between two beads and scanned with a confocal microscope using 488 nm illumination. A 44.8 kb DNA was constructed from ligating together two

identical tandem 22.4 kb DNA, each containing 8×*OB* and 1×*OA* site (Methods). The *OA* site is omitted from the diagram for simplicity. Right: representative kymographs showing the binding of KorB at the *OB* cluster in the presence or absence of CTPγS or CTP. Scale bars represent fluorescence intensity on the kymographs. Kymographs were taken in a buffer-only channel, after 60 s incubation in the protein channel, to reduce the fluorescence background. **c**, CTP-dependent N engagement is essential for KorB to repress transcription from a distance. Left: schematic diagrams of promoter–*xylE* reporter constructs. Right: values shown are fold of repression, which is a ratio of XylE activities from cells co-harbouring a reporter plasmid and a KorB-expressing plasmid to that of cells co-harbouring a reporter plasmid and an empty plasmid (KorB-minus control). Data are represented as mean values ± s.d. from three replicates. See Extended Data Fig. 2c for the absolute values of XylE activities and an α-KorB immunoblot from lysates of cells used in the same experiments.

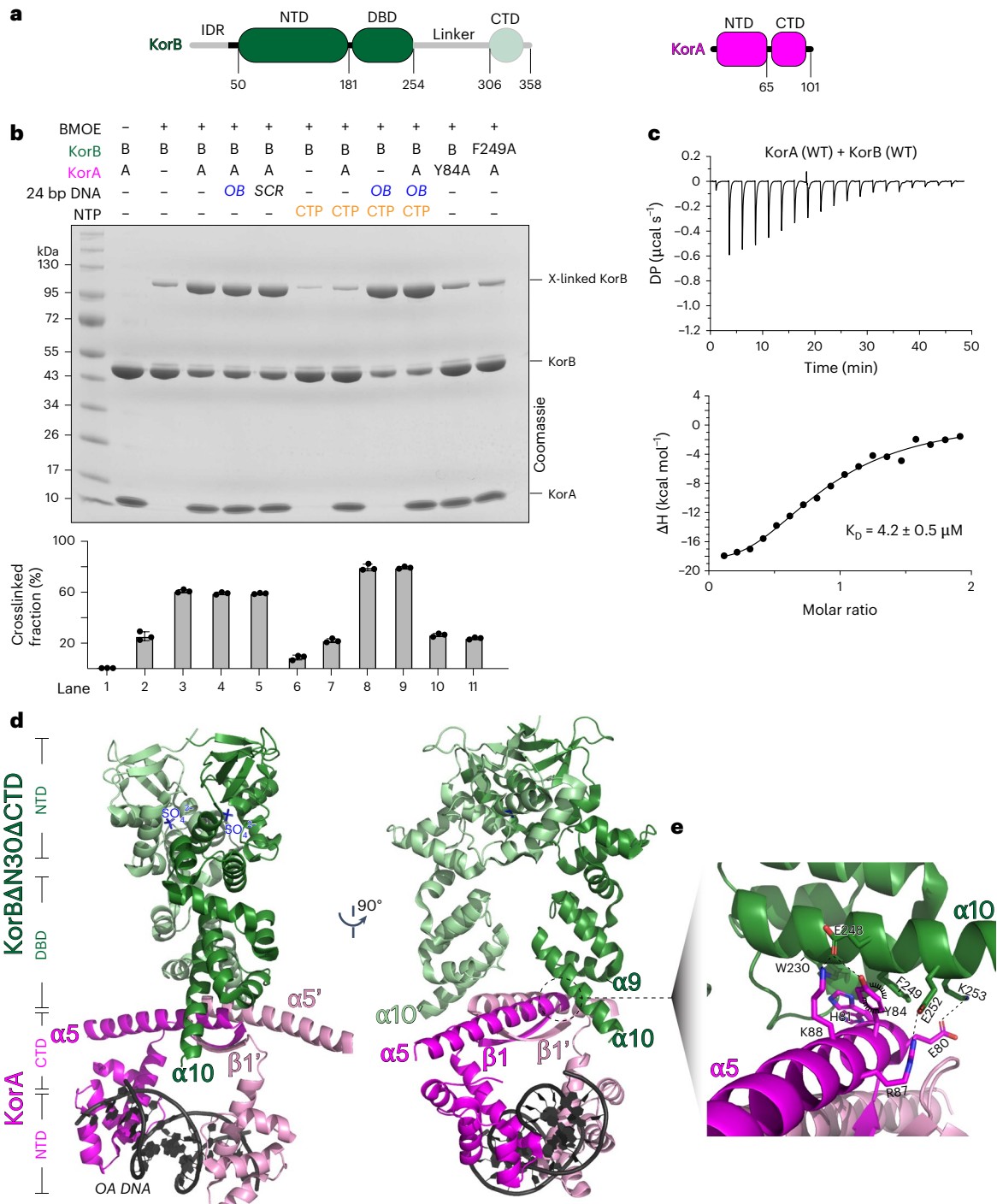

**Fig. 3 | KorA promotes the N engagement of KorB independent of CTP and *OB* DNA in vitro. a**, The domain architecture of KorB (same as Fig. 1a) and KorA. The KorBΔN30ΔCTD variant that was used for crystallization lacks the 30 N-terminal amino acids, the linker and the CTD (faded green). KorA has an N-terminal *OA* DNA-binding domain (NTD) and a C-terminal dimerization domain (CTD). **b**, SDS–PAGE analysis of BMOE crosslinking products of 8 µM of KorB (S47C) dimer (or the variant S47C F249A) ± 8 µM of KorA dimer (WT or the variant Y84A) ± 1 µM 24 bp *OB*/scrambled *OB* DNA ± 1 mM CTP. X indicates a crosslinked form of KorB. Quantification of the crosslinked fraction is shown below each representative image. Data are represented as mean values ± s.d. from three replicates. **c**, Analysis of the interaction of KorB with KorA by ITC. The experiment

was duplicated. **d**, KorA–DNA docks onto the DBD of a clamp-closed KorB in the co-crystal structure of a KorBΔN30ΔCTD–KorA–DNA complex. Left: the side view of the co-crystal structure of KorBΔN30ΔCTD (dark green and light green) bound to a KorA (magenta and pink) on a 14 bp *OA* DNA duplex (black). Right: The front view of the KorBΔN30ΔCTD–KorA–DNA co-crystal structure. **e**, Helix α10 (dark green) of KorB interacts with helix α5 (magenta) of KorA. Hydrogen bonds are shown as dashed lines, pi-stacking interaction between the aromatic ring of Y84 in KorA with the E248–F249 peptide bond of KorB and between the aromatic ring of F249 in KorB with the E80–H81 peptide bond of KorB are shown as black semi-circles (Extended Data Fig. 6b,c).

or absence of KorB in vivo (Fig. 2c). *OB* DNA was either engineered 4 bp (*OB* proximal) or 1.5 kb (*OB* distal) upstream of the −35 element of the core promoter of the RK2 *trbB* gene (Fig. 2c). The presence of KorB led to ~4- to 5-fold transcriptional repression compared with a KorB-minus control for both promoter–reporter constructs, consistent with previous findings[19] (Fig. 2c and Extended Data Fig. 2c). The NTD-engagement-defective KorB (R117A) and KorB (N146A) variants were capable of repressing transcription from an *OB*-proximal promoter but not from an *OB*-distal promoter (Fig. 2c). Overall, our data suggest that the ability of KorB to bind CTP and close the clamp to slide on DNA is crucial for long-range gene silencing.

## KorA promotes the NTD engagement of KorB independent of CTP and *OB* DNA

KorB has previously been reported to require KorA (Fig. 3a) to strengthen transcriptional repression, especially at *OB*-distal promoters[19,25,26], yet it is unclear how KorA and KorB cooperate to do so. We hypothesized that KorA might modulate the CTP-dependent activities of KorB. To investigate this possibility, we pre-incubated purified cysteine-less KorA with KorB (S47C) in the presence or absence of *OB* DNA and/or CTP before crosslinking by BMOE. KorB (S47C) crosslinked at ~20% in the absence of KorA (lane 2) (Fig. 3b) but increased to ~60% in the presence of KorA alone (lane 3) or additionally with *OB* or scrambled *OB* DNA (lanes 4 and 5) (Fig. 3b). In the presence of CTP alone, KorA increased the crosslinking efficiency of KorB (S47C) to ~20% (lane 7) (Fig. 3b), again higher than the ~10% crosslinked fraction observed in the absence of KorA (lane 6) (Fig. 3b). KorB (S47C) crosslinked maximally at ~80% efficiency when KorA, CTP and *OB* DNA were all included (lane 9) (Fig. 3b). KorA also improved the crosslinking efficiency of KorB (S47C) when *OA* DNA was present (lanes 3 and 4) (Extended Data Fig. 5a). It is worth noting that for both KorB (R117A) and KorB (N146A) variants, which are defective in NTD engagement, KorA further increases the crosslinking efficiency in all conditions (Extended Data Fig. 5b). Overall, our results suggest that KorA promotes NTD engagement of KorB. KorA likely does so via direct interaction with KorB[24,25], which we confirmed by ITC (Fig. 3c).

## KorA–DNA docks onto the DBD locking the KorB clamp

To determine the mechanism by which KorA induces the KorB closed-clamp conformation, we solved a 2.7 Å resolution structure of a tripartite complex between KorBΔN30ΔCTD and KorA bound to its 14 bp *OA* DNA duplex (Fig. 3d and Extended Data Fig. 6a). The asymmetric unit contains two subunits of KorA that dimerize via the C-terminal β1 strands and α5 helices (Fig. 3d). The DBDs (helices α1–4) from opposing KorA subunits dock into adjacent major grooves on the *OA* DNA (Fig. 3d). The asymmetric unit also contains two subunits of KorBΔN30ΔCTD whose opposing NTDs (strands β1–3 and helices α1–4) are self-dimerizing (interface area = ~1,250 Å$^2$) (Fig. 3d), showing that KorBΔN30ΔCTD in the tripartite complex has already adopted an NTD-engaged conformation, even though CTPγS was not added to the crystallization set-up (see Supplementary Information for further structural analysis and

discussion). Therefore, we reasoned that KorA–DNA may facilitate or capture KorBΔN30ΔCTD in the NTD-engaged state.

The DNA-bound KorA dimer docks underneath the *OB*-binding domain of KorBΔN30ΔCTD, specifically via α5 of KorA interacting directly with α10 of KorBΔN30ΔCTD (Fig. 3d). A network of specific interactions is established at the KorA–KorB interface, including KorA Y84 and KorB F249 (Fig. 3e and Extended Data Fig. 6b,c). ITC data verified observed interactions, showing that KorA (Y84A)[24] and KorB (F249A) completely abolished the interaction with KorB (wild type (WT)) and KorA (WT), respectively (Extended Data Fig. 5c). Furthermore, neither KorA (Y84A) nor KorB (S47C F249A) increased KorB crosslinking beyond ~20% (lanes 10 and 11 versus lane 3, Fig. 3b). Overall, our results showed that KorA, via its C-terminal α5, directly interacts with the DBD of KorB to lock KorB in an NTD-engaged closed-clamp state.

## KorA blocks KorB diffusion on DNA and increases retention time at operator *OA*

Next, we used optical tweezers to investigate the importance of KorAB interaction for DNA sliding in the presence of CTP. Individual DNA molecules were engineered to contain an *OA* site ~3 kb away from the 8×*OB* sites (Fig. 4a). Without CTP, kymographs showed that fluorescence signals from AF488-labelled KorB and AF647-labelled KorA were confined only to the corresponding positions of *OB* and *OA* sites, respectively (Fig. 4a, top right panel). When CTP was included, the location of the fluorescence signal from AF647–KorA remained confined to *OA* sites. By contrast, AF488–KorB signals were found outside of the *OB* clusters, consistent with KorB–CTP sliding on DNA (Fig. 4a, bottom right panel). However, we observed the accumulation of AF488–KorB fluorescence signal between *OB* and its proximal *OA* site (Fig. 4a, bottom right panel), suggesting that *OA*-bound KorA and/or *OA*-bound KorA–KorB complex might block the sliding of KorB–CTP.

In addition, we constructed another DNA substrate with 8×*OB* and two copies of the *OA* sequence to increase the chances of KorA binding to this site (Fig. 4b and Extended Data Fig. 7a). We identified four distinct cases regarding KorB localization and the simultaneous binding of KorA and KorB in the presence of CTP (Fig. 4b). In case I, KorB localized between the *OA* and *OB* sites. Case I is most frequent at ~45% of the 182 recorded events and consistent with *OA*-bound KorA and/or *OA*-bound KorA–KorB complex blocking the sliding of KorB–CTP. We occasionally observed trespassing of KorB beyond the *OA* site; trespassing is likely due to a transient disruption of the KorA–KorB complex or KorB diffusing beyond *OA* before the complex formation. In case II (~21%), KorB localized between *OA* and the *OB*-proximal bead. We interpret case II as follows: KorB was bound to *OB* and, in the presence of CTP, could freely diffuse in either direction. In both cases I and II, a subsequent KorB protein binding to *OB* might act as a roadblock, preventing a previously loaded KorB from passing through the *OB* site, thereby restricting KorB to either side of *OB*. The higher occurrence of case I compared with case II is likely due to the higher chance of KorB finding KorA within the *OA–OB* region given a particular lifetime of the KorB–DNA interaction. In case III (~25%), there were stable co-localizations of both KorA

**Fig. 4 | KorA can block the diffusion of KorB on DNA, and KorB increases the residence time of KorA at its operator *OA*. a**, Left: schematic of the C-trap optical tweezers experiment with positions of 8×*OB* and 1×*OA* clusters indicated. Right: representative kymograph showing the binding of AF488-labelled KorB and AF647-labelled KorA ± CTP. The upper panel kymograph was taken in a channel containing fluorescently labelled KorAB, hence the higher fluorescence background. The lower panel kymograph was taken in a buffer-only channel to reduce the fluorescence background. **b**, Left: schematic of the C-trap optical tweezers experiment. Right: representative kymographs of AF647–KorA and AF488–KorB in the presence of CTP for the four described cases. The frequency of occurrence for each case is indicated on the kymographs (the total number of recorded events *n* = 182). Kymographs were taken in a buffer-only channel, after a 60 s incubation in the protein channel, to reduce fluorescence background. **c**, Box plot showing the residence times (mean ± s.e.m.) of AF647–KorA alone

(2.9 ± 0.2 s, *n* = 122), in the presence of KorB (2.6 ± 0.2 s, *n* = 70) or KorB–CTP (10.1 ± 0.6 s, *n* = 125), that of AF647–KorA (Y84A) in the presence of KorB–CTP (3.7 ± 0.3 s, *n* = 148), and that of AF647–KorA in the presence of KorB (F249A) and 2 mM CTP (3.1 ± 0.2 s, *n* = 126) (see Extended Data Fig. 7b for representative kymographs). Box plots indicate the median and the 25th and 75th percentiles of the distribution, and whiskers extending to data within 1.5× interquartile range. Outliers are displayed as points beyond the whiskers. **d**, KorA captures the KorB–CTP clamp to heighten long-range transcriptional repression. Top: schematic diagrams of promoter–*xylE* reporter constructs. Bottom: values shown are fold of repression, which is a ratio of XylE activities from cells co-harbouring a reporter plasmid and KorB/A-expressing plasmid to that of cells co-harbouring a reporter plasmid and an empty plasmid (KorA/B-minus control). Data are represented as mean values ± s.d. from three replicates (Extended Data Fig. 8b).

and KorB at *OA*. Case III likely reflects a single diffusing KorB protein binding to KorA. Crucially, we never observed KorB localization at *OA* without KorA. Lastly, in case IV (~9%), we observed mostly static KorA and KorB binding at *OA* and *OB*, respectively.

During the investigation, we noted that the AF647–KorA fluorescence signal showed an on–off behaviour at the *OA* site in the presence of KorB but absence of CTP (Extended Data Fig. 7b). However, the AF647–KorA signal (at *OA*) was more stable over time when KorB and CTP were both included (Extended Data Fig. 7b), suggesting that KorB–CTP might reduce KorA dissociation from *OA*. To test this possibility, we quantified the residence time of AF647–KorA at *OA* either alone or in the presence of unlabelled KorB with or without CTP (Fig. 4c).

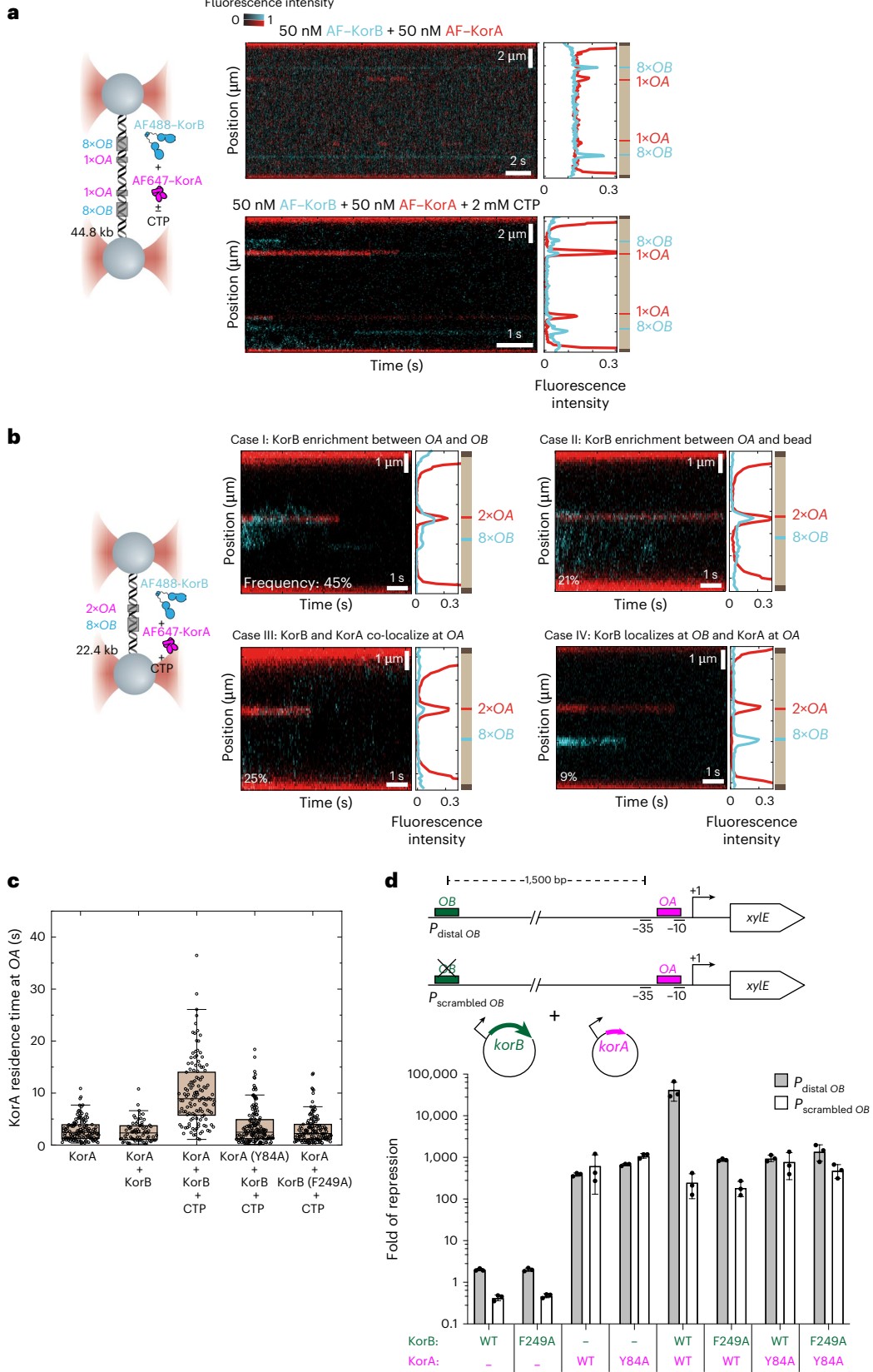

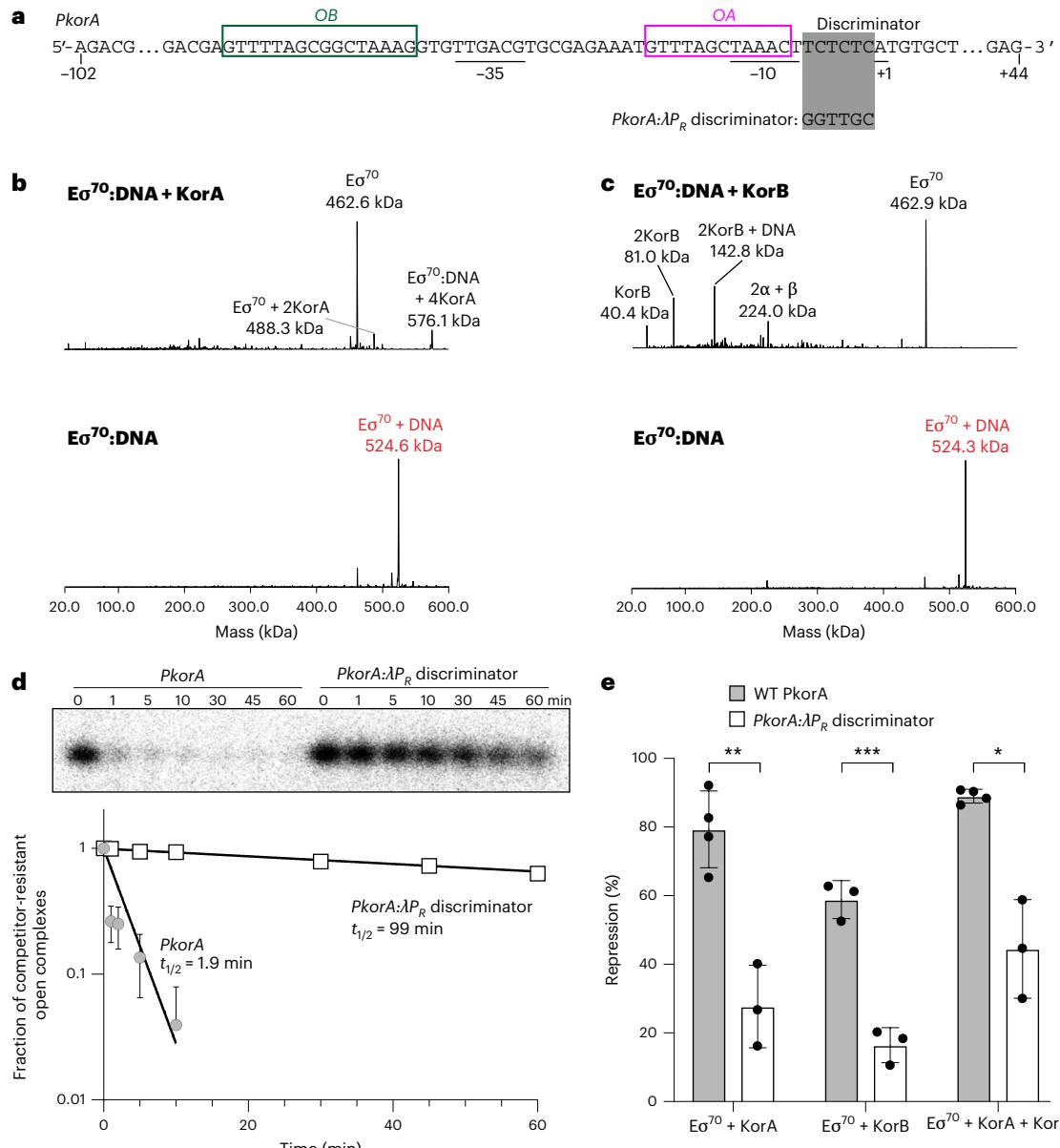

**Fig. 5 | KorA and KorB exploit unstable *E. coli* RNAP:*PkorA* DNA complexes to repress transcription initiation. a**, Promoter scaffold from the RK2 *korABF* operon (*PkorA*) is shown with core promoter elements (underlined), *OA* (magenta box), and *OB* (green box). Differences to the *PkorA*:λ*P_R* discriminator are depicted. **b**, Deconvolved nMS spectra of *E. coli* Eσ70 holoenzyme assembled on 100 bp *PkorA* DNA (Eσ70:DNA) with and without 2.5-fold excess KorA dimer in 150 mM ammonium acetate pH 7.5 and 0.01% Tween-20. **c**, Deconvolved nMS spectra of Eσ70:DNA with and without 2.5-fold excess KorB dimer in 500 mM ammonium acetate pH 7.5 and 0.01% Tween-20. **d**, Top: representative gel close-up on the abortive RNA product (5′-ApUpG-3′) transcribed by Eσ70 in in vitro abortive initiation half-life assays on the two *PkorA* linear scaffold variants. Bottom: plot of fraction of competitor-resistant open complexes (from normalized abortive RNA band intensities) against time. Data points from three

experimental replicates are mean values ± s.e.m. with an exponential trend line fit. Some error bars are too small or lead to negative values and thus were omitted. Estimated half-lives are shown adjacent to exponential decay trend lines. **e**, WT *PkorA* and *PkorA*:λ*P_R* discriminator in vitro transcription repression of Eσ70 in the presence of fivefold excess KorA and/or KorB. Data points from at least three experimental replicates are normalized to holoenzyme only control as mean values ± s.e.m. Eσ70 + KorA and Eσ70 + KorA + KorB conditions on WT *PkorA* has $n = 4$, while the rest are $n = 3$; this remains valid as repression quantities are normalized to a Eσ70-only control and background-corrected for each gel run, and s.e.m. are considered in statistical analysis. *P* values were calculated by unpaired two-tailed Welch's *t*-tests; *$P \leq 0.05$, **$P \leq 0.01$, ***$P \leq 0.001$. The *P* values are $3.66 \times 10^{-3}$ (Eσ70 + KorA), $0.635 \times 10^{-3}$ (Eσ70 + KorB) and $31.5 \times 10^{-3}$ (Eσ70 + KorA + KorB).

The residence time increased ~4.5-fold in the presence of KorB–CTP compared to with or without apo-KorB (Fig. 4c). It is worth noting that this effect was abolished when non-interacting AF488–KorA (Y84A) or KorB (F249A) variants were used instead (Fig. 4c and Extended Data Fig. 7b). Altogether, we suggest that KorA can block the diffusion of KorB on DNA, and KorB increases the residence time of KorA on DNA through their specific interaction as observed in the crystal structure. Finally, magnetic tweezers experiments showed no evidence that

KorB–CTP could condense *OA* and *OB* containing DNA in the presence of KorA (Extended Data Fig. 4).

### KorA captures and locks KorB–CTP clamp converting KorB to a local co-repressor

To investigate the interplay between KorA and KorB–CTP in vivo, we engineered a promoter–*xylE* reporter where the *OB* site is 1.5 kb upstream of the core promoter while *OA* overlaps with the –10 element

(Fig. 4d), mimicking KorAB-regulated promoters natively found on RK2 (ref. 8). KorA and KorB were induced from separate plasmids, and there was not sufficient evidence that their production altered the copy number of the reporter plasmid nor those of the *korAB* expression plasmids (Extended Data Fig. 8a). KorB alone repressed reporter expression weakly by approximately fourfold (Fig. 4d and Extended Data Fig. 8b). This low repression was also observed when KorB (F249A), which does not bind KorA, was expressed alone. Both KorA (WT) and the non-interacting KorA (Y84A) variant repressed transcription ~200-fold when produced alone, consistent with previous findings[24] (Fig. 4d). However, when WT KorB and KorA were co-expressed, transcriptional repression was increased to ~38,000-fold, indicating cooperation between the two proteins[24] (Fig. 4d). This cooperative transcriptional repression was abolished when KorA (WT) and KorB (F249A) or KorA (Y84A) and KorB (WT) were co-produced. As F249A and Y84A substitutions removed the ability of KorA to bind KorB and close the clamp, these activities are essential for effective and efficient transcriptional repression[24].

As KorA could bind apo-KorB (Fig. 3c), we wondered whether DNA-unbound KorB could cooperate with KorA to elicit the same high level of transcriptional repression in vivo. To investigate, we scrambled the distal *OB* site and measured XylE activity when WT KorAB or variants were co-produced (Fig. 4d). In the absence of *OB*, we reasoned that only apo-KorB and CTP-bound KorB (that is, opened clamp) could form inside the cells, while *OB*-stimulated DNA-entrapped KorB–CTP (that is, closed clamp) could not. As expected, in the absence of *OB*, no cooperative transcriptional repression was observed (Fig. 4d). Collectively, these results suggest that *OA*-bound KorA captures a DNA-entrapped sliding clamp of KorB to cooperatively repress promoters.

### KorB–CTP and KorA exploit unstable promoter complexes to repress transcription initiation

The molecular mechanism governing the co-repressive activity of KorAB remains elusive. The proximity of the *OA* site to the core promoter elements of RK2 genes[33,34] suggests a classical RNAP-occlusion mechanism. However, previous work suggested a potential interaction between KorB and RNAP, indicating a repression mechanism stabilizing a closed or intermediate state of a promoter complex where the DNA bubble is incompletely formed[35]. To investigate further, we first used in vitro transcription initiation assays with *Escherichia coli* RNAP and a linear DNA containing a model promoter *PkorA*[36] which features an *OA* site overlapping with the upstream half of its −10 element and an *OB* site directly upstream of the −35 element (Fig. 5a). Transcription initiation assays confirmed that KorA and KorB (with and without CTP) individually function as repressors (Extended Data Fig. 9a,b). The combined activity of KorAB was not significantly greater than KorA alone, but under the conditions of our assay the repression activity of KorA alone was quite high, so observing a significant increase with additional KorB became difficult (Extended Data Fig. 9a,b). Next, to investigate the possible binding of KorA and KorB to *E. coli* Eσ[70]–promoter complexes (Eσ[70]:DNA), we used native mass spectrometry (nMS). Introducing a 2.5-fold excess of KorA or KorB to *PkorA*-bound *E. coli* Eσ[70] (Eσ[70]:DNA) led to near-complete dissociation of Eσ[70] from DNA rather than ternary complex formation (Fig. 5b,c). No corresponding KorB-bound Eσ[70] peaks were observed (Fig. 5b,c), indicating that KorB does not interact with Eσ[70]. Minor Eσ[70]–KorA peaks were detected (Fig. 5b), but these are non-specific as the addition of 2.5-fold excess KorA to Eσ[70] without promoter DNA resulted in no such complex (Extended Data Fig. 9c).

Analysis of the *PkorA* promoter sequence revealed near-consensus −10 (TAAACT; consensus is TATAAT), −35 element (TTGACG; consensus is TTGACA)[37,38] and a consensus 17 bp spacer between the −10 and −35 elements (Fig. 5a). These features highly favour promoter melting by RNAP[39,40]. However, the discriminator (the sequence between the −10 element and the transcription start site) (Fig. 5a), crucial for the stability of an open RNAP–promoter complex (RPo), is highly unfavourable

(TCTCTC; consensus is GGGnnn)[41]. Indeed, the substitution of the WT discriminator with a favoured discriminator from bacteriophage λ $P_R$ (Fig. 5a)[42,43] increased the half-life of *PkorA* RPo from 2 min to over 90 min, stabilizing the RPo over 50-fold (Fig. 5d). The substituted discriminator showed a significant decrease in in vitro transcription repression when KorA and KorB were added individually or combined (Fig. 5e). The instability of the *PkorA* RPo complex, coupled with the observations of KorA and KorB binding to DNA but not to Eσ[70], suggests the following model. *PkorA* RPo is inherently unstable, resulting in frequent RNAP dissociation that allows KorA and KorB to bind their respective operator sites and, upon sliding, form a repressome on the promoter that occludes RNAP from the promoter. Our findings suggest that KorAB exploits RK2 promoter kinetic instabilities (that is, weak discriminators leading to short half-lives) to competitively occlude RNAPs from DNA.

## Discussion

KorB is important for regulating gene expression for basic RK2 functions[7]. Our work has shown that CTP is required for KorB to form an *OB*-dependent sliding clamp on DNA, which enables KorB to travel a long distance to repress distal promoters (Fig. 6). Similar to chromosomal ParB[15,44], we propose that KorB loads at the *OB* site, binds CTP and then switches to a closed-clamp conformation through NTD engagement, allowing it to escape the high-affinity loading site and slide away while entrapping DNA (Fig. 6a).

A sliding clamp is seemingly incompatible with transcription repression. Such a clamp could, for example, slide past the core promoter region, providing access to RNAP. Accordingly, our in vivo transcriptional reporter assay showed that KorB alone only repressed promoters weakly (Fig. 2c). It has also been shown that transcribing RNAPs can steadily traverse the *B. subtilis* ParB–DNA partition complex in vitro[32], and ParB sliding to neighbouring DNA does not affect gene expression in vivo[45]. There are exceptions; for example, plasmid P1 ParB autoregulates its expression[46–48] and *Pseudomonas aeruginosa*, *Vibrio cholerae* and *Streptococcus pneumoniae* ParBs control transcription of neighbouring genes, but again the repression is weak[49–51]. Our data suggest that KorA has a role in switching KorB from a sliding clamp, ineffective at transcription repression, to a sitting clamp which is more effective at repression. Stationary *OA*-bound KorA captures KorB sliding towards it from a distal *OB* site (Fig. 6b). As *OA* is invariably found next to core promoter elements[8], a stationary *OA*-bound KorA–KorB co-complex likely provides a larger and more persistent steric hindrance to RNAP than KorA or KorB alone (Fig. 6b). Furthermore, a KorAB complex improves the retention of KorA at *OA* (Fig. 4c), providing a more stable hindrance to RNAP (Fig. 6a, b). Our finding that KorA binding causes apo-KorB to close the clamp independently of *OB* DNA and CTP (Fig. 3b) suggests that KorA might prolong the closed-clamp state of KorB–CTP, even in the case that bound KorB eventually hydrolyses CTP (Fig. 6a). We speculate that a self-reinforcing interplay between KorA and KorB on DNA cooperatively creates a super-repressive complex. Finally, our data also suggest that this co-repression occurs via a competitive occlusion mechanism (Fig. 5), with KorAB exploiting an intrinsically unstable open RPo to exclude RNAP holoenzymes from the target promoters (Fig. 6b).

Transcriptional regulation by a DNA-sliding clamp, as observed for KorB–CTP here, is reminiscent of the activation of virulence genes in *Shigella flexneri* by a VirB-CTP sliding clamp[52–54] or the activation of T4-phage late genes by a phage-encoded PCNA-like gp45 protein clamp[55,56]. The ability of KorA to trap sliding clamp KorB is functionally similar to eukaryotic CTCF and cohesin, respectively[57], where a DNA-binding protein CTCF restrains the translocation and loop extrusion by cohesin at specific DNA sites[57,58]. Our insights into KorAB here might have an impact beyond the bacterial transcription field, providing a conceptual advance in our understanding of phage, bacterial and eukaryotic transcriptional regulation.

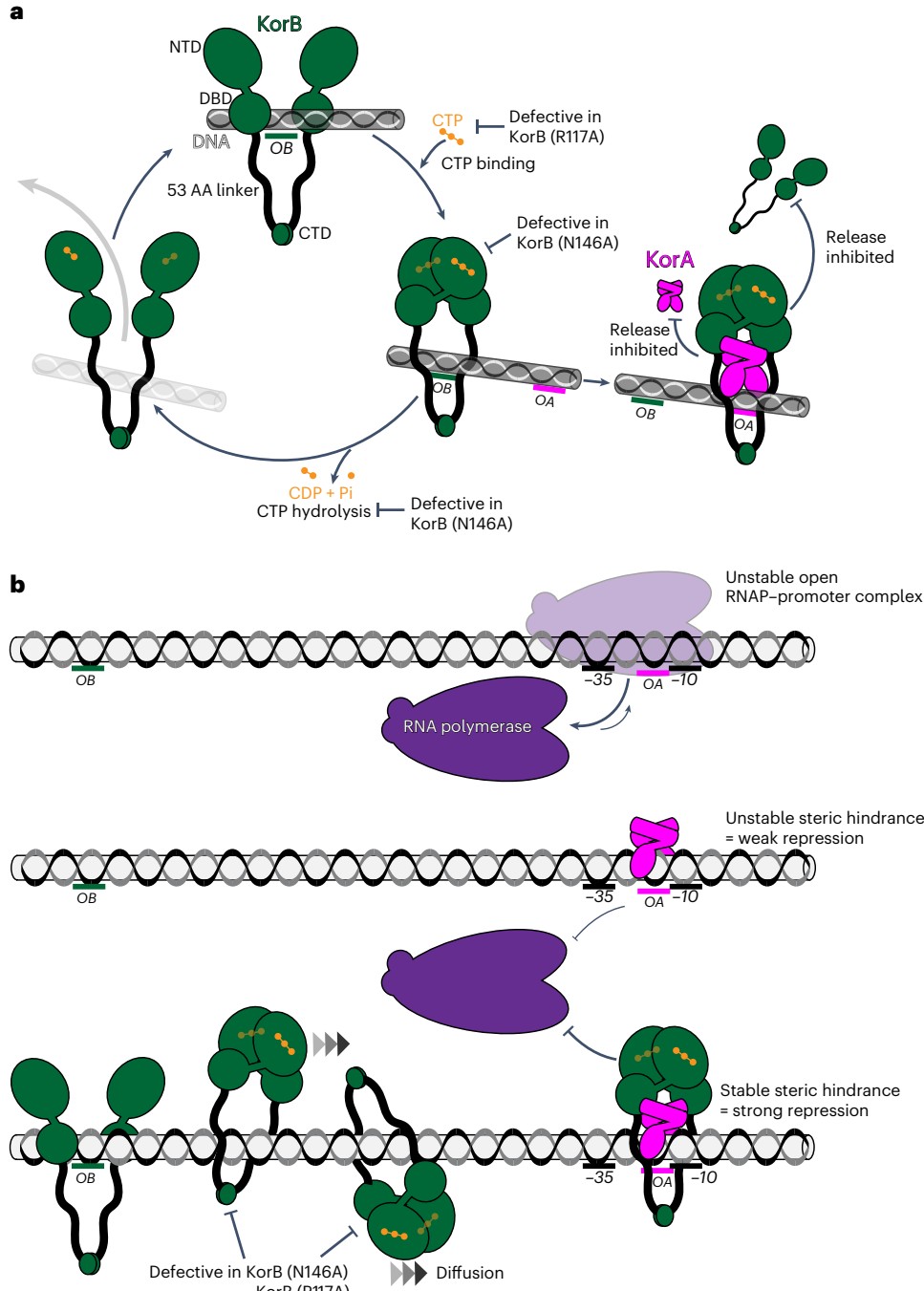

**Fig. 6 | A proposed model for the CTP-dependent clamp-sliding activity of KorB, clamp-locking activity of KorA and their cooperation to heighten long-range transcription repression. a**, KorB (dark green) loading, sliding, and release cycle. Loading KorB is likely an open clamp, in which *OB* DNA binds at the DBD. The presence of CTP (orange) and *OB* DNA likely triggers KorB clamp closing. In this state, KorB can slide away from the *OB* site by diffusion while entrapping DNA. CTP hydrolysis and/or the release of hydrolytic products (CDP and inorganic phosphate Pi) likely reopen the clamp to release DNA. Substitutions that affect various KorB functions are also indicated on the schematic diagram. KorA (magenta) bound at *OA* can form a complex with and promote or trap KorB in a closed clamp state. In this state, KorB most likely still entraps DNA. The tripartite KorAB–DNA reduces the release of KorB from DNA as well as the release of KorA from *OA* DNA. **b**, A model for KorA–KorB cooperation to enhance long-range transcription repression. On RK2, *OB* can

position kilobases away from the target core promoter elements while *OA* is almost invariably near these core promoter elements. Owing to an unfavourable discriminator sequence, the RNAP holoenzyme–promoter DNA complex is inherently unstable. In the absence of KorB–CTP, KorA binds *OA* with a low retention time, thus only providing an unstable steric hindrance to occlude RNAP (magenta) from the core promoter elements, resulting in weak transcriptional repression. In the presence of CTP, KorB loads at a distal *OB* site, binds CTP, closes the clamp and slides by diffusion to reach the distal *OA* site. *OA*-bound KorA captures and locks KorB in a clamp-closed conformation. In this state, the KorAB–DNA co-complex presents a larger and more stable steric hindrance. As a result, the KorAB–DNA co-complex can exploit the unstable RPo and occludes RNAP more effectively, hence stronger transcriptional repression than each protein alone can provide.

## Methods

### Strains, media and growth conditions

*E. coli* strains were grown in lysogeny broth (LB) medium. When appropriate, the media was supplemented with antibiotics at the following concentrations (liquid/solid (µg ml⁻¹)): carbenicillin (50/100), chloramphenicol (20/30), kanamycin (30/50), streptomycin (50/50) and tetracycline (12.5/12.5).

### Plasmid and strain construction

Plasmids and strains used or generated in this study are listed in Supplementary Tables 1 and 2, respectively.

For plasmid construction, a double-stranded DNA fragment containing a desired sequence was chemically synthesized (gBlocks, IDT). The target plasmid was double-digested with restriction enzymes and gel-purified. A 10 µl reaction mixture was created with 5 µl 2× Gibson master mix (NEB) and 5 µl of combined equimolar concentration of purified backbone and gBlock(s). This mixture was incubated at 50 °C for 60 min. Gibson assembly was possible owing to shared sequence similarity between the digested backbone and the gBlock fragment(s). All resulting plasmids were verified by Sanger sequencing (Genewiz) or whole-plasmid sequencing (Plasmidsaurus).

**Construction of pET21b::korB–(his)$_6$ (WT and mutants).** DNA fragments containing mutated *korB* genes (*korB**) were chemically synthesized (gBlocks, IDT). The NdeI-HindIII-cut pET21b plasmid backbone and *korB**gBlocks fragments were assembled using a 2× Gibson master mix (NEB). Gibson assembly was possible owing to a 37 bp sequence shared between the NdeI-HindIII-cut pET21b backbone and the gBlocks fragment.

**Construction of pET21b::korA–(his)$_6$ (WT and mutants).** DNA fragments containing mutated *korA* genes (*korA**) were chemically synthesized (gBlocks, IDT). The NdeI-HindIII-cut pET21b plasmid backbone and *korA**gBlocks fragments were assembled using a 2× Gibson master mix (NEB). Gibson assembly was possible owing to a 37 bp sequence shared between the NdeI-HindIII-cut pET21b backbone and the gBlocks fragment.

**Construction of pBAD33::korB (WT and mutants).** DNA fragments containing mutated *korB* genes (*korB**) were chemically synthesized (gBlocks, IDT). The SacI-HindIII-cut pBAD33 plasmid backbone and *korB** gBlocks fragments were assembled using a 2× Gibson master mix (NEB). Gibson assembly was possible owing to a 38 bp sequence shared between the SacI-HindIII-cut pBAD33 backbone and the gBlocks fragment.

**Construction of pDM1.2::korA (WT and mutants).** DNA fragments containing mutated *korA* genes (*korA**) were chemically synthesized (gBlocks, IDT). The EcoRI-SalI-cut pDM2.1 plasmid backbone and *korA** gBlocks fragments were assembled using a 2× Gibson master mix (NEB). Gibson assembly was possible owing to a 38 bp sequence shared between the EcoRI-SalI-cut pDM2.1 backbone and the gBlocks fragment.

**Construction of pSC101::PkorA (WT and mutants).** DNA fragments containing mutated *PkorA* promoters (*PkorA**) were chemically synthesized (gBlocks, IDT). The BamHI-cut pSC101 plasmid backbone and *PkorA** gBlocks fragments were assembled using a 2× Gibson master mix (NEB). Gibson assembly was possible owing to a 38 bp sequence shared between the BamHI-cut pSC101 backbone and the gBlocks fragment.

**Construction of pUC19::146-bp-PkorA.** A 146 bp DNA fragment containing *PkorA* was chemically synthesized (gBlocks, IDT) and subsequently 5′-phosphorylated using T4 PNK (NEB). Phosphorylated 146 bp *PkorA* DNA was blunt-end ligated with a dephosphorylated SmaI-cut pUC19 using T4 DNA ligase (NEB).

**Construction of pUC19::146-bp-PkorA:λP$_R$.** To clone the 146 bp *PkorA:λP$_R$* discriminator DNA into pUC19, pUC19::146bp-*PkorA* was used as a template for site-directed mutagenesis with Q5 DNA Polymerase (NEB) using the primers 5′-AACATTTCTCGCACG-3′ and 5′-TAGCTAAACTGGTTGCATGTGCTGGCG-3′ at an annealing temperature of 57 °C. The resulting PCR product was introduced into *E. coli* DH5α, and carbenicillin-resistant colonies were isolated. Subsequently, plasmids were isolated and verified by whole-plasmid sequencing (Plasmidsaurus).

**Construction of *E. coli* DH5α and BL21 pLysS strains containing the desired combinations of plasmids.** Plasmids were introduced/co-introduced into *E. coli* DH5α or *E. coli* BL21 pLysS via heat shock transformation (42 °C, 30 s) in the required combinations (Supplementary Table 2).

### Protein overexpression and purification

Proteins used or generated in this study are listed in Supplementary Table 3.

**KorA and KorB (WT and mutants).** Protein preparation for crystallization, ITC and biochemical experiments, excluding BMOE crosslinking, was performed as follows. C-terminally His-tagged KorA and KorB (WT and mutants) were expressed from the plasmid pET21b in *E. coli* Rosetta (BL21 DE3) pLysS competent cells (Merck). Overnight culture (120 ml) was used to inoculate 6 l of LB with selective antibiotics. Cultures were incubated at 37 °C with shaking at 220 r.p.m. until OD$_{600}$ of ~0.6. Cultures were cooled for 2 h at 4 °C before isopropyl-β-ᴅ-1-thiogalactopyranoside (IPTG) was added to a final concentration of 0.5 mM. The cultures were incubated overnight at 16 °C with shaking at 220 r.p.m. before cells were pelleted by centrifugation. Cell pellets were resuspended in buffer A (100 mM Tris–HCl, 300 mM NaCl, 10 mM imidazole, 5% (*v/v*) glycerol, pH 8.0) with 5 mg lysozyme (Merck) and a cOmplete EDTA-free protease inhibitor cocktail tablet (Merck) at room temperature for 30 min with gentle rotation. Cells were lysed on ice with 10 cycles of sonication: 15 s on/15 s off at an amplitude of 20 µm and pelleted at 32,000 *g* for 35 min at 4 °C, and the supernatant filtered through a 0.22 µm sterile filter (Sartorius). Clarified lysate was loaded onto a 1 ml HisTrap HP column (Cytiva) pre-equilibrated with buffer A. Protein was eluted from the column using an increasing gradient of imidazole (10–500 mM) in the same buffer. Desired protein fractions were pooled and diluted in buffer B (100 mM Tris–HCl, 20 mM NaCl, 5% *v/v* glycerol, pH 8.0) until the final concentration of NaCl was 60 mM. Pooled fractions were loaded onto a 1 ml Heparin HP column (Cytiva) pre-equilibrated with buffer B. Protein was eluted from the column using an increasing gradient of NaCl (20–1,000 mM) in the same buffer. Desired protein fractions were pooled and loaded onto a preparative-grade HiLoad 16/600 Superdex 75 pg gel filtration column (GE Healthcare) pre-equilibrated with elution buffer (10 mM Tris–HCl, 150 mM NaCl, pH 8.0). Desired fractions were identified and analysed for purity via sodium dodecyl sulfate–polyacrylamide gel electrophoresis (SDS–PAGE) before being pooled. Aliquots were flash frozen in liquid nitrogen and stored at −80 °C. For protein samples to be used for protein–nucleotide binding ITC experiments, Mg$^{2+}$ was introduced via an overnight dialysis step at 4 °C in buffer containing 10 mM Tris–HCl, 150 mM NaCl and 5 mM MgCl$_2$, pH 8.0 before concentration and quantification as described above.

Protein preparations for BMOE crosslinking were purified using a one-step Ni-affinity column with all buffers adjusted to pH 7.4 for optimal crosslinking. Purified proteins were subsequently desalted using a PD-10 column (Merck) before being concentrated using an Amicon Ultra-4 10 kDa cut-off spin column (Merck). Final protein samples were aliquoted and stored at −80 °C in storage buffer (100 mM Tris–HCl, 300 mM NaCl, 5% *v/v* glycerol, 1 mM Tris(2-carboxyethyl) phosphine, pH 7.4).

Both biological (new sample preparations from a stock aliquot) and technical (same sample preparation) replicates were performed for assays in this study. Protein concentrations were determined by Bradford assay and reported as concentrations of *KorA or KorB dimers*.

***E. coli* His$_{10}$–PPX–RNAP.** Plasmid pVS11 (also called pEcrpoABC(-XH) Z)[59,60] was used to overexpress each subunit of *E. coli* RNAP (full-length α, β, ω) as well as β′-PPX-His$_{10}$ (PPX; PreScission protease site, LEVLFQGP, GE Healthcare) were co-introduced with a pACYCDuet-1::*E.coli rpoZ* into *E. coli* BL21(DE3) to ensure saturation of all RNAPs with *E. coli* RpoZ. Cells were grown in the presence of 100 μg ml$^{-1}$ ampicillin and 34 μg ml$^{-1}$ chloramphenicol to an OD$_{600}$ of ~0.6 in a 37 °C shaker. Protein expression was induced with 1 mM IPTG (final concentration) for 4 h at 30 °C. Cells were collected by centrifugation and resuspended in 50 mM Tris–HCl pH 8.0, 5% *w/v* glycerol, 10 mM dithiothreitol (DTT), 1 mM phenylmethylsulfonyl fluoride (PMSF) and 1× protease inhibitor cocktail. For 200× protease inhibitor cocktail (40 ml volume), the following are dissolved into 100% ethanol: 696 mg PMSF, 1.248 g benzamidine, 20 mg chymostatin, 20 mg leupeptin, 4 mg pepstatin A and 40 mg aprotinin.

After lysis by French press (Avestin) at 4 °C, the lysate was centrifuged twice for 30 min each. Polyethyleneimine (PEI, 10% *w/v* pH 8.0, Acros Organics, Thermo Fisher Scientific) was slowly added to the supernatant to a final concentration of ~0.6% PEI with continuous stirring. The mixture was stirred at 4 °C for an additional 25 min, then centrifuged for 1.5 h at 4 °C. The pellets were washed three times with 50 mM Tris–HCl pH 8.0, 500 mM NaCl, 10 mM DTT, 5% *w/v* glycerol, 1 mM PMSF, and 1× protease inhibitor cocktail. For each wash, the pellets were homogenized and then centrifuged again. RNAP was eluted by washing the pellets three times with 50 mM Tris–HCl pH 8.0, 1 M NaCl, 10 mM DTT, 5% *w/v* glycerol, 1× protease inhibitor cocktail and 1 mM PMSF. The PEI elutions were combined and precipitated overnight with ammonium sulfate at a final concentration of 35% *w/v*. The mixture was centrifuged, and the pellets were resuspended in 20 mM Tris–HCl pH 8.0, 1 M NaCl, 5% *w/v* glycerol and 1 mM β-mercaptoethanol (BME). The mixture was loaded onto two 5 ml HiTrap IMAC HP columns (Cytiva) for a total column volume of 10 ml. RNAP(β′-PPX-His$_{10}$) was eluted at 250 mM imidazole in column buffer (20 mM Tris–HCl pH 8.0, 1 M NaCl, 5% *w/v* glycerol and 1 mM BME). The eluted RNAP fractions were combined and dialysed into 10 mM Tris–HCl pH 8.0, 0.1 mM EDTA pH 8.0, 100 mM NaCl, 5% *w/v* glycerol and 5 mM DTT. The sample was then loaded onto a 40 ml Bio-Rex-70 column (Bio-Rad), washed with 10 mM Tris–HCl pH 8.0, 0.1 mM EDTA, 5% *w/v* glycerol and 5 mM DTT in isocratic steps of increasing concentration of NaCl (eluted at 0.5 M NaCl). The eluted fractions were combined, concentrated by centrifugal filtration, then loaded onto a 320 ml HiLoad 26/600 Superdex 200 column (Cytiva) pre-equilibrated in gel filtration buffer (10 mM Tris–HCl pH 8.0, 0.1 mM EDTA pH 8.0, 0.5 M NaCl, 5% *w/v* glycerol and 5 mM DTT). The eluted RNAP was concentrated to ~8–10 mg ml$^{-1}$ by centrifugal concentration, supplemented with glycerol to 20% *w/v*, flash frozen in liquid nitrogen and stored at −80 °C.

***E. coli* His$_{10}$-SUMO (small ubiquitin-like modifier)-σ$^{70}$.** Plasmid pSAD1403 (ref. [61]) was introduced into *E. coli* BL21(DE3). The cells were grown in the presence of 50 μg ml$^{-1}$ kanamycin to an OD$_{600}$ of ~0.6 at 37 °C. Protein expression was induced with 1 mM IPTG for 1–1.5 h at 30 °C. Cells were collected by centrifugation and resuspended in 20 mM Tris–HCl pH 8.0, 500 mM NaCl, 0.1 mM EDTA pH 8.0, 5 mM imidazole, 5% *w/v* glycerol, 0.5 mM BME, 1 mM PMSF and 1× protease inhibitor cocktail. After lysis by French press (Avestin) at 4 °C, cell debris was removed by centrifugation twice. The lysate was loaded onto two 5 ml HiTrap IMAC HP columns (Cytiva) for a total column volume of 10 ml. His$_{10}$-SUMO-σ$^{70}$ was eluted at 250 mM imidazole in 20 mM Tris–HCl pH 8.0, 500 mM NaCl, 0.1 mM EDTA pH 8.0, 5% *w/v* glycerol and 0.5 mM BME. Peak fractions were combined, cleaved with

Ulp1 protease and dialysed against 20 mM Tris–HCl pH 8.0, 500 mM NaCl, 0.1 mM EDTA pH 8.0, 5% *w/v* glycerol and 0.5 mM BME, resulting in a final concentration of 25 mM imidazole. The cleaved sample was loaded onto one 5 ml HiTrap IMAC HP to remove His$_{10}$-SUMO tag along with any remaining uncleaved σ$^{70}$. Untagged σ$^{70}$ fractions were pooled and diluted with 10 mM Tris–HCl pH 8.0, 0.1 mM EDTA pH 8.0, 5% *w/v* glycerol and 1 mM DTT until the conductivity corresponds to NaCl concentration slightly below 200 mM. The diluted sample was injected onto three 5 ml HiTrap Heparin HP columns (total column volume of 15 ml; Cytiva) which were pre-equilibrated at the same diluent buffer but with 200 mM NaCl, with a gradient to 1 M NaCl, with the first major peak as the target peak. The target peak sample was pooled and concentrated by centrifugal filtration before being loaded onto a HiLoad 16/60 Superdex 200 (Cytiva) which was pre-equilibrated in 20 mM Tris–HCl pH 8.0, 500 mM NaCl, 5% *w/v* glycerol and 1 mM DTT. Peak fractions of σ$^{70}$ were pooled, supplemented with glycerol to a final concentration of 20% *w/v*, flash frozen in liquid nitrogen and stored at −80 °C.

## Protein crystallization
Crystallization screens for both KorBΔN30ΔCTD–CTPγS and KorBΔN30ΔCTD–KorA–*OA* complexes were performed in sitting-drop vapour diffusion format in MRC2 96-well crystallization plates. Drops consisted of 0.3 μl precipitant solution and 0.3 μl protein complex with incubation at 293 K.

**KorBΔN30ΔCTD–CTPγS.** Purified His-tagged KorBΔN30ΔCTD was premixed at 20 mg ml$^{-1}$ with 1 mM MgCl$_2$ and 1 mM CTPγS in buffer containing 10 mM Tris–HCl and 150 mM NaCl, pH 8.0. The KorBΔN30ΔCTD–CTPγS crystals grew in a solution containing 160 mM LiOAc and 2.0 M ammonium sulfate. Suitable crystals were cryoprotected with 20% (*v/v*) ethylene glycol and mounted in Litholoops (Molecular Dimensions). Crystals were flash-cooled by plunging into liquid nitrogen.

**KorBΔN30ΔCTD–KorA–OA.** Purified His-tagged KorBΔN30ΔCTD was combined with purified His-tagged KorA in equimolar concentrations before being purified by gel filtration as described above. The protein complex was premixed at 20 mg ml$^{-1}$ with a 14 bp dsDNA (*OA*, TGTT-TAGCTAAACA) at a molar ratio 1:1.2 (protein complex to DNA) in buffer containing 10 mM Tris–HCl and 150 mM NaCl, pH 8.0. Crystals grew in a solution containing 1.95 M ammonium sulfate and 0.1 M NaOAc, pH 4.6. Suitable crystals were cryoprotected with 25% (*v/v*) glycerol and mounted in Litholoops (Molecular Dimensions). Crystals were flash-cooled by plunging into liquid nitrogen.

## Structure determination and refinement
X-ray data were recorded either on beamline I04-1 or beamline I03 at the Diamond Light Source using either an Eiger2 XE 9M or an Eiger2 XE 16M hybrid photon counting detector (Dectris), respectively, with crystals maintained at 100 K by a Cryojet cryocooler (Oxford Instruments). Diffraction data were integrated and scaled using DIALS (v. 3)[62] via the XIA2 (v. 3.9.dev0) expert system[62] then merged using AIMLESS (v. 0.7.7)[63]. The majority of the downstream analysis was performed through the CCP4i2 (v. 7.1.018) graphical user interface[64]. MolProbity (v. 4.4) was additionally used for validation of 3D atomic models. Data collection statistics are summarized in Supplementary Table 5.

X-ray data for KorBΔN30ΔCTD–CTPγS were collected from a single crystal at a wavelength of 0.9179 Å and processed to 2.3 Å resolution in space group *P*2$_1$2$_1$2$_1$, with approximate cell parameters of *a* = 58.7, *b* = 152.8 and *c* = 198.3 Å. Analysis of the likely composition of the asymmetric unit suggested that it could contain between four and eight copies of the KorB subunit with an estimated solvent content in the range of 43–72%.

Structural predictions for KorB were generated using Alpha-Fold2 (AF2)[65], as implemented through ColabFold[66]. There was good sequence coverage, and the predicted local distance difference test

(pLDDT) scores were generally good (for example, average of 85 from the rank 1 prediction). For a single subunit simulation, the predicted aligned error scores indicated a two-domain structure with very low confidence in the relative placement of the two domains, while for a dimer simulation, the predicted aligned error scores suggested high confidence in the relative placement of all four domains. Consistent with this, all five independently generated models were closely superposable.

The KorBΔN30ΔCTD–CTPγS complex structure was solved via molecular replacement using PHASER[67]. A dimer template was prepared from the rank 1 AF2 model using the 'Process Predicted Models' CCP4i2 task, which removed low-confidence regions (based on pLDDT) and converted the pLDDT scores in the $B$-factor field of the PDB coordinate files to pseudo $B$ factors. Initial attempts used an isomorphous dataset at 3.25 Å resolution. After much trial and error, searching with separate templates corresponding to the two KorB domains showed the most promise. PHASER (v. 2.8.3) produced a partial solution where three pairs of domains were juxtaposed such that they could be connected into single subunits using COOT (v. 0.9.6)[68]. One of the latter was then used as the template for a subsequent run, where PHASER placed five of these in the asymmetric unit (ASU), which were arranged as two dimers and a single subunit. Inspection of this solution in COOT revealed residual electron density adjacent to the latter which could be filled by a sixth subunit by extrapolation from one of the dimers. This final composition of six protomers per ASU gave an estimated solvent content of 57%. When the 2.3 Å resolution dataset became available, the preliminary model was refined directly against this in REFMAC5 (v. 5.8.0403)[69]. At this stage, residual electron density at the interface between the NTDs of each dimer indicated the presence of two symmetry-related ligands. These were built as CTP molecules, as it was not possible to define the locations of the sulfur atoms of CTPγS. The model was completed through several iterations of model building in COOT and restrained refinement in REFMAC5. Pairwise superpositions of the three dimers gave overall r.m.s. deviations of 0.63–1.18 Å, indicating that they were closely similar. Comparison against the AF2 dimer model showed that this experimental structure had a domain-swapped arrangement, while the predicted structure did not. Refinement and validation statistics are summarized in Supplementary Table 5.

X-ray data for KorBΔN30ΔCTD–KorA–*OA* were collected from a single crystal at a wavelength of 0.9763 Å (2 × 360° passes) and processed to 2.7 Å resolution in space group *C*2, with approximate cell parameters of $a = 173.2$, $b = 77.1$, $c = 84.6$ Å and $\beta = 107.4°$. Analysis of the likely composition of the asymmetric unit suggested that it contained a single complex comprising two copies of each of the KorA and KorB subunits and a single DNA duplex, giving an estimated solvent content of 61%.

The structure was solved via molecular replacement using PHASER. Separate templates were prepared for the two KorB domains from the A chain of the above KorBΔN30ΔCTD–CTPγS complex, for the KorA subunit by taking a single chain from the previously solved KorA–DNA complex (PDB code 2W7N)[70] and for the DNA by generating an ideal B-form DNA duplex in COOT from the palindromic sequence TGTTTAGCTAAACA. PHASER was able to locate the four domains expected for a KorB dimer and the DNA duplex, but not the two KorA subunits. However, after refinement in REFMAC5, a clear difference density was visible for the missing KorA subunits. These could be manually placed from a superposition of the KorA–DNA complex. Several sulfates were built into the density, presumably derived from the precipitant. Two of these occupied positions equivalent to the β-phosphates of the CTP ligands in the previous structure. The model was completed through several iterations of model building in COOT and restrained refinement in REFMAC5. In contrast to the KorB dimer from the CTP complex, this dimer does not have a domain-swapped architecture. Moreover, a superposition of this KorB dimer onto the AF2 dimer model revealed that they were almost identical at the protein backbone level, giving

an overall r.m.s. deviation of only 1.01 Å. Refinement and validation statistics are summarized in Supplementary Table 5.

## DNA preparation for in vitro NTPase, crosslinking and ITC experiments

Palindromic single-stranded DNA oligonucleotides (*OB*, GGGAT<u>ATTT-TAGCGGCTAAAAGG</u>A; *OA*, <u>TGTTTAGCTAAACA</u>) (100 µM in 1 mM Tris–HCl pH 8.0, 5 mM NaCl buffer) were heated at 98 °C for 5 min before being left to cool down to room temperature overnight to form 50 µM dsDNA. The core sequence of *OB* or *OA* is underlined.

## Measurement of CTPase activity by EnzChek phosphate release assay

CTP hydrolysis was monitored using an EnzCheck Phosphate Assay Kit (Thermo Fisher Scientific). Samples (100 µl) containing a reaction buffer supplemented with an increasing concentration of CTP (1, 5, 10, 50, 100, 500 and 1,000 µM), 0.5 µM of 24 bp *OB* DNA, and 1 µM dimer concentration of KorB (WT or mutants) were assayed in a CLARIOstar Plus plate reader (BMG Labtech) at 25 °C for 5 h with readings every 2 min with continuous orbital shaking at 300 r.p.m. between reads. The reaction buffer (1 ml) typically contained 740 µl ultrapure water, 50 µl 20× reaction buffer (100 mM Tris–HCl, 2 M NaCl and 20 mM MgCl$_2$, pH 8.0), 200 µl 2-amino-6-mercapto-7-methylpurine riboside (MESG) substrate solution and 10 µl purine nucleoside phosphorylase enzyme (one unit). Reactions with buffer only or buffer + CTP + 24 bp *OB* DNA only were also included as controls. The inorganic phosphate standard curve was also constructed according to the instruction guidelines. The results were analysed using Excel (v. 16.92) and plotted in GraphPad Prism (v. 9.5.1). The CTPase rates were calculated using a linear regression fitting in GraphPad Prism 9. Error bars represent standard deviations from triplicate experiments.

## In vitro crosslinking assay using a sulfhydryl-to-sulfhydryl crosslinker BMOE

A 50 µl mixture of 8 µM dimer concentration of KorB WT or mutants ± 1 mM NTP ± 0.5 µM 24 bp dsDNA containing *OB* or scrambled *OB* was assembled in a reaction buffer (10 mM Tris–HCl pH 7.4, 100 mM NaCl and 1 mM MgCl$_2$) and incubated for 5 min at room temperature. BMOE was added to a final concentration of 1 mM, and the reaction was quickly mixed by three pulses of vortexing. The reaction was then immediately quenched through the addition of SDS–PAGE sample buffer containing 23 mM BME. Samples were heated to 50 °C for 5 min before being loaded on 12% Novex WedgeWell Tris-Glycine gels (Thermo Fisher Scientific). Protein bands were stained with an InstantBlue Coomassie protein stain (Abcam), and band intensity was quantified using ImageJ (v. 2.14.0/1.54f). The results were analysed in Excel and plotted using GraphPad Prism 9.

For the experiments containing KorA in addition to KorB, an equimolar amount was used (8 µM of WT or mutants, dimer concentration). The reaction was otherwise assembled identically and loaded on 4–20% Novex WedgeWell Tris-Glycine gels (Thermo Fisher Scientific) for sufficient separation of KorA in the samples. Band intensity was quantified using ImageJ, and the results were analysed in Excel and plotted using GraphPad Prism 9.

## ITC

All ITC experiments were recorded using a MicroCal PEAQ ITC instrument (Malvern Panalytical). Experiments were performed at 25 °C. For protein–nucleotide binding experiments, all components were in 10 mM Tris–HCl, 150 mM NaCl and 5 mM MgCl$_2$, pH 8.0 buffer. For protein–protein binding experiments, all components were in 10 mM Tris–HCl and 150 mM NaCl, pH 8.0 buffer. For each ITC run, the calorimetric cell was filled with 20 µM dimer concentration of KorB (WT or mutant), and a single injection of 0.4 µl of 500 µM small-molecule nucleotides or 200 µM protein partner was performed first, followed

by 19 injections of 2 µl each. Injections were carried out at 150 s intervals with a stirring speed of 750 r.p.m. The raw titration data were integrated and fitted to a one-site binding model using the built-in software of the MicroCal PEAQ ITC instrument. Each experiment was run in duplicate. Controls of ligand into buffer and buffer into protein were performed with no signal observed. Where required, representative data are presented.

### *xylE* reporter gene assays

Reporter gene assays were carried out using a modified version of ref. 71. In short, *E. coli* DH5α cells containing relevant expression plasmids were grown to a logarithmic phase from a 1:100 dilution of overnight culture. Induction of KorB WT/mutant expression was achieved with 0.2% (*PkorA*; Fig. 5e) and 0.02% (*PtrbB*; Fig. 3) arabinose. In the three plasmid experiments, induction of KorA WT/Y84A required no additional IPTG. About 10 or 50 ml of culture was pelleted and resuspended in 500 µl resuspension buffer (0.1 M sodium phosphate buffer pH 7.4, 10% $v/v$ acetone). From this point onwards, samples were kept on ice. Cells were disrupted using sonication at 10 µm for 10 s and subsequently pelleted. The supernatant was transferred to a fresh microcentrifuge tube and assayed for catechol 2,3-oxygenase activity. Samples were diluted 1:10 in reaction buffer (0.1 M sodium phosphate buffer pH 7.4, 200 µM catechol) and incubated at room temperature for 1 min before the absorbance at 374 nm was determined using a BioMate 3 spectrophotometer (Thermo Fisher Scientific). Protein concentration, determined using Bradford assay, was used to normalize the samples. The results were analysed in Excel and plotted using GraphPad Prism 9.

### Immunoblot analysis

For western blot analysis samples, 200 ng total protein lysate was resuspended in 1× SDS–PAGE sample buffer and heated to 95 °C for 10 min before loading. Denatured samples were run on 12% Novex WedgeWell gels (Thermo Fisher Scientific) at 150 V for 55 min. Resolved proteins were transferred to PVDF membranes using the Trans-Blot Turbo Transfer System (BioRad) and incubated with a 1:5,000 dilution of α-KorB primary antibody (Cambridge Research Biochemicals) or with 1:300 dilution of α-KorA. Membranes were washed and subsequently probed with a 1:10,000 dilution of mouse α-rabbit HRP-conjugated secondary antibody (Abcam). Blots were imaged after incubation with SuperSignal West PICO PLUS Chemiluminescent Substrate (Thermo Fisher Scientific) using an Amersham Imager 600 (GE HealthCare). Loading controls of denatured 200 ng total protein lysate were run on 12% Novex WedgeWell gels (Thermo Fisher Scientific) at 150 V for 55 min and stained with InstantBlue Coomassie protein stain (Abcam).

### Protein labelling with Alexa Fluor for confocal optical tweezers (C-trap) experiments

C-terminally His-tagged versions of KorB (A6C) and KorA (WT/Y84A variant) with an extra cysteine residue at the C-terminus were coupled to maleimide-conjugated Alexa Fluor (AF) 488 and 647, respectively. A6C was selected because it resides in a surface-exposed intrinsically disordered region at the N-terminal region of KorB. His-tagged KorB (A6C) and His-tagged KorA-extra C were purified as described for the WT proteins. About 250 µl of 50 µM KorB (A6C) or KorA-extra C were incubated with 0.3 mM tris-carboxyethyl phosphine for 30 min at room temperature in a buffer containing 10 mM Tris–HCl and 300 mM NaCl, pH 7.4. Subsequently, 6 µl of 30 mM AF488 or AF647 (dissolved in DMSO) was added, and the reaction was incubated with rotation at 4 °C overnight. The conjugate solution was then loaded onto a Superdex increase 200 pg 10/300 gel filtration column (pre-equilibrated with 10 mM Tris–HCl, 300 mM NaCl, pH 8.0) to separate labelled KorB/A from unincorporated fluorophore. AF-labelled KorB/A was pooled and concentrated before storage as described for WT KorB and KorA.

### Design and construction of a DNA plasmid with 16×*OB* sites and 1×*OA* site for magnetic tweezers experiments

A DNA plasmid containing 16×*OB* sites (ATTTTAGCGGCTAAAAG) and 1×*OA* site (AATGTTTAGCTAAACCTT) was produced by modification of a pUC19 plasmid containing a single copy of each site separated by 1,016 bp (pUC19_v1), following several cloning steps and methods described elsewhere[29,72].

First, the original pUC19 plasmid (4,886 bp) with one of each site was enlarged by introducing a piece of DNA obtained from a lab plasmid. This resulted in a larger plasmid (7,699 bp) which, after digestion with appropriate restriction enzymes, produces the central part of a magnetic tweezers DNA construct with centred *OB* and *OA* sites.

To increase the number of *OB* sites, two long oligonucleotides (Supplementary Table 4) containing 2×*OB* sites separated by 40 bp with a PshAI restriction site in the middle of this region were annealed by heating at 95 °C for 5 min and cooling down to 20 °C at a rate of −1 °C min$^{-1}$ in hybridization buffer (10 mM Tris–HCl pH 8.0, 1 mM EDTA, 200 mM NaCl and 5 mM MgCl$_2$) followed by a phosphorylation step of the 5′-terminal ends by the T4 PNK (NEB). This dsDNA duplex was ligated into the previous plasmid of 7,699 bp that already contained 1×*OA* and 1×*OB* sites digested with PshAI restriction enzyme (NEB) and dephosphorylated. These oligonucleotides were designed to lose the original PshAI site at both ends after ligation, so that once ligated into a cloning plasmid they could not be cleaved again by PshAI. The single bona fide PshAI site located in the middle of the duplex allows for repetition of the ligation process to be repeated as many times as desired in the cloning plasmid to add new pairs of *OB* sites. Plasmids containing 1×*OA* site and up to 8×*OB* have been obtained following this procedure. Note that in one of the rounds of cloning and by chance, half of a previous duplex was lost during the ligation process, and therefore the final plasmid contains 8×*OB* instead of 9×*OB* as expected.

A plasmid with 1×*OA* site and 16×*OB* was produced by PCR amplifying an 8×*OB* cassette with Phusion High-Fidelity DNA Polymerase (Thermo Scientific) (see Supplementary Table 4 for primer sequences). The PCR fragment was then digested with SpeI and XhoI (both from NEB) and ligated into the plasmid already containing 1×*OA* site and 8×*OB* copies previously digested with XbaI (NEB) and XhoI and dephosphorylated. This resulted in a plasmid with 1×*OA* site and 16×*OB* sites (8,705 bp). Plasmids were introduced into *E. coli* DH5α competent cells, and potentially positive colonies were then selected by colony PCR. Plasmids were purified from the cultures using a QIAprep Spin Miniprep Kit (QIAGEN), analysed by restriction enzyme digestion and finally verified by Sanger sequencing. This plasmid was subsequently used to produce a magnetic tweezers dsDNA construct.

### Construction of large plasmids with different combinations of *OB* and *OA* sites for confocal optical tweezers (C-Trap) experiments

C-Trap experiments were performed on three types of molecule: one containing only a single copy of the *OA* site without any *OB* site and two different versions containing 1×*OA* or 2×*OA* sites together with 8×*OB* sites. Therefore, different large plasmids were cloned. First, a large DNA plasmid with 1×*OA* site was fabricated by ligating a DNA fragment containing a single copy of the *OA* site into a previously prepared large plasmid in our laboratory that did not contain either of these sites. The fragment containing the *OA* site was obtained by PCR amplification using Phusion High-Fidelity DNA Polymerase with the pUC19 plasmid containing a single copy of the *OB* and *OA* sites as a template (see Supplementary Table 4 for sequences of oligonucleotides). The PCR fragment was digested with KpnI (NEB) and ligated into the previously large plasmid prepared in our laboratory that did not contain either of these sites, digested with KpnI and dephosphorylated. This resulted in a large plasmid with a single *OA* site (20,985 bp). A DNA fragment with 8×*OB* sites was then inserted over this new large plasmid. The 8×*OB* fragment was obtained by PCR amplification (see Supplementary

Table 4 for sequences of oligonucleotides) using Phusion High-Fidelity DNA Polymerase with the magnetic tweezers plasmid that contained 1×*OA* and 8×*OB* sites as a template. The PCR fragment was digested with SpeI and XbaI and ligated into the large plasmid, which already contained 1×*OA* site, digested with XbaI and dephosphorylated, resulting in a large plasmid with 1×*OA* site and 8×*OB* sites (22,394 bp). The large DNA plasmid containing 8×*OB* sites and 2×*OA* sites (22,733 bp) was produced by inserting a new copy of the *OA* site into the previously described plasmid digested with NruI (NEB). A new 339 bp dsDNA fragment with a copy of the *OA* site was obtained by digestion of the pUC19 plasmid containing a single copy of the *OB* and *OA* sites with SfoI and HpaI (NEB). The dsDNA fragment was gel extracted, purified and ligated into the NruI-linearized large plasmid described before. The plasmids were cloned and analysed as described before for magnetic tweezers plasmids, looking for a final plasmid with the new *OA* site in the same orientation as the previous one. These plasmids were subsequently used to prepare various C-Trap dsDNA constructs.

## Magnetic tweezers dsDNA construct with 16×*OB* and 1×*OA* sites

The dsDNA construct for magnetic tweezers experiments consisted of a central dsDNA fragment of 8,693 bp containing 16×*OB* and 1×*OA* sites, obtained by digestion with NotI and ApaI (NEB) of the final magnetic tweezers plasmid described above, flanked by two highly labelled DNA fragments, one with digoxigenins and the other with biotins, of 997 bp and 140 bp, respectively, used as immobilization handles. The biotinylated handle was shorter to minimize the attachment of two beads per DNA tether. Handles for magnetic tweezers constructs were prepared by PCR (see Supplementary Table 4 for sequences of oligonucleotides) with 200 μM final concentration of each dNTP (dGTP, dCTP, dATP), 140 μM dTTP and 66 μM Bio-16-dUTP or Dig-11 dUTP (Roche) using plasmid pSP73-JY0 (ref. [73]) as template, followed by digestion with the restriction enzyme ApaI or PspOMI (NEB), respectively. Labelled handles were ligated to the central part overnight using T4 DNA Ligase (NEB). The sample was then ready for use in magnetic tweezers experiments without further purification. The DNAs were never exposed to intercalating dyes or UV radiation during their production and were stored at 4 °C. The sequence of the central part of the magnetic tweezers construct is included in Supplementary Table 6.

## C-Trap dsDNA constructs with different combinations of *OB* and *OA* sites

C-Trap experiments were performed on three types of molecule: one containing a single copy of the *OA* site without any *OB* sites and two different versions containing 1×*OA* or 2×*OA* sites together with 8×*OB* sites. The C-Trap dsDNA construct consisting of a large central fragment of 22,394 bp containing 8×*OB* and 1×*OA* sites was produced by digestion of the large C-trap plasmid with NotI. Without further purification, the fragment was ligated to highly biotinylated handles of ~1 kbp ending in PspOMI. Handles for C-Trap constructs were prepared by PCR (see Supplementary Table 4 for sequences of oligonucleotides) as described for biotin-labelled magnetic tweezers handles. These handles were highly biotinylated to facilitate the capture of DNA molecules in the C-Trap experiments. As both sides of the DNA fragment end in NotI, it is possible to generate tandem (double length) tethers flanked by the labelled handles. The sample was ready for use in C-Trap experiments without further purification. The dsDNA constructs with 8×*OB* and 2×*OA* sites (22,733 bp) or with only a 1×*OA* site (20,985 bp) were equally prepared, but they were obtained by digestion with NotI of the other large C-Trap plasmids described above: the one with 8×*OB* and 2×*OA* sites or the one with a single copy of the *OA* site without any *OB* sites, respectively. The DNAs were not exposed to intercalating dyes or UV radiation during their production and were stored at 4 °C. The sequence of the central part of the C-Trap construct is included in Supplementary Table 6.

## Confocal optical tweezers experiments

Confocal optical tweezers experiments were carried out using a dual optical tweezers set-up combined with confocal microscopy and microfluidics (C-Trap; Lumicks)[74,75]. A computer-controlled stage allowed rapid displacement of the optical traps within a five-channel fluid cell, allowing the transfer of the tethered DNA between different channels separated by laminar flow. Channel 1 contained 4.38 μm streptavidin-coated polystyrene beads (Spherotech). Channel 2 contained the DNA substrate labelled with multiple biotins at both ends. Both DNA and beads were diluted in 20 mM HEPES pH 7.8, 100 mM KCl and 5 mM MgCl$_2$. A single DNA tether was assembled by first capturing two beads in channel 1, one in each optical trap, and fishing for a DNA molecule in channel 2. The tether was then transferred to channel 3 filled with reaction buffer (10 mM Tris pH 8, 100 mM NaCl, 5 mM MgCl$_2$ and 1 mM DTT) to verify the correct length of the DNA by force–extension curves. The DNA was then incubated for 1 min in channel 4 filled with KorB and/or KorA proteins in reaction buffer and supplemented with 2 mM CTP as indicated. To reduce the fluorescence background in single KorB diffusion measurements, imaging was occasionally performed in channel 3 after protein incubation in channel 4 (as indicated in figure legends). All the fluorescence intensities in kymograms were normalized, and the scales in the intensity profiles were adjusted for better visualization. Scale bars in figures represent fluorescence intensity on the kymographs.

The system is equipped with three laser lines for confocal microscopy (488, 532 and 635 nm). In this study, the 488 nm laser was used to excite AF488–KorB and the 635 nm laser to excite AF647–KorA, with emission filters of 500–525 nm and 650–750 nm, respectively. Protein-containing channels were passivated with BSA (0.1% $w/v$ in PBS) for 30 min before the experiment. Kymographs were generated by single line scans between the two beads using a pixel size of 100 nm and a pixel time of 0.1 ms, resulting in a typical time per line of 22.4 ms. The confocal laser intensity at the sample was 2.2 μW for the 488 laser and 1.92 μW for the 635 laser. Experiments were performed in constant-force mode at 15 pN.

## Magnetic tweezers experiments

Magnetic tweezers experiments were performed using a homemade set-up that was previously described[76,77]. Briefly, optical images of micron-sized superparamagnetic beads tethered to a glass surface by DNA substrates were acquired using a ×100 oil immersion objective and a CCD camera operating at 120 Hz. Real-time image analysis allows the spatial coordinates of the beads to be determined with nanometre accuracy in the $x$, $y$ and $z$ directions. We controlled the stretching force of the DNA by using a step motor coupled to a pair of magnets located above the sample. The applied force is quantified from the Brownian motion of the bead and the extension of the tether, obtained by direct comparison of images taken at different focal planes[78,79].

Magnetic tweezers experiments were performed as follows. The DNA sample containing 16×*OB* sites was diluted in 10 mM Tris–HCl pH 8.5 and 1 mM EDTA and mixed with 1-μm-diameter magnetic beads (Dynabeads, MyOne Streptavidin, Invitrogen) for 10 min. Magnetic beads were previously washed three times with PBS and resuspended in PBS/BSA at a 1:10 dilution. The DNA to bead ratio was adjusted to obtain as many single-tethered beads as possible. After incubation, we introduced the DNA–bead sample in a double-PARAFILM (Sigma)-layer flow cell and allowed them to sink for 10 min to promote the binding of the digoxigenin (DIG)-labelled end of the DNA to the anti-DIG glass-coated surface. Then, a force of 5 pN was applied to remove non-attached molecules from the surface. The chamber was washed with ~500 μl of PBS before experiments. Torsionally constrained molecules and beads containing more than a single DNA molecule were identified from their distinct rotation–extension curves and discarded for further analysis. Force–extension curves were generated by measuring the extension of the tethers at decreasing forces from 5.5 pN to 0.002 pN. The curves

were first measured on naked DNA molecules, and then the experiment was repeated using different concentrations of *B. subtilis* ParB and KorB ± KorA in a reaction buffer (10 mM Tris pH 8, 100 mM NaCl, 5 mM MgCl$_2$, 1 mM DTT and 0.1 mg ml$^{-1}$ BSA) supplemented with 2 mM CTP. Data were analysed and plotted using OriginPro (v. 2022b) software.

### Construction of *PkorA* and *PkorA*:λ*P$_R$* linear scaffolds

The plasmid pUC19::146-bp-*PkorA* was used as a template to amplify a 146 bp linear *PkorA* DNA scaffold using primers 5′-AGACGAAAGCCCGGTTTCCGGG-3′ and 5′-CTCCGCGCCTTGGTTGAA CATAG-3′ in a PCR reaction with *Taq* DNA polymerase (Promega) at an annealing temperature of 65 °C (Supplementary Table 4). The correct band was gel extracted from a 2% *w/v* agarose gel and eluted into TE buffer. The plasmid (pUC19-146-bp-*PkorA*:λ*P$_R$*) was similarly used as a template to amplify a linear 146 bp *PkorA*:λ*P$_R$* DNA scaffold (Supplementary Table 1).

### Construction of a *PkorA* linear scaffold for nMS

The *PkorA* sequence was shortened to its minimal elements as a 100 bp DNA scaffold and synthesized as separate PAGE-purified top and bottom strand oligos (IDT) (Supplementary Table 6). The two strands were resuspended separately to 1 mM solutions in 10 mM Tris−HCl pH 8.0, 50 mM NaCl and 1 mM EDTA, pH 8.0. The strands were mixed in a 1:1 molar ratio for a 500 μM dsDNA (final concentration) and were heated to 95 °C before being cooled down to 25 °C in a 1 °C stepwise decrease using a Thermocycler PCR machine (Eppendorf). The resulting dsDNA was assayed by 2% *w/v* agarose gel electrophoresis for purity and was quantified using a Qubit (Invitrogen) dsDNA broad-range quantification kit.

### Construction of a promoter bubble DNA for half-life abortive initiation assays

An ideal promoter bubble DNA (generated with a non-complementary intervening sequence) was used as a competitor DNA for the in vitro half-life abortive initiation assays (Supplementary Table 6)[80]. The top and bottom strands were synthesized and annealed as described for WT *PkorA* 100-bp DNA scaffold used in nMS.

### Electrophoretic mobility shift assays

To check for *E. coli* Eσ$^{70}$ binding to the *PkorA* DNA scaffolds, 50 nM of *PkorA* DNA scaffold was mixed with 50 nM *E. coli* Eσ$^{70}$ and incubated at 37 °C for 5 min. In all biochemical experiments except nMS, Eσ$^{70}$ was assembled by incubating *E. coli* His$_{10}$−PPX−RNAP with a fivefold molar excess of σ$^{70}$ at 37 °C for 15 min (excess σ$^{70}$ was not purified away). The Eσ$^{70}$−DNA sample was then loaded onto a 4.5% native Tris−borate−EDTA (90 mM TBE pH 8.3) polyacrylamide gel and run at a constant current of 15 mA for 1.5 h at 5–10 °C in a cold room. The gel was stained for dsDNA using GelRed (Biotium).

### In vitro abortive initiation assay for transcription initiation repression

To assay for transcription initiation repression, we monitored levels of abortive initiation RNA products in an NTP-restricted in vitro promoter-based transcription reaction. The 146 bp WT *PkorA* and *PkorA*:λ*P$_R$* discriminator linear DNA scaffolds were used to assay KorA and KorB-mediated repression on WT and mutant DNA-derived transcription, respectively. *E. coli* Eσ$^{70}$ was assembled as described for electrophoretic mobility shift assays. The transcription buffer consists of 50 mM Tris−HCl pH 8.0, 10 mM MgCl$_2$, 150 mM KCl, 0.1 mg ml$^{-1}$ BSA and 1 mM DTT. Assembly of protein−DNA mixes before the addition of NTP mix involved a sequential incubation of DNA (10 nM), *E. coli* Eσ$^{70}$ (50 nM) and KorAB factors (250 nM; and saturating CTP where relevant) at 37 °C in 5 min incubations. NTP mix (50 μM GTP, 250 μM ApU dinucleotide, 0.05 μCi per μl reaction volume of α-$^{32}$P-GTP (PerkinElmer)) was added once all factors were added and incubated for 10 min at 37 °C to synthesize one abortive RNA band (5′-ApUpG*-3′; +1 to +3 RNA). Reactions

were quenched using a 2× STOP buffer (45 mM TBE pH 8.3, 8 M urea, 30 mM EDTA pH 8.0, 0.05% *w/v* bromophenol blue and 0.05% *w/v* xylene cyanol). Reaction samples were analysed on a 23% denaturing PAGE (19:1 acrylamide to bis-acrylamide, 90 mM TBE pH 8.3, 6 M urea) for 1.5–2 h at a constant voltage of 800 V, and the gel was exposed to a storage phosphor screen for 1–2 h and imaged using a Typhoon Phosphorimager (Cytiva). Band intensities were quantified on ImageJ[81], with measured values subtracted of the background and normalized to an *E. coli* Eσ$^{70}$−DNA only control (no repression) for values to be averaged among replicates (*n* of at least 3 in all repression conditions). Repression as percentage values was calculated as (1-normalized intensity) × 100%, graphed and statistically analysed in GraphPad Prism using unpaired Welch's *t*-tests.

### In vitro abortive initiation assay for half-life estimation

To estimate half-lives of *E. coli* Eσ$^{70}$-open promoter complexes on both the WT and mutant *PkorA* DNA scaffolds, we adapted the in vitro abortive initiation assay by including a competitor DNA to compete with the promoter DNA of interest. In this setup, the assay is similar to the previous abortive initiation assay[80], except that competitor DNA is added in the beginning, and NTP mix is added afterwards at several varying timepoints. Eσ$^{70}$ was incubated in transcription buffer for 5 min at 37 °C before adding promoter DNA scaffold and incubated for another 15 min at 37 °C. This formed a master mix to which competitor DNA (100 nM final concentration) was added (*t* = 0 s was transcribed as a separate reaction) and was incubated at 37 °C throughout. Aliquots were taken out at desired time points (*t* = 0, 1, 2, 5, 10, 30, 45, 60 min), to which NTP mix was added to start transcription of abortive RNAs. In this way, abortive RNA production is a proxy for the proportion of open promoter complexes remaining as time goes on, as Eσ$^{70}$ that dissociates from the promoter binds to the ideal bubble competitor DNA and does not easily dissociate (re-binding to the *PkorA* scaffold is thereby negligible). Quantification of abortive RNA bands was the same as performed for the repression assays[80]. Plots of normalized band intensities against time resulted in a double exponential curve, with the slow-decaying component (that is, basal noise-level signal) dominating at time points with low fractions of competitor-resistant open promoter complexes. Due to the rapid decline in WT *PkorA* open complexes, a correction was applied by subtracting values that occur at *t* ≥ 30 min to remove the slow-decaying component and derive the true half-life from the fast decay as a single exponential decay curve. This correction was not applied to *PkorA*:λ*P$_R$* discriminator open complex values as the time points measured did not decline to noise-level signal.

The plots were presented on a semi-log scale, with the fraction of competitor-resistant open promoter complexes on a log$_{10}$ scale and time on a linear scale, with single exponential decay trend lines fitted (WT *PkorA* with $R^2$ = 0.7637; *PkorA*:λ*P$_R$* discriminator with $R^2$ = 0.9934). Analysis and plotting were performed using Microsoft Excel (v. 16.92) and GraphPad Prism (v. 9.5.1).

### Preparation of *E. coli* Eσ$^{70}$ holoenzyme for nMS

Eσ$^{70}$ was formed by mixing purified His$_{10}$−PPX−RNAP and a 2.5-fold molar excess of σ$^{70}$ and incubated for 15 min at 37 °C. Eσ$^{70}$ was buffer exchanged into 20 mM Tris−HCl pH 8.0, 500 mM NaCl and 5 mM DTT (to remove most glycerol from protein storage buffer) by centrifugal filtration and purified on a Superose 6 Increase 10/300 GL column (Cytiva) pre-equilibrated in 50 mM Tris−HCl pH 8.0, 150 mM KCl, 10 mM MgCl$_2$ and 1 mM DTT. The eluted Eσ$^{70}$ was concentrated to ~10–12 mg ml$^{-1}$ (~21–26 μM) by centrifugal filtration. Purified Eσ$^{70}$ was supplemented with glycerol to a final concentration of 20% *w/v*, flash frozen in liquid nitrogen and stored at −80 °C.

### nMS analysis

Frozen samples were thawed and reconstituted in various combinations. Sample concentrations for conditions without Eσ$^{70}$ used 2 μM WT *PkorA* DNA and 5 μM KorA/B factors. For conditions with Eσ$^{70}$, the

sample contained 5 μM Eσ[70], 5.5 μM WT *PkorA* DNA and 2.5-fold molar excess (12.5 μM) of KorA/B factors. For KorB samples with CTP, 0.5 mM CTP was used to ensure saturation of KorB.

For nMS analysis, samples were buffer exchanged into either 150 mM, 300 mM or 500 mM ammonium acetate pH 7.5, 0.01% Tween-20 using Zeba microspin desalting columns with a 7 kDa or 40 kDa molecular weight cut-off (Thermo Scientific). A 2–3 μl aliquot of the sample was then loaded into a gold-coated quartz capillary tip that was prepared in-house and then electrosprayed into an Exactive Plus with extended mass range instrument (Thermo Fisher Scientific) with a static direct infusion nanospray source[82]. The typical MS parameters for all samples included the following: spray voltage, 1.20–1.22 kV; capillary temperature, 125–150 °C; in-source dissociation, 0–10 V; S-lens RF level, 200; resolving power, 8,750 at *m/z* of 200; automatic gain control target, $1 \times 10^6$; maximum injection time, 200 ms; number of microscans, 5; total number of scans, at least 100. Additional specific nMS parameters were injection flatapole, 8 V; interflatapole, 4 V; bent flatapole, 4 V; high-energy collision dissociation, 150–200 V; ultrahigh vacuum pressure, $5.2$–$5.8 \times 10^{-10}$ mbar. Mass calibration in positive extended mass range mode was performed using cesium iodide. For data processing, the collected nMS spectra were visualized using Thermo Xcalibur Qual Browser (v. 4.2.47). Spectral deconvolution was performed using UniDec v. 4.2.0[83,84] with the following general parameters: for background subtraction (if applied), subtract curve 10; smooth charge state distribution, enabled; peak shape function, Gaussian; degree of Softmax distribution (beta parameter), 10–20.

The expected masses for the component proteins of the Eσ[70] include α, 36,511.7 Da; β, 150,632.2 Da; β′, 158,008.1 Da (includes one $Mg^{2+}$ and two $Zn^{2+}$ ions); ω (lost N-terminal methionine), 10,105.4 Da; and σ[70], 70,263.3 Da. The expected masses for the Kor proteins and the DNA scaffold used are KorA monomer, 12,825.6 Da; KorB (lost N-terminal methionine), 40,399.6 Da; and *PkorA* dsDNA, 61,667.2 Da. The observed mass deviations (calculated as the percentage difference between the measured and expected masses relative to the expected mass) ranged from 0.001% to 0.3% with a typical mass deviation of 0.12%.

### Determination of plasmid copy number by deep sequencing

Cell cultures for deep sequencing were grown in the same conditions as for promoter–*xylE* reporter assays. Cells from 5 ml of these cultures were resuspended in 300 μl of Puregene cell lysis solution (Qiagen) before incubation at 50 °C for 10 min. About 2 μl of 20 mg ml⁻¹ RNaseA was added, and the samples were mixed by inverting 25 times before incubation at 37 °C for 60 min. The samples were cooled to room temperature, and 100 μl of Puregene protein precipitation solution (Qiagen) was added. Samples were briefly vortexed and incubated on ice for 10 min, and protein precipitate was removed via centrifugation at 17,000 *g* for 10 min. The supernatant was transferred to a fresh microcentrifuge tube containing 600 μl of isopropanol and mixed by inverting 50 times. Precipitated DNA was pelleted via centrifugation, and the pellet was washed with 600 μl of 70% ethanol before final centrifugation at 17,000 *g* for 1 min. The supernatant was carefully removed, and the DNA pellet was resuspended in 100 μl of distilled water.

Illumina sequencing libraries were prepared using the tagmentation-based and PCR-based Illumina DNA Prep kit and custom IDT 10 bp unique dual indices with a target insert size of 280 bp. No additional DNA fragmentation or size selection steps were performed. The DNA samples were sequenced using 200 Mb Illumina whole-genome sequencing on an Illumina NovaSeq X Plus system (SeqCenter). Sequencing data were mapped back to a reference genome (constructed by concatenating the sequence of the *E. coli* DH5α genome and the sequences of the promoter–*xylE* reporter and those of *korA* and/or *korB* expression plasmids) using Bowtie 2 on default settings as part of the Galaxy open source platform[85,86]. Count read coverage per base was calculated using the Perbase command line package (https://github.com/sstadick/perbase) using the following command:

perbase only-depth *.bam > output.csv. Using Python (v. 3.11.8), the reads were normalized, and the average read coverage of the chromosome and plasmids was used to calculate the plasmid copy number per chromosome for each sample. Data were subsequently analysed and plotted using GraphPad Prism 9.

### Statistics and reproducibility

Experiments were performed in triplicates unless stated otherwise. No statistical method was used to pre-determine the sample size. No data were excluded from the analysis. Details of statistical tests and the *P* values are reported in the legends of relevant figures.

### Reporting summary

Further information on research design is available in the Nature Portfolio Reporting Summary linked to this article.

### Data availability

The crystallographic structures of KorBΔN30ΔCTD and KorBΔN30ΔCTD–KorA-*OA* have been deposited in the PDB with accession codes: 8QA8 and 8QA9, respectively. The crystallographic structure of KorA–DNA complex (PDB code: 2W7N) was also used in this study. Deep sequencing data generated in this study have been deposited in the GEO database under the accession code GSE274567. All uncropped images and data presented in figures are available via Mendeley Data at https://doi.org/10.17632/8cw3ygfssy.1 (ref. [86]). Source data are provided with this paper.

### Code availability

No custom code was used or developed for the analysis of data presented in this paper.

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

## Acknowledgements

We thank M. Tišma and C. Dekker for preliminary experiments, S. Gruber and members of our laboratories for helpful comments and M. Dillingham (University of Bristol) for providing purified *B. subtilis* ParB. This work is supported by the Royal Society University Fellowship Renewal URF\R\201020 and the Lister Institute fellowship (T.B.K.L.); by the Wellcome Trust Investigator grant 221776/Z/2/Z (T.B.K.L. and T.C.M.); by the Biotechnology and Biological Sciences Research Council-funded Institute Strategic Program Harnessing Biosynthesis for Sustainable Food and Health (HBio) (BB/X01097X/1); by grants from the National Institutes of Health R35 GM118130 (S.A.D.), R01 GM14450 (E.A.C.), P41 GM109824 and P41 GM103314 (B.T.C.); by grants PID2020-112998GB-I00 (F.M.-H.) funded by MICIU/AEI/10.13039/501100011033, PID2023-146255NB-I00 (F.M.-H.) funded by MICIU/AEI/10.13039/501100011033 and European Regional Development Fund (FEDER, EU), and by grant FJC2020-044824-I (F.B.-P.) funded by MICIU/AEI/10.13039/501100011033 and European Union Next Generation Recovery, Transformation and Resilience Plan (EU/PRTR). We also thank Diamond Light Source for access to beamlines I04-1 and I03 under proposal MX25108 (D.M.L.).

## Author contributions

Conceptualization: T.C.M., C.M.T., S.A.D., E.A.C., F.M.-H. and T.B.K.L. Data curation: T.C.M., F.B.-P., J.C., C.A.-R., P.D.B.O., D.M.L., E.A.C., F.M.-H. and T.B.K.L. Formal analysis: T.C.M., F.B.-P., J.C., S.B., P.D.B.O., G.G., J.E.A.M., D.M.L., S.A.D., E.A.C., F.M.-H. and T.B.K.L. Funding acquisition: B.T.C., D.M.L., S.A.D., E.A.C., F.M.-H. and T.B.K.L. Investigation: T.C.M., F.B.-P., J.C., C.A.-R., S.B., P.D.B.O., G.G., D.M.L. and

T.B.K.L. Methodology: T.C.M., F.B.-P., J.C., P.D.B.O., B.T.C., E.A.C., F.M.-H. and T.B.K.L. Original draft preparation: T.C.M., F.B.-P., J.C., C.A.-R. and T.B.K.L. Review and editing: T.C.M., F.B.-P., J.C., C.M.T., G.G., J.E.A.M., D.M.L., E.A.C., F.M.-H. and T.B.K.L.

## Competing interests

The authors declare no competing interests.

## Additional information

**Extended data** is available for this paper at https://doi.org/10.1038/s41564-024-01915-3.

**Correspondence and requests for materials** should be addressed to Thomas C. McLean, Seth A. Darst, Elizabeth A. Campbell, Fernando Moreno-Herrero or Tung B. K. Le.

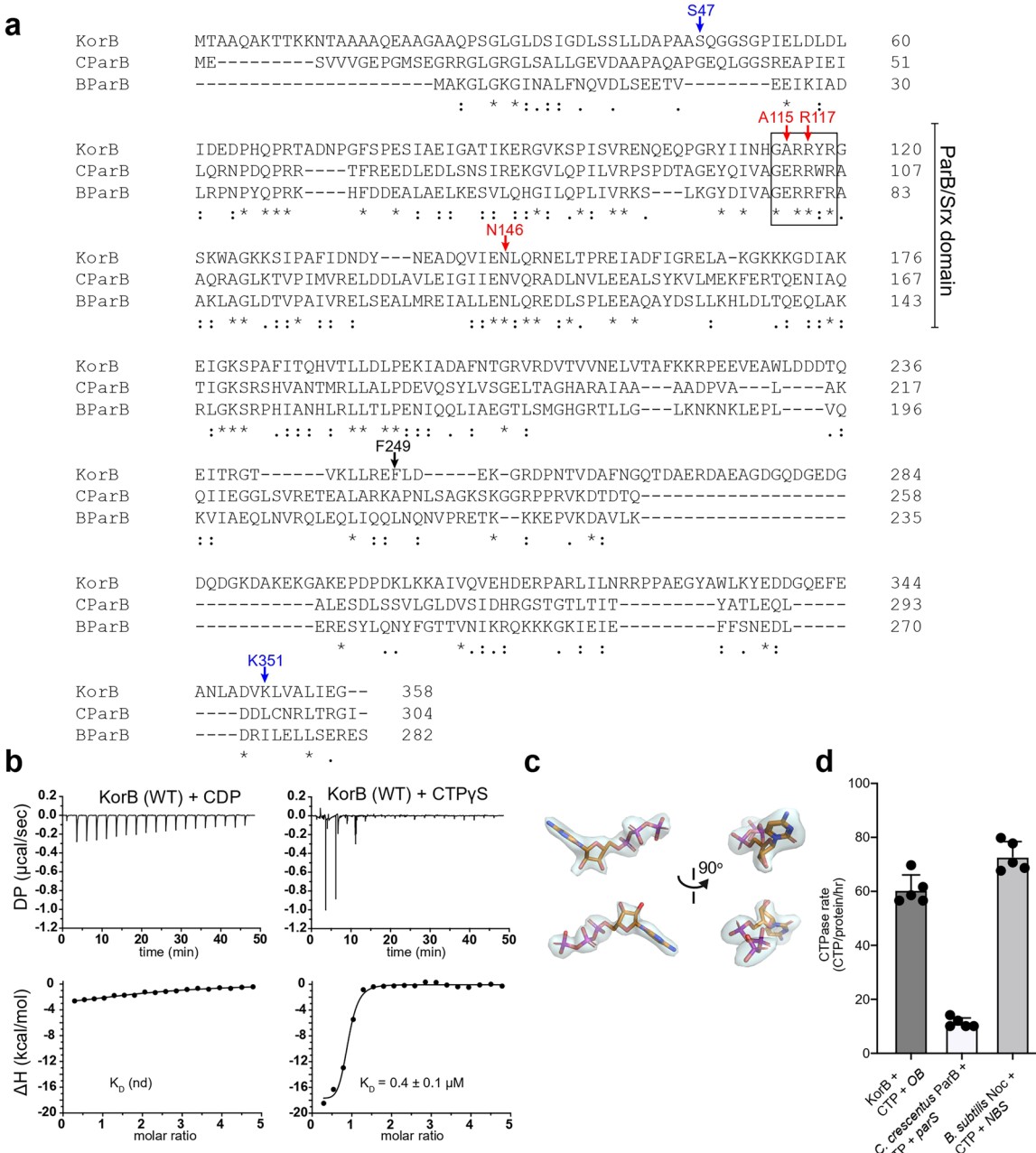

**a**

```
                     S47
KorB    MTAAQAKTTKKNTAAAAQEAAGAAQPSGLGLDSIGDLSSLLDAPAASQGGSGPIELDLDL    60
CParB   ME--------SVVVGEPGMSEGRRGLGRGLSALLGEVDAAPAQAPGEQLGGSREAPIEI    51
BParB   --------------------MAKGLGKGINALFNQVDLSEETV-------EEIKIAD      30
                       :   * *:.:: .   .     .
                                                        A115 R117
                                                     *        :
KorB    IDEDPHQPRTADNPGFSPESIAEIGATIKERGVKSPISVRENQEQPGRYIINH|GARRYR|G   120
CParB   LQRNPDQPRR----TFREEDLEDLSNSIREKGVLQPILVRPSPDTAGEYQIVA|GERRWR|A   107
BParB   LRPNPYQPRK----HFDDEALAELKESVLQHGILQPLIVRKS---LKGYDIVA|GERRFR|A    83
          :   :* ***      *   * : ::    ::*: .*: **.   * *  * **:*
                              N146
KorB    SKWAGKKSIPAFIDNDY---NEADQVIENLQRNELTPREIADFIGRELA-KGKKKGDIAK   176
CParB   AQRAGLKTVPIMVRELDDLAVLEIGIIENVQRADLNVLEEALSYKVLMEKFERTQENIAQ   167
BParB   AKLAGLDTVPAIVRELSEALMREIALLENLQREDLSPLEEAQAYDSLLKHLDLTQEQLAK   143
          ::  **  .::*  :: :          ::**:** :*.   *   *      .:  ::*:
KorB    EIGKSPAFITQHVTLLDLPEKIADAFNTGRVRDVTVVNELVTAFKKRPEEVEAWLDDDTQ   236
CParB   TIGKSRSHVANTMRLLALPDEVQSYLVSGELTAGHARAIAA---AADPVA---L----AK   217
BParB   RLGKSRPHIANHLRLLTLPENIQQLIAEGTLSMGHGRTLLG---LKNKNKLEPL----VQ   196
         :*** .:::  : ** **::: .:  *  :                        ..:
                          F249
KorB    EITRGT------VKLLREFLD-----EK-GRDPNTVDAFNGQTDAERDAEAGDGQDGEDG   284
CParB   QIIEGGLSVRETEALARKAPNLSAGKSKGGRPPRVKDTDTQ------------------   258
BParB   KVIAEQLNVRQLEQLIQQLNQNVPRETK-KKEPVKDAVLK------------------   235
         ::          * :: :       *   :   . *:
KorB    DQDGKDAKEKGAKEPDPDKLKKAIVQVEHDERPARLILNRRPPAEGYAWLKYEDDGQEFE   344
CParB   -----------ALESDLSSVLGLDVSIDHRGSTGTLTIT---------YATLEQL-----   293
BParB   -----------ERESYLQNYFGTTVNIKRQKKKGKIEIE---------FFSNEDL-----   270
                    *   ..     *.:.:     . : :        :  . *:
          K351
KorB    ANLADVKLVALIEG--   358
CParB   ----DDLCNRLTRGI-   304
BParB   ----DRILELLSERES   282
              *        *   .
```

ParB/Srx domain

**b**

KorB (WT) + CDP

KorB (WT) + CTPγS

$K_D$ (nd)

$K_D$ = 0.4 ± 0.1 μM

**c**

90°

**d**

KorB +
CTP + OB

*C. crescentus* ParB +
CTP + *parS*

*B. subtilis* Noc +
CTP + NBS

**Extended Data Fig. 1 | KorB is a CTPase enzyme. a**, A sequence alignment between KorB, *Caulobacter crescentus* ParB (CParB), and *Bacillus subtilis* ParB (BParB). Residues R117, N146, and A115 (red) of KorB are indicated on its amino acid sequence. The position of the P-motif 2 (GARRYR for KorB and GERRxR for canonical ParB), and the N-terminal ParB/Srx-like domain are also indicated on the sequence alignment. Residues S47 or K351C (blue) were substituted by cysteine to generate KorB (S47C) and KorB (K351C) variants which were subsequently used in BMOE crosslinking assays (Fig. 2a and Extended Data Fig. 2a). Residue F249 of KorB that contacts Y84 of KorA is shown with a black arrow (see also Fig. 3e and Extended Data Fig. 6c). **b**, Analysis of the interaction of KorB with CDP and CTPγS by ITC. KorB binding to CDP was qualitatively much weaker than to CTP, but the data precluded the estimation of a binding affinity through

curve fitting. Each experiment was duplicated. Regression curves were fitted, and binding affinities ($K_D$) were shown. **c**, An omit difference map for CTPγS was calculated after removing the ligands from the final structure and re-refining to convergence at 2.3 Å resolution. Shown are orthogonal views of the two ligands from the chain A-B dimer only, together with their associated omit density displayed as a semi-transparent cyan surface contoured at 2.0 σ. It was not possible to unambiguously assign the positions of the ligand sulfur atoms, thus they were modeled as CTP molecules. **d**, CTP hydrolysis rates of KorB, *C. crescentus* ParB, and *B. subtilis* Noc were measured by continuous detection of released inorganic phosphates (see Methods). CTPase rates were measured at 1 mM concentrations of CTP and in the presence of 0.5 μM of cognate DNA duplexes. Data are represented as mean values ± SD from five replicates.

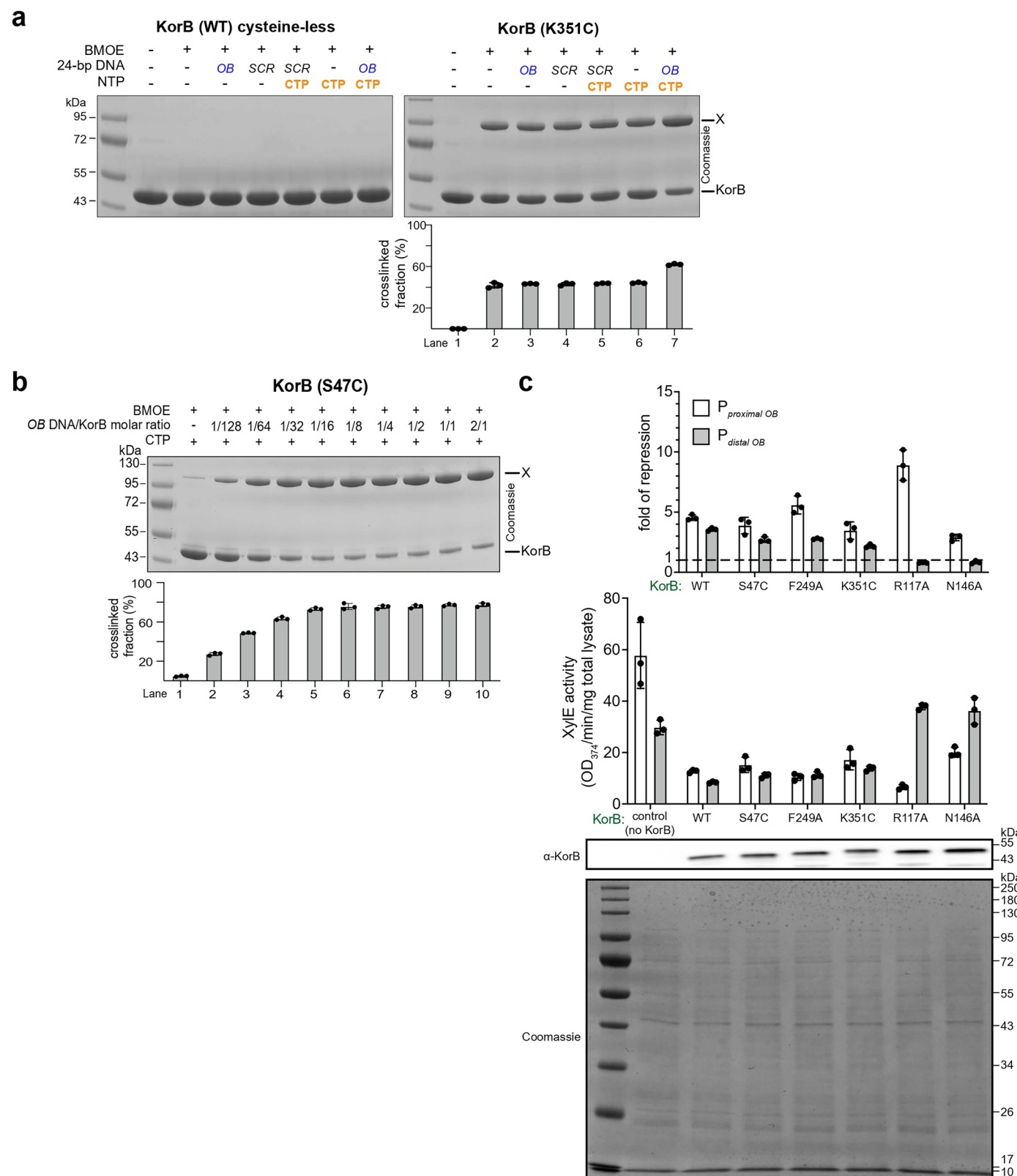

**Extended Data Fig. 2 | See next page for caption.**

**Extended Data Fig. 2 | CTP and *OB* DNA promote the engagement of the N-terminal domain of KorB *in vitro* and are essential for KorB to repress transcription from a distance. a**, (left panel) KorB is cysteine-less and did not crosslink in the presence of BMOE. (right panel) The CTD of KorB is likely a constant dimerization domain, as judged by the crosslinking of KorB (K351C) variant. K351C is predicted to be crosslinkable based on symmetry-related interactions observed in the previously published crystal structure of KorB CTD (PDB: 1IGQ). SDS-PAGE analysis of BMOE crosslinking products of 8 μM of KorB (K351C) dimer (and variants) ± 0.5 μM 24 bp *OB*/scrambled *OB* DNA ± 1 mM CTP. X indicates a crosslinked form of KorB. Quantification of the crosslinked fraction is shown below each representative image. Data are represented as mean values ± SD from three replicates. **b**, Sub-stoichiometric concentrations of *OB* are sufficient to promote crosslinking of KorB (S47C). SDS-PAGE analysis of BMOE crosslinking products of 8 μM of KorB (S47C) dimer + 1 mM CTP + increasing concentration of 24 bp *OB* DNA (from 1/128 to 2/1 *OB*-to-KorB molar ratio). A ratio of ~16-fold less *OB* DNA to KorB was sufficient to achieve maximal crosslinking. Data are represented as mean values ± SD from three replicates. **c**, Substitutions S47C, K351C, F249A, R117A, and N146A on KorB and their impact on KorB's ability to repress *OB*-proximal or distal promoters, as judged by promoter-*xylE* reporter assays. (top panel) Values shown are fold of repression, a ratio of XylE activities from cells co-harboring a reporter plasmid and KorB-expressing plasmid to that of cells co-harboring a reporter plasmid and an empty plasmid (KorB-minus control). Data are represented as mean values ± SD from three replicates. (bottom panel) Absolute values of XylE activities from the same assay. An α-KorB immunoblot and loading controls (Coomassie-stained SDS-PAGE) from lysates of cells used in the same experiments are also shown below.

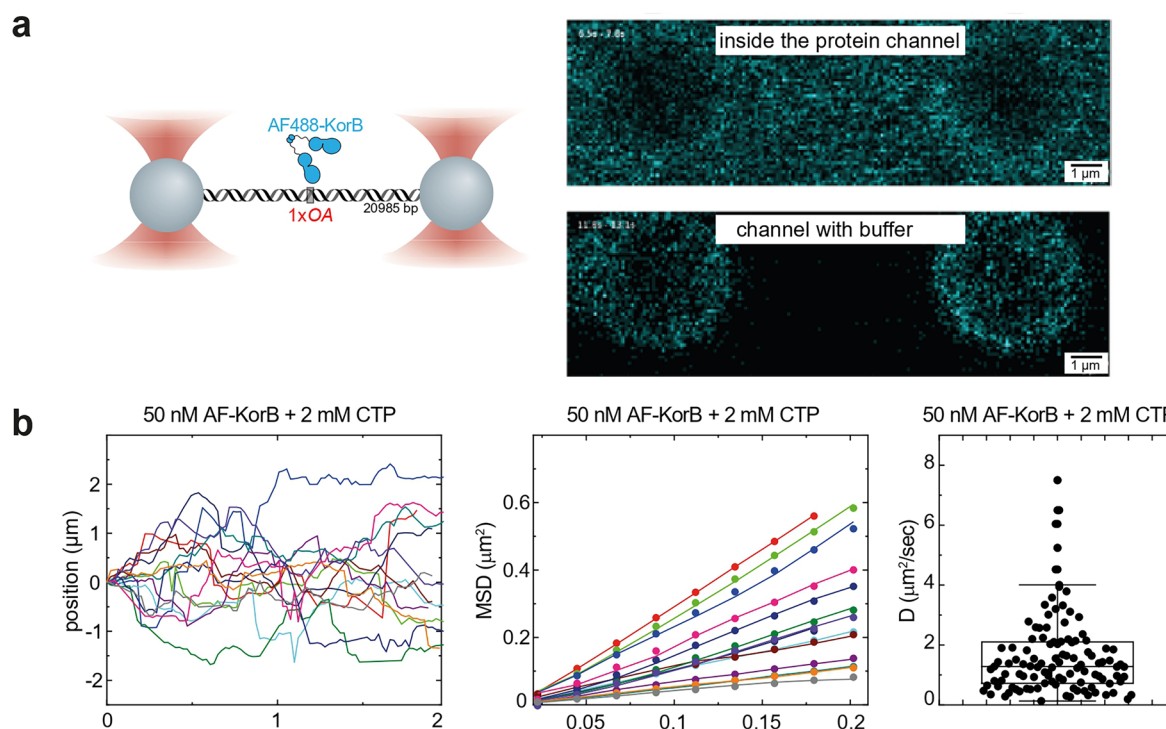

**Extended Data Fig. 3 | CTP enables KorB diffusion on *OB*-containing DNA.**
**a**, AF488-KorB did not bind DNA lacking *OB* site. (left panel) Representative cartoon showing a DNA containing an *OA* site (but no *OB* site) trapped between the two beads in the optical tweezers. (upper right panel) A scan showing DNA trapped between two beads, inside a protein channel containing 50 nM AF488-KorB and 2 mM CTP. (lower right panel) the same DNA molecule, after a minute of incubation with AF488-KorB + CTP in the protein channel, was subsequently transferred to a channel with buffer. We did not observe any binding event, only a high background from the fluorescently labeled protein. **b**, Determination of the diffusion constant of KorB. (left panel) Representative KorB trajectories measured on the DNA (n = 111). (middle panel) Mean squared displacement (MSD) of KorB trajectories for different time intervals ($\Delta t$). (right panel) The diffusion constant of KorB ($1.61 \pm 0.12\ \mu m^2/s$, mean ± SEM) was calculated as half of the slope of the linear fit of MSD versus $\Delta t$. Box plots indicate the median, the 25th and 75th percentiles of the distribution, and whiskers extending to data within 1.5× interquartile range. Outliers are displayed as points beyond the whiskers.

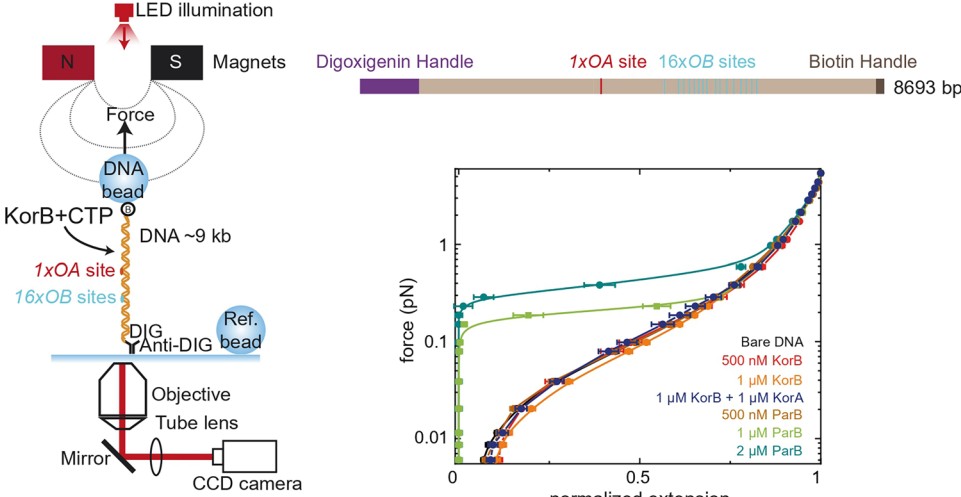

**Extended Data Fig. 4 | KorB did not condense a DNA containing 1x*OA* and 16x*OB* sites in the presence of CTP or KorA.** (left panel) Cartoon of the basic magnetic tweezers (MT) components and the layout of the experiment, and a schematic representation of a DNA containing 1x*OA* and a 16x*OB* cluster. The positions of *OA* and *OB* sites are represented to scale. (right panel) Average force-extension curves of bare DNA molecules (n = 56) and in the presence of different dimer concentrations of KorB + 2 mM CTP (500 nM, n = 11, and 1 μM, n = 13) or *Bacillus subtilis Bs*ParB + 2 mM CTP (500 nM, n = 11, 1 μM, n = 21 and 2 μM, n = 17), and in the presence of 1 μM KorB + 1 μM KorA + 2 mM CTP (n = 10). Data are represented as mean values ± SEM from three replicates.

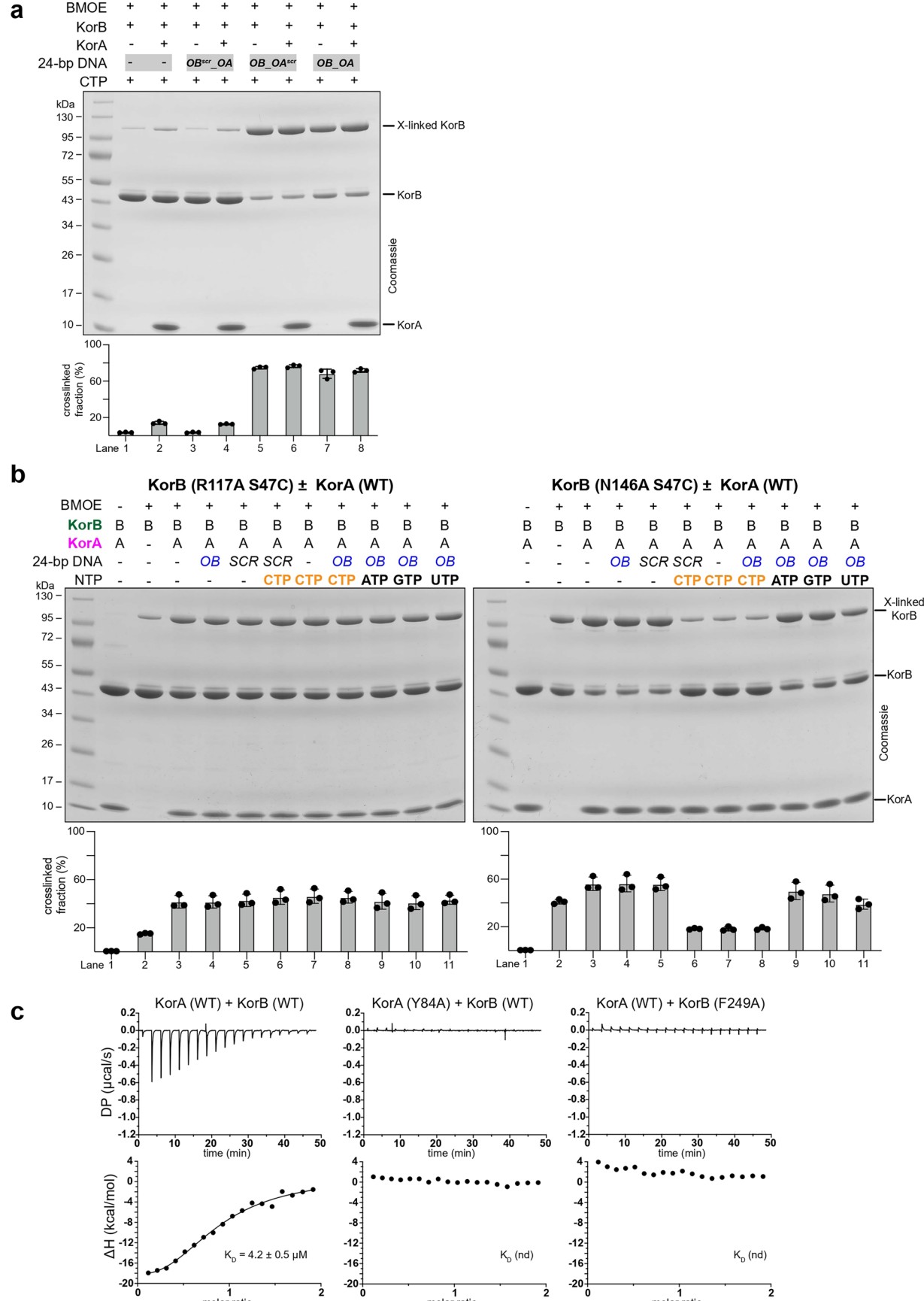

**Extended Data Fig. 5 | See next page for caption.**

**Extended Data Fig. 5 | KorA interacts with KorB to promote the N-engagement of KorB. a**, SDS-PAGE analysis of BMOE crosslinking products of 8 µM of KorB (S47C) dimer ± 8 µM dimer concentration of KorA + 1 mM CTP ± 1 µM 24 bp DNA containing both *OB* and *OA* sites (*OB_OA*) or both scrambled *OB* site and *OA* (*OB^{SCR}_OA*) or both *OB* and scrambled *OA* site (*OB_OA^{SCR}*). X indicates a crosslinked form of KorB (S47C). Quantification of the crosslinked fraction is shown below each representative image. Data are represented as mean values ± SD from three replicates. **b**, KorA can promote clamp-defective KorB (R117A) and KorB (N146A) variants to N-engage. SDS-PAGE analysis of BMOE crosslinking products of

8 µM of KorB (S47C R117A) dimer and KorB (S47C N146A) dimer ± 8 µM of KorA dimer ± 0.5 µM 24 bp *OB*/scrambled (SCR) *OB* DNA ± 1 mM CTP. X indicates a crosslinked form of KorB (S47C R117A) or KorB (S47C N146A). Quantification of the crosslinked fraction is shown below each representative image. Data are represented as mean values ± SD from three replicates. **c**, Substitutions Y84A on KorA or F249A on KorB eliminated KorA-KorB interaction. Analysis of the interaction of KorB (WT or variant) with KorA (WT or variant) by ITC. Each experiment was duplicated. Regression curves were fitted, and binding affinities ($K_D$) were shown.

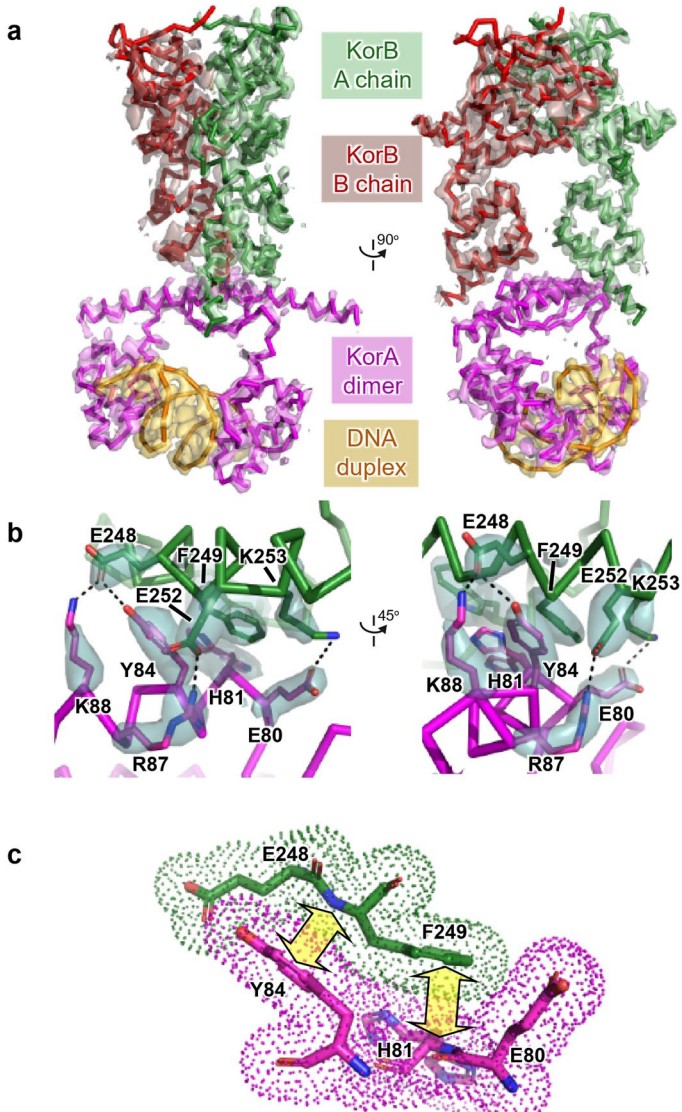

**Extended Data Fig. 6 | Structure of the KorA-KorB-DNA ternary complex.**
The crystal structure of the KorA-KorB-DNA complex was determined at 2.7 Å
resolution. **a**, A series of omit difference maps were calculated by separately
removing parts of the final structure and re-refining to convergence at 2.7 Å
resolution. Maps for KorB chain A (green), KorB chain B (red), the KorA dimer
(magenta) and the DNA duplex (orange) are displayed as semi-transparent
surfaces on a color-coded backbone trace of the structure, contoured at 2.0 σ
and shown as orthogonal views. **b**, Close-up of a KorB-KorA interface with only
the side chains of key residues displayed. Also shown is omit difference density
(semi-transparent cyan surface, contoured at 2.0 σ) calculated for the model
after the removal of these side chains and re-refining. **c**, Further detail on the
KorB-KorA interface with color-coded van der Waals dots illustrating intimate
contact. In addition to the hydrogen bonds highlighted in panel b, the aromatic
ring of Y84 in KorA makes pi-pi interactions with the E248-F249 peptide bond
of KorB, which is reciprocated by the aromatic ring of F249 in KorB making pi-pi
interactions with the E80-H81 peptide bond of KorB. These interactions are
indicated by the pale yellow double-headed arrows.

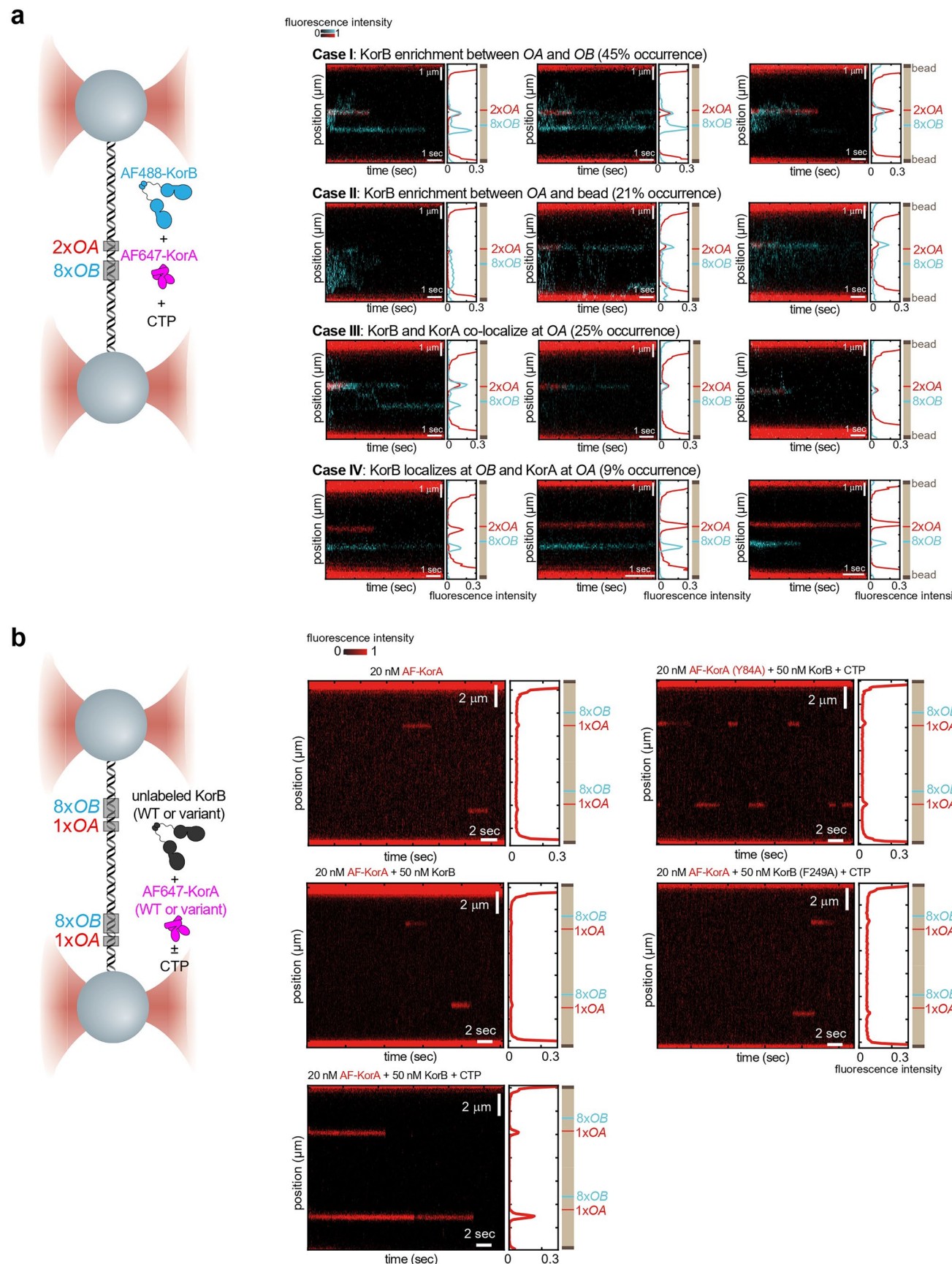

**Extended Data Fig. 7 | See next page for caption.**

**Extended Data Fig. 7 | KorA can block the diffusion of KorB on DNA, and KorB increases the residence time of KorA at its operator *OA*. a**, KorA can block the diffusion of KorB on DNA. (left panel) Schematic of the C-trap optical tweezers experiments where a DNA containing a cluster of 8x*OB* sites and 2x*OA* sites was tethered between two beads and scanned with a confocal microscope using 488 nm and 635 nm illumination. (right panel) More representative kymographs showing the distribution of AF647-KorA and AF488-KorB in the presence of CTP along a DNA for the four cases described. The frequency of occurrence for each case is indicated (the total number of recorded events n = 182). Kymographs were taken in a buffer-only channel to reduce fluorescence background, following a 60 s incubation in the protein channel. **b**, KorB increases the residence time of KorA at *OA*. (left panel) Schematic of the C-trap optical tweezers experiments where a DNA containing two clusters of 8x*OB* sites and 1x*OA* site were tethered between two beads and scanned with a confocal microscope using 635 nm illumination. (right panel) Representative kymograph showing the binding of AF647-labeled KorA either alone or in the presence of unlabeled KorB, the binding of AF647-KorA in the presence of unlabeled KorB and 2 mM CTP, the binding of AF647-KorA (Y84A) in the presence of unlabeled KorB and 2 mM CTP, and the binding of AF647-KorA in the presence of unlabeled KorB (F249A) and 2 mM CTP. Concentrations of proteins are shown above each representative image. KorA residence time experiments were performed in the protein channel using unlabeled KorB and a lower concentration of AF-KorA to minimize fluorescence background.

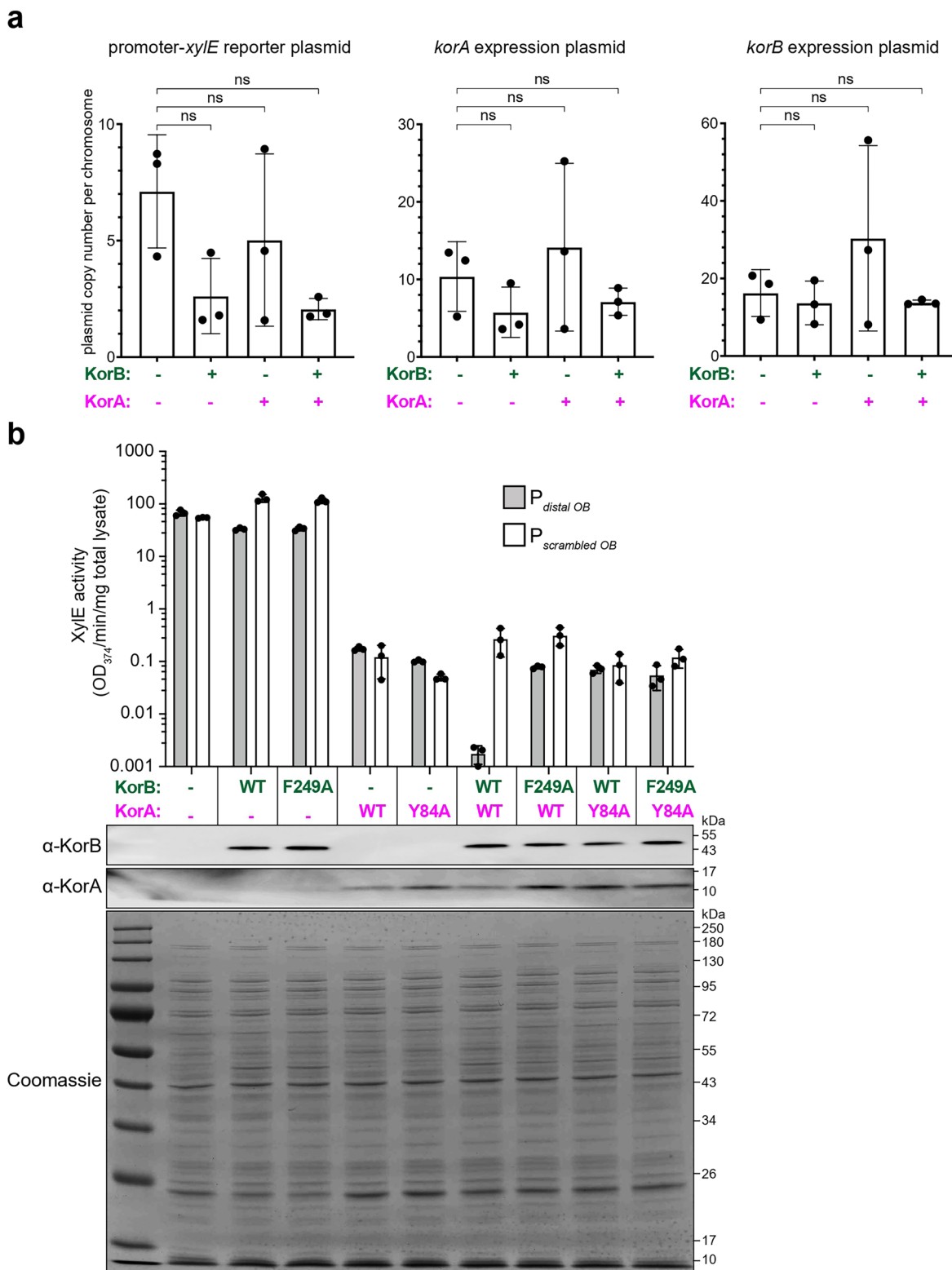

**Extended Data Fig. 8 | KorA and KorB co-operatively heighten long-range transcriptional repression.** Promoter-*xylE* reporter assays were used to measure promoter activities in the presence or absence of KorAB (WT or variants) (see Fig. 4d for the schematic diagrams of promoter-*xylE* reporter constructs). **a**, The copy number of plasmids used in promoter-*xylE* reporter assays. Illumina whole-genome deep sequencing was employed to determine the copy number (relative to the chromosome) for each of the three plasmids used in the promoter-*xylE* reporter assays. Data are represented as mean values ± SD from three replicates. There was not sufficient evidence of a difference in plasmid copy number under any tested conditions (one-way ANOVA statistical test with the null hypothesis that there is no difference in plasmid copy number across all four conditions, three replicates, $P = 0.10$ (left panel, not significant, ns), 0.39 (middle panel, ns), and 0.37 (right panel, ns)). **b**, Absolute values of XylE activities from the same promoter-*xylE* reporter assay in Fig. 4d. Immunoblots (α-KorB and α-KorA) and loading controls (Coomassie-stained SDS-PAGE) from lysates of cells used in the same experiments are also shown below. Data are represented as mean values ± SD from three replicates.

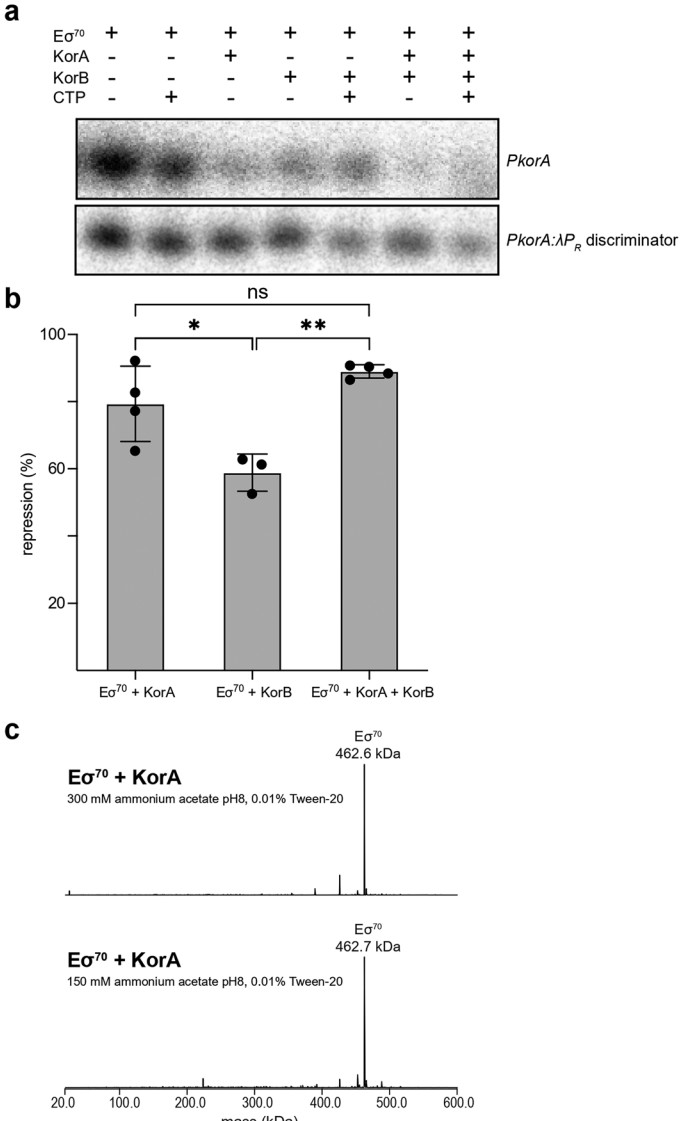

**Extended Data Fig. 9 | KorA and KorB are transcriptional repressors that exclude *E. coli* RNAP from promoters. a**, Representative urea-PAGE gel closeup on the abortive RNA product (5'-ApUpG*-3'; *radiolabelled on guanosine alpha-phosphate) transcribed in in vitro abortive initiation assays on WT *PkorA* and the *PkorA*:λ$P_R$ discriminator mutant linear DNA scaffolds with different mixes of Eσ70 and/or fivefold excess KorA, fivefold excess KorB and/or saturating CTP. **b**, WT *PkorA in vitro* transcription repression of *E. coli* RNAP:σ70 holoenzyme (Eco Eσ70) in the presence of fivefold excess KorA and/or KorB. Experiments were repeated at least three times. All values are normalized to holoenzyme only

control as mean values ± SEM. Eσ70 + KorB condition has n = 3 while the rest are n = 4, and this remains valid as repression quantities are normalized to a Eσ70-only control and background-corrected for each gel run, and SEMs are considered in statistical analysis. *P* value was calculated by an unpaired Welch's t-test; * p ≤ 0.05, ** p ≤ 0.01. The *P* values are 0.1827 (KorA vs KorA + KorB), 0.0281 (KorA vs KorB), and 0.0067 (KorB vs KorA + KorB). **c**, Deconvolved native mass spectra of Eσ70 (5 µM) with 2.5-fold excess KorA dimer electrospray ionized in buffer of 300 mM or 150 mM ammonium acetate pH 8.0 and 0.01% Tween-20. No peak of Eσ70:KorA was observed.

# Reporting Summary

## Statistics

For all statistical analyses, confirm that the following items are present in the figure legend, table legend, main text, or Methods section.

| n/a | Confirmed | |
|---|---|---|
| ☐ | ☒ | The exact sample size (*n*) for each experimental group/condition, given as a discrete number and unit of measurement |
| ☐ | ☒ | A statement on whether measurements were taken from distinct samples or whether the same sample was measured repeatedly |
| ☐ | ☒ | The statistical test(s) used AND whether they are one- or two-sided<br>*Only common tests should be described solely by name; describe more complex techniques in the Methods section.* |
| ☒ | ☐ | A description of all covariates tested |
| ☒ | ☐ | A description of any assumptions or corrections, such as tests of normality and adjustment for multiple comparisons |
| ☐ | ☒ | A full description of the statistical parameters including central tendency (e.g. means) or other basic estimates (e.g. regression coefficient) AND variation (e.g. standard deviation) or associated estimates of uncertainty (e.g. confidence intervals) |
| ☐ | ☒ | For null hypothesis testing, the test statistic (e.g. *F*, *t*, *r*) with confidence intervals, effect sizes, degrees of freedom and *P* value noted<br>*Give P values as exact values whenever suitable.* |
| ☒ | ☐ | For Bayesian analysis, information on the choice of priors and Markov chain Monte Carlo settings |
| ☒ | ☐ | For hierarchical and complex designs, identification of the appropriate level for tests and full reporting of outcomes |
| ☒ | ☐ | Estimates of effect sizes (e.g. Cohen's *d*, Pearson's *r*), indicating how they were calculated |

*Our web collection on statistics for biologists contains articles on many of the points above.*

## Software and code

Policy information about availability of computer code

| Data collection | Amersham Imager 600 (GE Healthcare), LUMICKS C-trap instrument (Lumicks) |
|---|---|
| Data analysis | AIMLESS version 0.7.7 https://www.ccp4.ac.uk/; BUCCANEER version 1.6.11 https://www.ccp4.ac.uk/;CCP4i2 version 7.1.018 https://www.ccp4.ac.uk/;COOT version 0.9.6 https://www2.mrc-lmb.cam.ac.uk/personal/pemsley/coot/;DIALS version 3.dev.659-g0b5a5c991 https://dials.github.io/; Excel 2016 Microsoft RRID: SCR_016137;MolProbity version 4.4 http://molprobity.biochem.duke.edu/;PHASER version 2.8.3 https://phenix-online.org/;PyMOL Tversion 2 he PyMOL Molecular Graphics System https://pymol.org/2/;R version 3.2.4 R Foundation for Statistical Computing https://www.r-project.org/;REFMAC5 version 5.8.0403 https://www.ccp4.ac.uk/;XIA2 version 3.9.dev0 https://xia2.github.io/index.html;ColabFold v1.5.5 https://github.com/sokrypton/ColabFold;GraphPadPrism v10 GraphPad RRID: SCR_002798;ImageJ NIH RRID: SCR_003070;Thermo Xcalibur Qual Browser v. 4.2.47 ThermoFisher Cat# OPTON-30965; OriginPro v10.2 https://store.originlab.com/store/. |

For manuscripts utilizing custom algorithms or software that are central to the research but not yet described in published literature, software must be made available to editors and reviewers. We strongly encourage code deposition in a community repository (e.g. GitHub). See the Nature Portfolio guidelines for submitting code & software for further information.

## Data

Policy information about availability of data

All manuscripts must include a data availability statement. This statement should provide the following information, where applicable:
- Accession codes, unique identifiers, or web links for publicly available datasets
- A description of any restrictions on data availability
- For clinical datasets or third party data, please ensure that the statement adheres to our policy

The crystallographic structures of KorBdeltaN30deltaCTD and KorBdeltaN30deltaCTD-KorA-OA have been deposited in the PDB with accession codes: 8QA8 and 8QA9, respectively. The crystallographic structure of KorA-DNA complex (PDB code: 2W7N) was also used in this study. Deep sequencing data generated in this study have been deposited in the GEO database under the accession code GSE274567. All uncropped images and data presented in figures are available in Source Data.

## Research involving human participants, their data, or biological material

Policy information about studies with human participants or human data. See also policy information about sex, gender (identity/presentation), and sexual orientation and race, ethnicity and racism.

| | |
|---|---|
| Reporting on sex and gender | NA |
| Reporting on race, ethnicity, or other socially relevant groupings | NA |
| Population characteristics | NA |
| Recruitment | NA |
| Ethics oversight | NA |

Note that full information on the approval of the study protocol must also be provided in the manuscript.

# Field-specific reporting

Please select the one below that is the best fit for your research. If you are not sure, read the appropriate sections before making your selection.

☒ Life sciences    ☐ Behavioural & social sciences    ☐ Ecological, evolutionary & environmental sciences

For a reference copy of the document with all sections, see nature.com/documents/nr-reporting-summary-flat.pdf

# Life sciences study design

All studies must disclose on these points even when the disclosure is negative.

| | |
|---|---|
| Sample size | No sample size calculation was performed. For ITC experiment the number of replicate is 2, for all other experiment numbers of replicates are three or more (see relevant figure legend). The number if replicates is standard/common practice in the field. |
| Data exclusions | No data were excluded from the analyses |
| Replication | We routinely analyzed multiple independent strains, multiple different protein aliquots or purification to verify the phenotypes observed. All experiments were performed at least twice or more to ensure reproducibility, and similar results were obtained throughout. |
| Randomization | Strains for different experiments were selected randomly for inoculation from plate. All strains were grown under similar conditions, hence are equivalent at the start of the experiment. Observed differences are due to the difference in genotype of analyzed strains in this study. |
| Blinding | This was not necessary for in-bulk/population-averaged microbiology study. All strains were grown under similar conditions, hence are equivalent at the start of the experiment. Observed differences are due to the difference in genotype of analyzed strains in this study. |

# Reporting for specific materials, systems and methods

We require information from authors about some types of materials, experimental systems and methods used in many studies. Here, indicate whether each material, system or method listed is relevant to your study. If you are not sure if a list item applies to your research, read the appropriate section before selecting a response.

## Materials & experimental systems

| n/a | Involved in the study |
|-----|----------------------|
| ☐ | ☒ Antibodies |
| ☒ | ☐ Eukaryotic cell lines |
| ☒ | ☐ Palaeontology and archaeology |
| ☒ | ☐ Animals and other organisms |
| ☒ | ☐ Clinical data |
| ☒ | ☐ Dual use research of concern |
| ☒ | ☐ Plants |

## Methods

| n/a | Involved in the study |
|-----|----------------------|
| ☒ | ☐ ChIP-seq |
| ☒ | ☐ Flow cytometry |
| ☒ | ☐ MRI-based neuroimaging |

## Antibodies

| | |
|---|---|
| Antibodies used | polyclonal antibodies against KorB and KorA (custom synthesis, University of Birmingham, UK) (and further purified in the lab of Prof. Chris Thomas, University of Birmingham, UK) and Goat Anti-Rabbit IgG H&L (HRP) secondary antibody (ABCAM, UK cat#AB6721) |
| Validation | The specificity of primary antibodies against KorB and KorA used in this study was verified against lysates from deletion mutant strains or non-tagged strains of E. coli. Both antibodies have been used and reported in previous publications (see publications PMID: 7473715 and PMID: 10564465) from Prof. Chris Thomas' group at the University of Birmingham, UK. Secondary antibody: Goat Anti-Rabbit IgG H&L (HRP) secondary antibody is available commercially (ABCAM, UK cat#AB6721) |

## Plants

| | |
|---|---|
| Seed stocks | Not relevant |
| Novel plant genotypes | Not relevant |
| Authentication | Not relevant |

