## [Peer review file · Nature Microbiology]

KorB switching from DNA-sliding clamp to repressor mediates long-range gene silencing in a multi-drug resistance plasmid

Corresponding Author: Professor Tung Le

Version 0:

Reviewer comments:

Reviewer #1

(Remarks to the Author)

KorB is essential for segregation of the broad-host-range multidrug resistance plasmid RK2 plasmid and for regulating gene expression for all basic RK2 functions, including replication and conjugative transfer.

In the present manuscript, the authors present that the ParB-homologue is a CTPase, which is stimulated by OB-DNA. OB-DNA together with CTP promote dimerization of KorB. The presented data indicate that the capacity of KorB to bind CTP, close the clamp, and traverse along DNA is essential for long-range gene silencing. The manuscript identifies a direct interaction of KorA with KorB. The impressive crystal structure of the KorB/KorA/operator of KorA-DNA duplex together with the provided functional assays provides a sophisticated explanation of how KorA obstructs KorB diffusion on DNA and enhances retention time at the OA operator.

In summary, this is an impressive and technically sound piece of work! I strongly recommend publication.

To enhance the editorial clarity and refine the manuscript, I suggest the following revision: I believe the discussion can be significantly condensed to eliminate redundancy. Additionally, I question the necessity of including the 'Final Perspectives' section, as it may not contribute substantially to the overall coherence of this splendid manuscript.

Reviewer #2

(Remarks to the Author)

I have reviewed the MS. by McLean et al. working on gene regulation in a bacterial system by a transcription factor bound to a DNA site distal to the promoter. Although transcription factor regulating gene transcription from a distal DNA site is not common, in most of those cases the action of the transcription factors relies on DNA looping. In the current investigation the authors have conclusively shown that the transcription factor KorB, encoded by the RK2 plasmid in *E. coli*, represses a promoter by binding to a distal DNA site, not by DNA looping, but by sliding down the DNA to the promoter region. KorB encounters another factor, KorA, bound to a DNA site proximal to the promoter. This encounter presumably hinders RNA polymerase binding to the promoter. KorB has a CTPase activity which is stimulated by DNA binding. The authors used the state of the art biophysical techniques to demonstrate different steps of the process of KorB action in transcription repression: (i) isothermal titration calorimetry to determine CTP affinity of KorB and various CTP non-binding mutants; (ii) crystal structure determination together with cross-linking experiments identified CTP-dependent clamping of KorB to DNA; (iii) biolayer interferometry CTP-dependent (not hydrolysis) binding of KorB to its specific DNA site; (iv) crosslinking experiments demonstrating that KorA, by interacting with KorB, help engagement (proper dimerization?) of KorB-NTD; (v) crystal structure determination of KorB-NTD/KorA/KorA-DNA binding site ternary complex showing that the KorA/DNA complex facilitates proper dimerization of KorB-NTD that is more compatible for KorB sliding on DNA; and (vi) single molecule experiments with fluorescent-labeled KorA and KorB demonstrating that KorA- OA-DNA complex blocks sliding of KorB-CTP on DNA thus making KorB a transcriptional co-repressor.

The experimental results reported are very clear and convincing. The idea of a protein sliding on DNA has been demonstrated clearly. The paper is also written clearly. I recommend publication of the manuscript in Nature. However, I have only a few comments for the authors regarding (a) presentation of data in figure 3, and some discussions and interpretations of the data presented in figure 6. I suggest that the authors think about my suggestions, as presented below, and to make changes in discussing their data if they agree with my way of thinking.

Figures 3 and S3. The amount of reporter gene expressions shown in figure 3b may be presented in absolute amounts, and not in relative amounts to reflect that the various conditions used do not change the promoter strength in a major way.

Figure 6. I always thought that transcription repression in majority of cases in prokaryotic systems are not steric exclusion of RNA polymerase binding by repressors but trapping of bound RNA polymerase at some stages of initiation. (The authors need not buy the idea.) None of the results presented here discard the concept because the way the experiments are designed. For

example, a short DNA fragment may enhance dissociation of RNA polymerase from a repressor-DNA-RNA polymerase ternary complex. Indeed, figure 6B shows a small amount of RNA polymerase-DNA-KorA complex. Maybe, it would be further stabilized by the presence of KorB. I would have liked to see a DNase protection assays done with supercoiled DNA in the presence various combinations of the protein components.

Reviewer #3

(Remarks to the Author)

The manuscript (ms) "Molecular switching of a DNA-sliding clamp to a repressor mediates long-range gene silencing in a multi-drug resistance plasmid" deals with long-distance gene regulation – a phenomenon known in eukaryotes but relatively rare in bacteria. It focuses on the DNA binding protein KorB that is necessary for plasmid RK2 inheritance but also for regulation of gene expression. It was known previously that KorB binds a silencer sequence called OB (operator of KorB) and that a small DNA-binding protein KorA is also necessary for gene regulation. The aim of the work was to figure out what was the mechanism of KorB's long-range gene expression regulation with respect to its interplay with KorA.

The ms contains a large amount of data from various experimental approaches, including partial structures of KorB, and KorB in complex with KorA and OA DNA. The main finding is that KorB slides along DNA in a CTP-dependent manner, and upon encountering KorA bound to its own operator (OA) near a promoter, changes conformation and forms a stable KorA-KorB complex that inhibits transcription from the promoter.

Comments:

Overall, the ms contains fundamentally novel insights and will be of interest to a wide range of readers. However, there are several discrepancies between the presented results and the suggested model that should be clarified.

1. It is argued in the ms (line 137) that CTP and OB are required for efficient KorB NTD dimerization. However, (line 161) it is also proposed that binding of KorB at OB is CTP-independent. Clearly, Fig. 2B shows binding of KorB to this sequence. How do you know, however, that CTP does not improve it further? Can you comment on it?
2. It is suggested that for the KorB movement on DNA, the presence of CTP but not its hydrolysis was necessary (line 182). Therefore, the results of the optical tweezers experiments (Fig. 2B) should be the same with CTP and its non-hydrolyzable analog (CTPyS). However, KorB seems more localized/less sliding in the left panel where CTP cannot be hydrolysed. Please, explain. Moreover, what is the model with respect to the force/energy that allows the KorB sliding on DNA when it does not hydrolyse CTP? Additionally, CTP hydrolysis seems to play a role in dimerization of KorB (compare Fig. 2A, lanes 7 of left- and right-hand panels) Please, explain.
3. It is claimed that the KorB-KorA interaction takes place at the OA sequence (Fig. 7). However, the presented crosslinking experiments (Fig 4B) were done with OB and showed that KorA increased KorB NTD engagement outside of OA. In Fig. 7, you show this happening only at OA. Modify/explain better in Discussion.
4. Abstract – modify the text so that it is clear that KorA already binds DNA. KorB, then, after sliding along DNA and encountering KorA, changes conformation and the resulting KorA-KorB complex inhibits transcription.
5. Except in the Materials and Methods, it is not mentioned which bacterium was the model system used in the study. It would be useful if the reader did not have to look for this information, it should be mentioned in the Introduction.
6. Page 2, first paragraph, last sentence. The long-range effect of topology on gene expression should be mentioned, e. g. PMID 9281422, PMID 34951454, PMID 31539491.
7. Although the results show that N146 is necessary for CTP hydrolysis, no rationale is provided why/how this mutant was designed – N146 is not even shown in Fig. 1D.
8. Figure 2A lane 2 and 3, panel one (wt) vs panel three (nonhydrolyzing mutant): The X-linking efficiencies between the two protein variants differ. Could you explain?
9. In Figures 3, 5, it is not clear what the „basal“ values were from which the x-fold of repression was calculated. Could you provide primary data to see the background values? Calculating fold changes from near background values can be misleading.
10. Figure 5 A is somewhat confusing. Why using different relative arrangements of OA, OB sites in the two panels? Can you find better representative images so that the relative positions of the OA, OB sites are identical in experiments where you are also comparing different conditions (presence of KorB, KorA, CTP)? Furthermore, in the same Figure, the right-hand side panel: why did not the upper OB site also bind KorB?

Reviewer #4

(Remarks to the Author)

McLean et al. present new insights into the function of KorB, a ParB-like protein involved in the partitioning and gene regulation of the plasmid RK2. KorB can repress genes at a distance and cooperates with its binding partner KorA, however, the underlying mechanisms remained unclear.

The authors show that KorB is a CTPase with characteristics similar to what has been reported for other ParB-like proteins. They present a crystal structure of a KorB fragment comprising the N-terminal CTP-binding domain (NTD) and the middle DNA-binding domain (DBD), bound to the CTP analogue CTPgS. The structure shows engaged NTDs, consistent with previous observations for other ParB-like CTPases. Cross-linking assays suggest that CTP promotes NTD engagement in the presence of the KorB-binding DNA sequence OB, confirming its ParB-like behavior.

The authors then employ a C-trap single-molecule setup to characterize DNA association of KorB. Recruitment of KorB to OB is independent of CTP. In the presence of CTP or CTPgS, however, KorB appears to release OB sites and diffuse along DNA. Bio-layer interferometry experiments using double-tethered or single-tethered DNA suggest that both nucleotides promote the conversion of KorB to a sliding clamp. In vivo gene expression experiments using KorB mutants defective in CTP binding or hydrolysis suggest that CTPase activity is required for long-distance repression.

Next, the authors show that KorA promotes engagement of the NTD even in the absence of nucleotide. They solve a crystal structure of the KorB NTD DBD fragment in complex with KorA and the KorA-binding DNA sequence OA. The NTDs are engaged in the absence of nucleotide and KorA contacts the bottom of the DBD. The authors then include KorA in their C-trap assay and find that the addition KorB and CTP increases its residence time on OA. Furthermore, KorA seems to act as a diffusion roadblock for KorB. The authors confirm that KorA and KorB have a strong synergistic effect on gene repression, and that this synergy is disrupted by mutations in the KorAB interface.

Finally, the authors propose that KorAB repression works best on promoters with weak sigma70 binding, based on in vitro transcription initiation experiments and displacement studies using native mass-spectrometry. They conclude with a model that aims to explain long-distance repression by KorAB. KorB associates with the RK2 plasmid at OB sites, converts to a sliding clamp upon CTP binding, and diffuses to a target promoter bound by KorA. KorAB then efficiently occludes the sigma factor binding site and represses the promoter.

I found the data and model intriguing, and the manuscript engaging and well-presented. The paper highlights how, in principle, a single binding site for a repressor can be used to regulate many genes by loading a sliding clamp. The paper also points out how this type of regulation can be modulated by additional factors, which may be used to fine-tune its effect on different target genes. I feel the story is ready for publication.

Here are a few comments/questions which I hope are helpful:

1. Any ideas on how, if at all, genes under KorAB repression can be induced? Does this solely depend on KorB levels?
2. Can we exclude loop formation? Is this notion based on the roadblock experiments from the literature and the magnetic tweezer experiments presented here? I was wondering if there is any form of condensation in magnetic tweezer experiments when KorA and OA are included in addition to KorB/OB.
3. The background signal in the C-trap experiments is a bit unclear. What does this look like in the absence of OB? Showing this could also confirm that CTP-bound diffusing KorB was recruited via OB.
4. Fig. 5D: KorB accumulates in a somewhat narrow zone close to OA. Why is that? I would expect it to diffuse back and forth over the full length of the DNA, and perhaps form a gradient towards OA. Alternatively, if it forms a stable complex with KorA, I would expect the proteins to co-localize. I think this should be discussed, as I feel it potentially contradicts the proposed model.
5. Introduction: "Long-range gene regulation by the DNA-binding protein KorB is essential for stable vertical inheritance and horizontal transmission of the broad-host-range multidrug resistance plasmid RK2, as well as for the fitness of its host bacterium". I am not sure if the cited publications show that gene regulation is an essential function of KorB. Are there any experiments that uncouple gene regulation from the more traditional ParB-like partitioning activities?
6. Introduction: The conserved role of ParABS systems in plasmid partitioning should be mentioned in addition to their role in chromosome segregation.
7. Introduction: The KorA binding site OA should be mentioned.
8. Fig. 1D: Possible to show N146, since targeted by mutagenesis?
9. Cross-linking: The mutants used seem to affect basal cross-linking levels. Why is that?
10. Cross-linking: OB DNA was used in sub-stoichiometric amounts (1 μ M DNA and 8 μ M KorB), yet KorB cross-linking is close to 90 %. Does this mean that something catalytic is going on? What happens at different concentrations of DNA?
11. Cross-linking: WT KorB shows decreased NTD cross-linking in the presence of CTP, but increased cross-linking when CTPgS is added. The protein responds differently to the two nucleotides, possibly due to hydrolysis of CTP. N146A does not hydrolyze CTP, but responds to CTP similarly to WT (decreased cross-linking) and does not respond much to CTPgS or CTP/CTPgS + OB. I expected that N146A would cross-link more when either nucleotide is used, which it doesn't. Does this suggest that the differences between CTP/CTPgS conformations are, at least in part, due to the different nucleotides used rather than nucleotide hydrolysis? CTP and CTPgS may not be fully equivalent, and the ITC data presented here perhaps supports this view (much higher affinity for CTPgS). I believe similar observations have been made for other proteins and nucleotide/analogue pairs. The CTPgS bound structure might not fully reflect the CTP bound state, but perhaps relate to a transition state, and one might want to reconsider modelling the nucleotide as CTPgS to avoid confusion (even if the position of the sulphur atom is unknown).
12. Fig. 2D: Labels are a bit unclear. What are the numbers in the plot and gradient referring to?
13. C-trap experiment: "To amplify the signal of KorB DNA binding, eight OB sites were present either in one or two clusters on the DNA (Figure 2B)". Why one OR two clusters? This needs some explanation. The figure should make this clearer in the cartoon, which currently suggests that two clusters were present.
14. C-trap experiment: What is meant by "to amplify the signal" (see above)? Does clustering of OB lead to stronger fluorescence, an increased probability of observations, or both?

15. Repression experiments: Do you know whether the reporter plasmid has the same copy number +/- KorA/B? Differences in replication or stability could influence expression levels.
16. Fig. 3D: Do all bands come from the same Western blot membrane? Is it possible to show a loading control, e.g. a stained membrane?
17. The KD of KorAB formation is rather low (4 μ M). Is binding stronger in the presence of OA? Has KorB cross-linking been tested in the presence of OA?
18. "Notably, for both KorB R117A and N146A variants, which are defective in NTD-engagement, the presence of KorA further increases the efficiency of crosslinking in all tested conditions (Figure S4A vs. Figure 2A)" – This would be much more convincing if Figure S4A contained a control without KorA, instead of referring to Figure 2A.
19. Fig. 5C: The residence time of KorA Y84A is compared to KorA WT. I would find this experiment even more convincing if the residence time of KorA WT could be tested in the presence of KorB and KorB F249A, as this would use the same KorA protein as the analyte.
20. In vitro transcription experiment: "Transcription initiation assays confirmed that KorA and KorB (with and without CTP) individually function as repressors (Figure S7A and B), while their combined activity is further augmented, consistent with our in vivo and single-molecule studies." I am not convinced that this system properly reconstitutes the synergy between KorA and KorB. The reported synergistic effect size, if any, is tiny and nowhere near the massive increase of KorAB-mediated repression in vivo.
21. Discussion: "we propose that KorB-CTP loads at the OB site and then switches to a closed-clamp conformation through NTD engagement" and "Once KorB-CTP slides away, another KorB-CTP molecule can bind a now-free OB and repeat the process". Don't the structure and single-molecule experiments suggest that nucleotide-free KorB associates with OB, and slides away upon CTP binding, rather than KorB-CTP recognizing OB?
22. Discussion: "and the clamp-trapping ability of CTCF is crucial to switch the function of cohesin between sister chromosome cohesion and gene expression regulation and genome folding". I'm not sure this statement is precise. The switch to genome folding i.e. loop extrusion does not require CTCF.
23. Is it stated somewhere if protein concentration refers to monomers or dimers?
24. I recommend including a strain table. A table of proteins containing the exact constructs used would be an additional plus (including tags, linkers, extra residues etc.). Adding supplemental DNA sequences used in binding experiments and gene expression studies would also be great.

Decision Letter:

1st May 2024

Dear Professor Le,

Thank you for your patience while your manuscript "Molecular switching of a DNA-sliding clamp to a repressor mediates long-range gene silencing in a multi-drug resistance plasmid" was under peer-review at Nature Microbiology. It has now been seen by 4 referees, whose expertise and comments you will find at the end of this email. Although they find your work of some potential interest, they have raised a number of concerns that will need to be addressed before we can consider publication of the work in Nature Microbiology.

In particular, you will see that the referees raise several concerns that some conclusions and mechanisms proposed in the model for KorAB long-range regulation are not fully supported by the data. Several questions were raised over the role of nucleotide binding and hydrolysis for KorB binding and sliding activity. In addition, there were concerns over how the data showing KorB and KorA localisation fit with a model suggesting interaction and "locking" or repression activity at OA sites. One of the referees also questioned whether the data supported a conclusion for RNAP occlusion and requested additional analyses. There were also several requests for additional controls to confirm effects of OB upon KorB binding activity, and for side-by-side controls in other experimental set-ups. We feel that these are critical points which would need to be addressed for us to further consider a revised manuscript, alongside the remaining issues outlined in the referees' reports, which are clear and should be straightforward to address.

Should further experimental data allow you to address these criticisms, we would be happy to look at a revised manuscript.

Please include a data availability statement as a separate section after Methods but before references, under the heading "Data Availability". This section should inform readers about the availability of the data used to support the conclusions of your study. This information includes accession codes to public repositories (data banks for protein, DNA or RNA sequences, microarray, proteomics data etc...), references to source data published alongside the paper, unique identifiers such as URLs to data repository entries, or data set DOIs, and any other statement about data availability. At a minimum, you should include the following statement: "The data that support the findings of this study are available from the corresponding author upon request",

mentioning any restrictions on availability. If DOIs are provided, we also strongly encourage including these in the Reference list (authors, title, publisher (repository name), identifier, year). For more guidance on how to write this section please see: <http://www.nature.com/authors/policies/data/data-availability-statements-data-citations.pdf>

* If you have not done so already we suggest that you begin to revise your manuscript so that it conforms to our Article format instructions at <http://www.nature.com/nmicrobiol/info/final-submission>. Refer also to any guidelines provided in this letter.

When submitting the revised version of your manuscript, please pay close attention to our [href="https://www.nature.com/nature-portfolio/editorial-policies/image-integrity">Digital Image Integrity Guidelines. and to the following points below:](https://www.nature.com/nature-portfolio/editorial-policies/image-integrity)

Link Redacted

Note: This url links to your confidential homepage and associated information about manuscripts you may have submitted or be reviewing for us. If you wish to forward this e-mail to co-authors, please delete this link to your homepage first.

Nature Microbiology is committed to improving transparency in authorship. As part of our efforts in this direction, we are now requesting that all authors identified as 'corresponding author' on published papers create and link their Open Researcher and Contributor Identifier (ORCID) with their account on the Manuscript Tracking System (MTS), prior to acceptance. This applies to primary research papers only. ORCID helps the scientific community achieve unambiguous attribution of all scholarly contributions. You can create and link your ORCID from the home page of the MTS by clicking on 'Modify my Springer Nature account'. For more information please visit www.springernature.com/orcid.

If you wish to submit a suitably revised manuscript we would hope to receive it within 6 months. If you cannot send it within this time, please let us know. We will be happy to consider your revision, even if a similar study has been accepted for publication at Nature Microbiology or published elsewhere (up to a maximum of 6 months).

Yours sincerely,

Reviewer Expertise:

Referee #1: structural biology
Referee #2: regulation of gene expression, nucleoid organisation, imaging
Referee #3: regulation of gene expression
referee #4: protein-DNA interactions, single-particle analysis

Reviewer Comments:

Reviewer #1 (Remarks to the Author):

KorB is essential for segregation of the broad-host-range multidrug resistance plasmid RK2 plasmid and for regulating gene expression for all basic RK2 functions, including replication and conjugative transfer.

In the present manuscript, the authors present that the ParB-homologue is a CTPase, which is stimulated by OB-DNA. OB-DNA together with CTP promote dimerization of KorB. The presented data indicate that the capacity of KorB to bind CTP, close the

clamp, and traverse along DNA is essential for long-range gene silencing. The manuscript identifies a direct interaction of KorA with KorB. The impressive crystal structure of the KorB/KorA/operator of KorA-DNA duplex together with the provided functional assays provides a sophisticated explanation of how KorA obstructs KorB diffusion on DNA and enhances retention time at the OA operator.

In summary, this is an impressive and technically sound piece of work! I strongly recommend publication.

To enhance the editorial clarity and refine the manuscript, I suggest the following revision: I believe the discussion can be significantly condensed to eliminate redundancy. Additionally, I question the necessity of including the 'Final Perspectives' section, as it may not contribute substantially to the overall coherence of this splendid manuscript.

Reviewer #2 (Remarks to the Author):

I have reviewed the MS. by McLean et al. working on gene regulation in a bacterial system by a transcription factor bound to a DNA site distal to the promoter. Although transcription factor regulating gene transcription from a distal DNA site is not common, in most of those cases the action of the transcription factors relies on DNA looping. In the current investigation the authors have conclusively shown that the transcription factor KorB, encoded by the RK2 plasmid in *E. coli*, represses a promoter by binding to a distal DNA site, not by DNA looping, but by sliding down the DNA to the promoter region. KorB encounters another factor, KorA, bound to a DNA site proximal to the promoter. This encounter presumably hinders RNA polymerase binding to the promoter. KorB has a CTPase activity which is stimulated by DNA binding. The authors used the state of the art biophysical techniques to demonstrate different steps of the process of KorB action in transcription repression: (i) isothermal titration calorimetry to determine CTP affinity of KorB and various CTP non-binding mutants; (ii) crystal structure determination together with cross-linking experiments identified CTP-dependent clamping of KorB to DNA; (iii) biolayer interferometry CTP-dependent (not hydrolysis) binding of KorB to its specific DNA site; (iv) crosslinking experiments demonstrating that KorA, by interacting with KorB, help engagement (proper dimerization?) of KorB-NTD; (v) crystal structure determination of KorB-NTD/KorA/KorA-DNA binding site ternary complex showing that the KorA/DNA complex facilitates proper dimerization of KorB-NTD that is more compatible for KorB sliding on DNA; and (vi) single molecule experiments with fluorescent-labeled KorA and KorB demonstrating that KorA-OA-DNA complex blocks sliding of KorB-CTP on DNA thus making KorB a transcriptional co-repressor.

The experimental results reported are very clear and convincing. The idea of a protein sliding on DNA has been demonstrated clearly. The paper is also written clearly. I recommend publication of the manuscript in *Nature*. However, I have only a few comments for the authors regarding (a) presentation of data in figure 3, and some discussions and interpretations of the data presented in figure 6. I suggest that the authors think about my suggestions, as presented below, and to make changes in discussing their data if they agree with my way of thinking.

Figures 3 and S3. The amount of reporter gene expressions shown in figure 3b may be presented in absolute amounts, and not in relative amounts to reflect that the various conditions used do not change the promoter strength in a major way.

Figure 6. I always thought that transcription repression in majority of cases in prokaryotic systems are not steric exclusion of RNA polymerase binding by repressors but trapping of bound RNA polymerase at some stages of initiation. (The authors need not buy the idea.) None of the results presented here discard the concept because the way the experiments are designed. For example, a short DNA fragment may enhance dissociation of RNA polymerase from a repressor-DNA-RNA polymerase ternary complex. Indeed, figure 6B shows a small amount of RNA polymerase-DNA-KorA complex. Maybe, it would be further stabilized by the presence of KorB. I would have liked to see a DNase protection assays done with supercoiled DNA in the presence various combinations of the protein components.

Reviewer #3 (Remarks to the Author):

The manuscript (ms) "Molecular switching of a DNA-sliding clamp to a repressor mediates long-range gene silencing in a multi-drug resistance plasmid" deals with long-distance gene regulation – a phenomenon known in eukaryotes but relatively rare in bacteria. It focuses on the DNA binding protein KorB that is necessary for plasmid RK2 inheritance but also for regulation of gene expression. It was known previously that KorB binds a silencer sequence called OB (operator of KorB) and that a small DNA-binding protein KorA is also necessary for gene regulation. The aim of the work was to figure out what was the mechanism of KorB's long-range gene expression regulation with respect to its interplay with KorA.

The ms contains a large amount of data from various experimental approaches, including partial structures of KorB, and KorB in complex with KorA and OA DNA. The main finding is that KorB slides along DNA in a CTP-dependent manner, and upon encountering KorA bound to its own operator (OA) near a promoter, changes conformation and forms a stable KorA-KorB complex that inhibits transcription from the promoter.

Comments:

Overall, the ms contains fundamentally novel insights and will be of interest to a wide range of readers. However, there are several discrepancies between the presented results and the suggested model that should be clarified.

1. It is argued in the ms (line 137) that CTP and OB are required for efficient KorB NTD dimerization. However, (line 161) it is also proposed that binding of KorB at OB is CTP-independent. Clearly, Fig. 2B shows binding of KorB to this sequence. How do you know, however, that CTP does not improve it further? Can you comment on it?

2. It is suggested that for the KorB movement on DNA, the presence of CTP but not its hydrolysis was necessary (line 182). Therefore, the results of the optical tweezers experiments (Fig. 2B) should be the same with CTP and its non-hydrolyzable analog (CTPyS). However, KorB seems more localized/less sliding in the left panel where CTP cannot be hydrolysed. Please, explain. Moreover, what is the model with respect to the force/energy that allows the KorB sliding on DNA when it does not hydrolyse CTP? Additionally, CTP hydrolysis seems to play a role in dimerization of KorB (compare Fig. 2A, lanes 7 of left- and right-hand panels) Please, explain.
3. It is claimed that the KorB-KorA interaction takes place at the OA sequence (Fig. 7). However, the presented crosslinking experiments (Fig 4B) were done with OB and showed that KorA increased KorB NTD engagement outside of OA. In Fig. 7, you show this happening only at OA. Modify/explain better in Discussion.
4. Abstract – modify the text so that it is clear that KorA already binds DNA. KorB, then, after sliding along DNA and encountering KorA, changes conformation and the resulting KorA-KorB complex inhibits transcription.
5. Except in the Materials and Methods, it is not mentioned which bacterium was the model system used in the study. It would be useful if the reader did not have to look for this information, it should be mentioned in the Introduction.
6. Page 2, first paragraph, last sentence. The long-range effect of topology on gene expression should be mentioned, e. g. PMID 9281422, PMID 34951454, PMID 31539491.
7. Although the results show that N146 is necessary for CTP hydrolysis, no rationale is provided why/how this mutant was designed – N146 is not even shown in Fig. 1D.
8. Figure 2A lane 2 and 3, panel one (wt) vs panel three (nonhydrolyzing mutant): The X-linking efficiencies between the two protein variants differ. Could you explain?
9. In Figures 3, 5, it is not clear what the „basal“ values were from which the x-fold of repression was calculated. Could you provide primary data to see the background values? Calculating fold changes from near background values can be misleading.
10. Figure 5 A is somewhat confusing. Why using different relative arrangements of OA, OB sites in the two panels? Can you find better representative images so that the relative positions of the OA, OB sites are identical in experiments where you are also comparing different conditions (presence of KorB, KorA, CTP)? Furthermore, in the same Figure, the right-hand side panel: why did not the upper OB site also bind KorB?

Reviewer #4 (Remarks to the Author):

McLean et al. present new insights into the function of KorB, a ParB-like protein involved in the partitioning and gene regulation of the plasmid RK2. KorB can repress genes at a distance and cooperates with its binding partner KorA, however, the underlying mechanisms remained unclear.

The authors show that KorB is a CTPase with characteristics similar to what has been reported for other ParB-like proteins. They present a crystal structure of a KorB fragment comprising the N-terminal CTP-binding domain (NTD) and the middle DNA-binding domain (DBD), bound to the CTP analogue CTPgS. The structure shows engaged NTDs, consistent with previous observations for other ParB-like CTPases. Cross-linking assays suggest that CTP promotes NTD engagement in the presence of the KorB-binding DNA sequence OB, confirming its ParB-like behavior.

The authors then employ a C-trap single-molecule setup to characterize DNA association of KorB. Recruitment of KorB to OB is independent of CTP. In the presence of CTP or CTPgS, however, KorB appears to release OB sites and diffuse along DNA. Bio-layer interferometry experiments using double-tethered or single-tethered DNA suggest that both nucleotides promote the conversion of KorB to a sliding clamp. In vivo gene expression experiments using KorB mutants defective in CTP binding or hydrolysis suggest that CTPase activity is required for long-distance repression.

Next, the authors show that KorA promotes engagement of the NTD even in the absence of nucleotide. They solve a crystal structure of the KorB NTD DBD fragment in complex with KorA and the KorA-binding DNA sequence OA. The NTDs are engaged in the absence of nucleotide and KorA contacts the bottom of the DBD. The authors then include KorA in their C-trap assay and find that the addition of KorB and CTP increases its residence time on OA. Furthermore, KorA seems to act as a diffusion roadblock for KorB. The authors confirm that KorA and KorB have a strong synergistic effect on gene repression, and that this synergy is disrupted by mutations in the KorAB interface.

Finally, the authors propose that KorAB repression works best on promoters with weak sigma70 binding, based on in vitro transcription initiation experiments and displacement studies using native mass-spectrometry. They conclude with a model that aims to explain long-distance repression by KorAB. KorB associates with the RK2 plasmid at OB sites, converts to a sliding clamp upon CTP binding, and diffuses to a target promoter bound by KorA. KorAB then efficiently occludes the sigma factor binding site and represses the promoter.

I found the data and model intriguing, and the manuscript engaging and well-presented. The paper highlights how, in principle, a

single binding site for a repressor can be used to regulate many genes by loading a sliding clamp. The paper also points out how this type of regulation can be modulated by additional factors, which may be used to fine-tune its effect on different target genes. I feel the story is ready for publication.

Here are a few comments/questions which I hope are helpful:

1. Any ideas on how, if at all, genes under KorAB repression can be induced? Does this solely depend on KorB levels?
2. Can we exclude loop formation? Is this notion based on the roadblock experiments from the literature and the magnetic tweezer experiments presented here? I was wondering if there is any form of condensation in magnetic tweezer experiments when KorA and OA are included in addition to KorB/OB.
3. The background signal in the C-trap experiments is a bit unclear. What does this look like in the absence of OB? Showing this could also confirm that CTP-bound diffusing KorB was recruited via OB.
4. Fig. 5D: KorB accumulates in a somewhat narrow zone close to OA. Why is that? I would expect it to diffuse back and forth over the full length of the DNA, and perhaps form a gradient towards OA. Alternatively, if it forms a stable complex with KorA, I would expect the proteins to co-localize. I think this should be discussed, as I feel it potentially contradicts the proposed model.
5. Introduction: "Long-range gene regulation by the DNA-binding protein KorB is essential for stable vertical inheritance and horizontal transmission of the broad-host-range multidrug resistance plasmid RK2, as well as for the fitness of its host bacterium". I am not sure if the cited publications show that gene regulation is an essential function of KorB. Are there any experiments that uncouple gene regulation from the more traditional ParB-like partitioning activities?
6. Introduction: The conserved role of ParABS systems in plasmid partitioning should be mentioned in addition to their role in chromosome segregation.
7. Introduction: The KorA binding site OA should be mentioned.
8. Fig. 1D: Possible to show N146, since targeted by mutagenesis?
9. Cross-linking: The mutants used seem to affect basal cross-linking levels. Why is that?
10. Cross-linking: OB DNA was used in sub-stoichiometric amounts (1 μ M DNA and 8 μ M KorB), yet KorB cross-linking is close to 90 %. Does this mean that something catalytic is going on? What happens at different concentrations of DNA?
11. Cross-linking: WT KorB shows decreased NTD cross-linking in the presence of CTP, but increased cross-linking when CTPgS is added. The protein responds differently to the two nucleotides, possibly due to hydrolysis of CTP. N146A does not hydrolyze CTP, but responds to CTP similarly to WT (decreased cross-linking) and does not respond much to CTPgS or CTP/CTPgS + OB. I expected that N146A would cross-link more when either nucleotide is used, which it doesn't. Does this suggest that the differences between CTP/CTPgS conformations are, at least in part, due to the different nucleotides used rather than nucleotide hydrolysis? CTP and CTPgS may not be fully equivalent, and the ITC data presented here perhaps supports this view (much higher affinity for CTPgS). I believe similar observations have been made for other proteins and nucleotide/analogue pairs. The CTPgS bound structure might not fully reflect the CTP bound state, but perhaps relate to a transition state, and one might want to reconsider modelling the nucleotide as CTPgS to avoid confusion (even if the position of the sulphur atom is unknown).
12. Fig. 2D: Labels are a bit unclear. What are the numbers in the plot and gradient referring to?
13. C-trap experiment: "To amplify the signal of KorB DNA binding, eight OB sites were present either in one or two clusters on the DNA (Figure 2B)". Why one OR two clusters? This needs some explanation. The figure should make this clearer in the cartoon, which currently suggests that two clusters were present.
14. C-trap experiment: What is meant by "to amplify the signal" (see above)? Does clustering of OB lead to stronger fluorescence, an increased probability of observations, or both?
15. Repression experiments: Do you know whether the reporter plasmid has the same copy number +/- KorA/B? Differences in replication or stability could influence expression levels.
16. Fig. 3D: Do all bands come from the same Western blot membrane? Is it possible to show a loading control, e.g. a stained membrane?
17. The KD of KorAB formation is rather low (4 μ M). Is binding stronger in the presence of OA? Has KorB cross-linking been tested in the presence of OA?
18. "Notably, for both KorB R117A and N146A variants, which are defective in NTD-engagement, the presence of KorA further increases the efficiency of crosslinking in all tested conditions (Figure S4A vs. Figure 2A)" – This would be much more convincing if Figure S4A contained a control without KorA, instead of referring to Figure 2A.
19. Fig. 5C: The residence time of KorA Y84A is compared to KorA WT. I would find this experiment even more convincing if the residence time of KorA WT could be tested in the presence of KorB and KorB F249A, as this would use the same KorA protein as the analyte.
20. In vitro transcription experiment: "Transcription initiation assays confirmed that KorA and KorB (with and without CTP) individually function as repressors (Figure S7A and B), while their combined activity is further augmented, consistent with our in vivo and single-molecule studies." I am not convinced that this system properly reconstitutes the synergy between KorA and KorB. The reported synergistic effect size, if any, is tiny and nowhere near the massive increase of KorAB-mediated repression in vivo.
21. Discussion: "we propose that KorB-CTP loads at the OB site and then switches to a closed-clamp conformation through NTD engagement" and "Once KorB-CTP slides away, another KorB-CTP molecule can bind a now-free OB and repeat the process". Don't the structure and single-molecule experiments suggest that nucleotide-free KorB associates with OB, and slides away upon CTP binding, rather than KorB-CTP recognizing OB?
22. Discussion: "and the clamp-trapping ability of CTCF is crucial to switch the function of cohesin between sister chromosome cohesion and gene expression regulation and genome folding". I'm not sure this statement is precise. The switch to genome folding i.e. loop extrusion does not require CTCF.
23. Is it stated somewhere if protein concentration refers to monomers or dimers?
24. I recommend including a strain table. A table of proteins containing the exact constructs used would be an additional plus (including tags, linkers, extra residues etc.). Adding supplemental DNA sequences used in binding experiments and gene expression studies would also be great.

Version 1:

Reviewer comments:

Reviewer #2

(Remarks to the Author)

I have reviewed the revised manuscripts. I am happy with the response of the authors in revising the manuscript. It now presents with strong evidence to show that an enhancer bound protein Slides on DNA to exert its action to repress transcription at the cognate promoter. It is a very novel mechanism of enhancer action. I recommend publication of the manuscript.
Sankar Adhya

Reviewer #3

(Remarks to the Author)

The authors have thoroughly answered my questions. Congratulations on an excellent piece of work!

Reviewer #4

(Remarks to the Author)

The authors have addressed all my comments and have included additional experimental evidence and controls to support their reasoning. I feel that the manuscript has significantly improved and should be published. I congratulate all authors on this exciting story.

I have only a few minor comments that I hope are helpful:

I may have missed this, but is there an explanation for why the background in Fig. 4a (AF-KorB + AF-KorA) is so high? If there is a technical reason, this should be mentioned somewhere.

p.9 l.329 "A subsequent KorB protein binding to OB might act as a roadblock, preventing the first KorB protein from passing through the OB site, thereby restricting KorB proteins to either side of OB."

Case II does not require a roadblock at OB, does it? KorB could just pass through OB and localize over the whole segment between OA and bead.

p.9 l.331 "The higher occurrence of case I compared to case II is likely due to the shorter distance between OB and OA compared to one between OB and the proximal bead, reflecting a higher chance of KorB detachment in the latter case."

I'm not completely convinced by this explanation for the accumulation of KorB on the OA proximal side. Shouldn't the off-rate for a given KorB be independent of the segment length? I could imagine that transient binding of KorA plays a role in enriching KorB on the OA proximal side. In a related matter, is it possible that OB alone acts as a roadblock by re-binding the sliding KorB? Another bound KorB might not be needed to block movement.

p.10 l.340 "During the investigation, we noted that the AF647-KorA fluorescence signal exhibited an on-off behavior at the OA site in the presence of KorB but absence of CTP (Extended Data Fig. 8b). However, the AF647-KorA signal (at OA) was more stable over time when KorB and CTP were both included (Extended Data Fig. 8b), suggesting that KorB-CTP might reduce KorA dissociation from OA."

The new experiment in Fig. 4c / Extended Data Fig. 8b is nice. For the sake of completeness, the authors might consider also reporting the residence time of KorA alone, as this would show that KorB CTP has indeed a stabilizing effect rather than the other conditions destabilizing KorA/OA.

p.10 l.340 "KorA and KorB were inducibly produced from separate plasmids, and their production did not alter the copy number of the promoter-xyIE reporter plasmid nor those of the korAB expression plasmids (Extended Data Fig. 9a)."

A few comments here:

- The p-values or significance level for the ANOVA should be mentioned in the figure or figure legend.
- The statistical test merely indicates that there isn't sufficient evidence to reject the null hypothesis, namely that the copy numbers are equal. It does not necessarily mean that there is no difference. Differences might become significant if more replicates were to be observed. I would thus recommend softening the statement above.
- There is a potential threefold reduction of the xyIE plasmid in the presence of KorB, which is somewhat close to the apparent four- to fivefold repression under the KorB-only condition. However, the new data convincingly show that the strong repressive effects of KorA and KorAB cannot be explained by potential copy number variations.

p.11 l.397 "Transcription initiation assays confirmed that KorA and KorB (with and without CTP) individually function as repressors, while their combined activity is further augmented (Extended Data Fig. 10 a-b). Notably, KorA emerges as the stronger repressor ($p \leq 0.05$; unpaired Welch's t-test; Extended Data Fig. 10b)."

Since a t-test for KorA vs. KorB is included, one might also consider showing this for KorA vs. KorAB.

p.12 l.426 "Our findings suggest that KorA and KorB exploit RK2 promoter kinetic instabilities (i.e. weak discriminators) to

competitively occlude RNAPs from thermodynamically strong promoters.”

I'm slightly puzzled by this statement. Could you unpack it a bit to make it more accessible? My intuition would be that slower off-rate kinetics also influence the thermodynamic equilibrium (assuming on-rates don't go down as well).

Decision Letter:

2nd October 2024

Dear Professor Le,

Thank you for your patience while your manuscript "Molecular switching of a DNA-sliding clamp to a repressor mediates long-range gene silencing in a multi-drug resistance plasmid" was under peer-review at Nature Microbiology. It has now been seen by 3 referees, whose expertise and comments you will find at the of this email. You will see from their comments below that while they find your work of interest, there are a few remaining points to address. We are very interested in the possibility of publishing your study in Nature Microbiology, but would like to consider your response to these concerns in the form of a revised manuscript before we make a final decision on publication.

In particular, you will see that Referee #4 still has a few points for which they ask for further clarification or to discuss alternative interpretations. There is also a request to provide statistical support for several statements, and to provide residence time measurements for KorA binding, to add further support for the conclusion that KorB binding stabilises the interaction. We believe that these points should be straightforward to address.

If you have not done so already please begin to revise your manuscript so that it conforms to our Article format instructions at <http://www.nature.com/nmicrobiol/info/final-submission/>

The usual length limit for a Nature Microbiology Article is six display items (figures or tables) and 3,000 words. We have some flexibility, and can allow a revised manuscript at 3,500 words, but please consider this a firm upper limit. There is a trade-off of ~250 words per display item, so if you need more space, you could move a Figure or Table to Supplementary Information.

Some reduction could be achieved by focusing any introductory material and moving it to the start of your opening 'bold' paragraph, whose function is to outline the background to your work, describe in a sentence your new observations, and explain your main conclusions. The discussion should also be limited. Methods should be described in a separate section following the discussion, we do not place a word limit on Methods.

Nature Microbiology titles should give a sense of the main new findings of a manuscript, and should not contain punctuation. Please keep in mind that we strongly discourage active verbs in titles, and that they should ideally fit within 90 characters each (including spaces).

Please include a data availability statement as a separate section after Methods but before references, under the heading "Data Availability". This section should inform readers about the availability of the data used to support the conclusions of your study. This information includes accession codes to public repositories (data banks for protein, DNA or RNA sequences, microarray, proteomics data etc...), references to source data published alongside the paper, unique identifiers such as URLs to data repository entries, or data set DOIs, and any other statement about data availability. At a minimum, you should include the following statement: "The data that support the findings of this study are available from the corresponding author upon request", mentioning any restrictions on availability. If DOIs are provided, we also strongly encourage including these in the Reference list (authors, title, publisher (repository name), identifier, year). For more guidance on how to write this section please see: <http://www.nature.com/authors/policies/data/data-availability-statements-data-citations.pdf>

To improve the accessibility of your paper to readers from other research areas, please pay particular attention to the wording of the paper's opening bold paragraph, which serves both as an introduction and as a brief, non-technical summary in about 150 words. If, however, you require one or two extra sentences to explain your work clearly, please include them even if the paragraph is over-length as a result. The opening paragraph should not contain references. Because scientists from other sub-disciplines will be interested in your results and their implications, it is important to explain essential but specialised terms concisely. We suggest you show your summary paragraph to colleagues in other fields to uncover any problematic concepts.

If your paper is accepted for publication, we will edit your display items electronically so they conform to our house style and will reproduce clearly in print. If necessary, we will re-size figures to fit single or double column width. If your figures contain several parts, the parts should form a neat rectangle when assembled. Choosing the right electronic format at this stage will speed up the processing of your paper and give the best possible results in print. We would like the figures to be supplied as vector files -

EPS, PDF, AI or postscript (PS) file formats (not raster or bitmap files), preferably generated with vector-graphics software (Adobe Illustrator for example). Please try to ensure that all figures are non-flattened and fully editable. All images should be at least 300 dpi resolution (when figures are scaled to approximately the size that they are to be printed at) and in RGB colour format. Please do not submit Jpeg or flattened TIFF files. Please see also 'Guidelines for Electronic Submission of Figures' at the end of this letter for further detail.

Figure legends must provide a brief description of the figure and the symbols used, within 350 words, including definitions of any error bars employed in the figures.

Please include a statement before the acknowledgements naming the author to whom correspondence and requests for materials should be addressed.

Finally, we require authors to include a statement of their individual contributions to the paper -- such as experimental work, project planning, data analysis, etc. -- immediately after the acknowledgements. The statement should be short, and refer to authors by their initials. For details please see the Authorship section of our joint Editorial policies at http://www.nature.com/authors/editorial_policies/authorship.html

* include a point-by-point response to any editorial suggestions and to our referees. Please include your response to the editorial suggestions in your cover letter, and please upload your response to the referees as a separate document.

* ensure it complies with our format requirements for Letters as set out in our guide to authors at www.nature.com/nmicrobiol/info/gta/

* state in a cover note the length of the text, methods and legends; the number of references; number and estimated final size of figures and tables

* resubmit electronically if possible using the link below to access your home page:

Link Redacted

*This url links to your confidential homepage and associated information about manuscripts you may have submitted or be reviewing for us. If you wish to forward this e-mail to co-authors, please delete this link to your homepage first.

Please ensure that all correspondence is marked with your Nature Microbiology reference number in the subject line.

Nature Microbiology is committed to improving transparency in authorship. As part of our efforts in this direction, we are now requesting that all authors identified as 'corresponding author' on published papers create and link their Open Researcher and Contributor Identifier (ORCID) with their account on the Manuscript Tracking System (MTS), prior to acceptance. This applies to primary research papers only. ORCID helps the scientific community achieve unambiguous attribution of all scholarly contributions. You can create and link your ORCID from the home page of the MTS by clicking on 'Modify my Springer Nature account'. For more information please visit www.springernature.com/orcid.

We hope to receive your revised paper within three weeks. If you cannot send it within this time, please let us know.

Yours sincerely,

Reviewer Expertise:

Reviewers Comments:

Reviewer #2 (Remarks to the Author):

I have reviewed the revised manuscripts. I am happy with the response of the authors in revising the manuscript. It now presents with strong evidence to show that an enhancer bound protein Slides on DNA to exert its action to repress transcription at the cognate promoter. It is a very novel mechanism of enhancer action. I recommend publication of the manuscript.
Sankar Adhya

Reviewer #3 (Remarks to the Author):

The authors have thoroughly answered my questions. Congratulations on an excellent piece of work!

Reviewer #4 (Remarks to the Author):

The authors have addressed all my comments and have included additional experimental evidence and controls to support their reasoning. I feel that the manuscript has significantly improved and should be published. I congratulate all authors on this exciting story.

I have only a few minor comments that I hope are helpful:

I may have missed this, but is there an explanation for why the background in Fig. 4a (AF-KorB + AF-KorA) is so high? If there is a technical reason, this should be mentioned somewhere.

p.9 l.329 "A subsequent KorB protein binding to OB might act as a roadblock, preventing the first KorB protein from passing through the OB site, thereby restricting KorB proteins to either side of OB."

Case II does not require a roadblock at OB, does it? KorB could just pass through OB and localize over the whole segment between OA and bead.

p.9 l.331 "The higher occurrence of case I compared to case II is likely due to the shorter distance between OB and OA compared to one between OB and the proximal bead, reflecting a higher chance of KorB detachment in the latter case."

I'm not completely convinced by this explanation for the accumulation of KorB on the OA proximal side. Shouldn't the off-rate for a given KorB be independent of the segment length? I could imagine that transient binding of KorA plays a role in enriching KorB on the OA proximal side. In a related matter, is it possible that OB alone acts as a roadblock by re-binding the sliding KorB? Another bound KorB might not be needed to block movement.

p.10 l.340 "During the investigation, we noted that the AF647-KorA fluorescence signal exhibited an on-off behavior at the OA site in the presence of KorB but absence of CTP (Extended Data Fig. 8b). However, the AF647-KorA signal (at OA) was more stable over time when KorB and CTP were both included (Extended Data Fig. 8b), suggesting that KorB-CTP might reduce KorA dissociation from OA."

The new experiment in Fig. 4c / Extended Data Fig. 8b is nice. For the sake of completeness, the authors might consider also reporting the residence time of KorA alone, as this would show that KorB CTP has indeed a stabilizing effect rather than the other conditions destabilizing KorA/OA.

p.10 l.340 "KorA and KorB were inducibly produced from separate plasmids, and their production did not alter the copy number of the promoter-xyIE reporter plasmid nor those of the korAB expression plasmids (Extended Data Fig. 9a)."

A few comments here:

- The p-values or significance level for the ANOVA should be mentioned in the figure or figure legend.
- The statistical test merely indicates that there isn't sufficient evidence to reject the null hypothesis, namely that the copy numbers are equal. It does not necessarily mean that there is no difference. Differences might become significant if more replicates were to be observed. I would thus recommend softening the statement above.
- There is a potential threefold reduction of the xyIE plasmid in the presence of KorB, which is somewhat close to the apparent four- to fivefold repression under the KorB-only condition. However, the new data convincingly show that the strong repressive effects of KorA and KorAB cannot be explained by potential copy number variations.

p.11 l.397 "Transcription initiation assays confirmed that KorA and KorB (with and without CTP) individually function as repressors, while their combined activity is further augmented (Extended Data Fig. 10 a-b). Notably, KorA emerges as the stronger repressor ($p \leq 0.05$; unpaired Welch's t-test; Extended Data Fig. 10b)."

Since a t-test for KorA vs. KorB is included, one might also consider showing this for KorA vs. KorAB.

p.12 l.426 "Our findings suggest that KorA and KorB exploit RK2 promoter kinetic instabilities (i.e. weak discriminators) to competitively occlude RNAPs from thermodynamically strong promoters."

I'm slightly puzzled by this statement. Could you unpack it a bit to make it more accessible? My intuition would be that slower off-rate kinetics also influence the thermodynamic equilibrium (assuming on-rates don't go down as well).

Version 2:

Reviewer comments:

Reviewer #4

(Remarks to the Author)

The authors have thoroughly addressed all my comments. I congratulate them once again on their excellent work and strongly support its publication.

Decision Letter:

Our ref: NMICROBIOL-24020635B

6th November 2024

Dear Tung,

Thank you for submitting your revised manuscript "Molecular switching of a DNA-sliding clamp to a repressor mediates long-range gene silencing in a multi-drug resistance plasmid" (NMICROBIOL-24020635B). It has now been seen by the original referees and their comments are below. The reviewers find that the paper has improved in revision, and therefore we'll be happy in principle to publish it in Nature Microbiology, pending minor revisions to comply with our editorial and formatting guidelines.

We are now performing detailed checks on your paper and will send you a checklist detailing our editorial and formatting requirements in about two weeks. Please do not upload the final materials and make any revisions until you receive this additional information from us.

Thank you again for your interest in Nature Microbiology Please do not hesitate to contact me if you have any questions.

Sincerely,

Reviewer #4 (Remarks to the Author):

The authors have thoroughly addressed all my comments. I congratulate them once again on their excellent work and strongly support its publication.

Version 3:

Decision Letter:

12th December 2024

Dear Professor Le,

I am pleased to accept your Article "KorB switching from DNA-sliding clamp to repressor mediates long-range gene silencing in a multi-drug resistance plasmid" for publication in Nature Microbiology. Thank you for having chosen to submit your work to us and many congratulations.

You may wish to make your media relations office aware of your accepted publication, in case they consider it appropriate to organize some internal or external publicity. Once your paper has been scheduled you will receive an email confirming the publication details. This is normally 3-4 working days in advance of publication. If you need additional notice of the date and time of publication, please let the production team know when you receive the proof of your article to ensure there is sufficient time to coordinate. Further information on our embargo policies can be found here:

<https://www.nature.com/authors/policies/embargo.html>

Please note that *Nature Microbiology* is a Transformative Journal (TJ). Authors may publish their research with us through the traditional subscription access route or make their paper immediately open access through payment of an article-processing charge (APC). Authors will not be required to make a final decision about access to their article until it has been accepted. [Find out more about Transformative Journals](https://www.springernature.com/gp/open-research/transformative-journals)

Authors may need to take specific actions to achieve [compliance](https://www.springernature.com/gp/open-research/funding/policy-compliance-faqs) with funder and institutional open access mandates. If your research is supported by a funder that requires immediate open access (e.g. according to [Plan S principles](https://www.springernature.com/gp/open-research/plan-s-compliance)) then you should select the gold OA route, and we will direct you to the compliant route where possible. For authors selecting the subscription publication route, the journal's standard licensing terms will need to be accepted, including [self-archiving policies](https://www.nature.com/nature-portfolio/editorial-policies/self-archiving-and-license-to-publish). Those licensing terms will supersede any other terms that the author or any third party may assert apply to any version of the manuscript.

With kind regards,

P.S. Click on the following link if you would like to recommend Nature Microbiology to your librarian <http://www.nature.com/subscriptions/recommend.html#forms>

** Visit the Springer Nature Editorial and Publishing website at [www.springernature.com/editorial-and-publishing-jobs](http://editorial-jobs.springernature.com?utm_source=ejP_NMicro_email&utm_medium=ejP_NMicro_email&utm_campaign=ejp_NMicro) for more information about our career opportunities. If you have any questions please click [here](mailto:editorial.publishing.jobs@springernature.com).**

We are very grateful to the editor and all reviewers for their insightful and supportive comments on our manuscript. We have performed additional experiments and revised the manuscript accordingly. All source data, their replicates, and uncropped gel images have been uploaded; the total number of main figures and extended data figures are six and ten, respectively, according to guidelines from *Nature Microbiology*. An article file with revisions made in Track Changes has also been uploaded. Detailed responses to the specific points that reviewers have raised are given below.

REVIEWER COMMENTS

Reviewer #1 (Remarks to the Author):

KorB is essential for segregation of the broad-host-range multidrug resistance plasmid RK2 plasmid and for regulating gene expression for all basic RK2 functions, including replication and conjugative transfer. In the present manuscript, the authors present that the ParB-homologue is a CTPase, which is stimulated by OB-DNA. OB-DNA together with CTP promote dimerization of KorB. The presented data indicate that the capacity of KorB to bind CTP, close the clamp, and traverse along DNA is essential for long-range gene silencing. The manuscript identifies a direct interaction of KorA with KorB. The impressive crystal structure of the KorB/KorA/operator of KorA-DNA duplex together with the provided functional assays provides a sophisticated explanation of how KorA obstructs KorB diffusion on DNA and enhances retention time at the OA operator. In summary, this is an impressive and technically sound piece of work! I strongly recommend publication.

We thank the reviewer for the encouraging feedback on our manuscript.

To enhance the editorial clarity and refine the manuscript, I suggest the following revision: I believe the discussion can be significantly condensed to eliminate redundancy. Additionally, I question the necessity of including the 'Final Perspectives' section, as it may not contribute substantially to the overall coherence of this splendid manuscript.

We agree, and we have now removed the Final Perspectives section. We also rewrote the last section of the Discussion to be more succinct.

Reviewer #2 (Remarks to the Author):

I have reviewed the MS. by McLean et al. working on gene regulation in a bacterial system by a transcription factor bound to a DNA site distal to the promoter. Although transcription factor regulating gene transcription from a distal DNA site is not common, in most of those cases the action of the transcription factors relies on DNA looping. In the current investigation the authors have conclusively shown that the transcription factor KorB, encoded by the RK2 plasmid in *E. coli*, represses a promoter by binding to a distal DNA site, not by DNA looping, but by sliding down the DNA to the promoter region. KorB encounters another factor, KorA, bound to a DNA site proximal to the promoter. This encounter presumably hinders RNA polymerase binding to the promoter to the promoter. KorB has a CTPase activity which is stimulated by DNA binding. The authors used the state of the art biophysical techniques to demonstrate different steps of the process of KorB action in transcription repression: (i) isothermal titration calorimetry to determine CTP affinity of KorB and various CTP non-binding mutants; (ii) crystal structure determination together with cross-linking experiments identified CTP-dependent clamping of KorB to DNA; (iii) biolayer interferometry CTP-dependent (not hydrolysis) binding of KorB to its specific DNA site; (iv) crosslinking experiments demonstrating that KorA, by interacting with KorB, help engagement (proper dimerization?) of KorB-NTD; (v) crystal structure determination of KorB-NTD/KorA/KorA-DNA binding site ternary complex showing that the KorA/DNA complex facilitates proper dimerization of KorB-NTD that is more compatible for KorB sliding on DNA; and (vi) single molecule experiments with fluorescent-labeled KorA and KorB demonstrating that KorA-OA-DNA complex blocks sliding of KorB-CTP on DNA thus making KorB a transcriptional co-repressor.

The experimental results reported are very clear and convincing. The idea of a protein sliding on DNA has been demonstrated clearly. The paper is also written clearly. I recommend publication of the manuscript in *Nature*. However, I have only a few comments for the authors regarding (a) presentation of data in figure 3, and some discussions and interpretations of the data presented in

figure 6. I suggest that the authors think about my suggestions, as presented below, and to make changes in discussing their data if they agree with my way of thinking.

1) Figures 3 and S3. The amount of reporter gene expressions shown in figure 3b may be presented in absolute amounts, and not in relative amounts to reflect that the various conditions used do not change the promoter strength in a major way.

We agree and we have now included supplementary figures (Extended Data Fig. 2c and 9b) to show absolute values from promoter-*xylE* reporter assays. These supplementary figures complemented well the main figures (Fig. 2c and Fig. 4d) that showed folds of repression. In the legends of main figures, we further explained how folds of repression were calculated and pointed out extra supplementary figures that report absolute values to readers.

2) Figure 6. I always thought that transcription repression in majority of cases in prokaryotic systems are not steric exclusion of RNA polymerase binding by repressors but trapping of bound RNA polymerase at some stages of initiation. (The authors need not buy the idea.) None of the results presented here discard the concept because the way the experiments are designed. For example, a short DNA fragment may enhance dissociation of RNA polymerase from a repressor-DNA-RNA polymerase ternary complex. Indeed, figure 6B shows a small amount of RNA polymerase-DNA-KorA complex. May be, it would be further stabilized by the presence of KorB. I would have liked to see a DNase protection assays done with supercoiled DNA in the presence various combinations of the protein components.

Generally, the field has found that transcription repression of initiation occurs canonically via steric exclusion of RNAPs from the core promoter elements (Browning and Busby, 2016; Rojo, 1999). The most famous example is the *lac* repressor (LacI) and its cognate DNA binding site the *lac* operator, where the operator sits between the transcription start site and the start codon of the first gene in the *lac* operon *lacZ*, blocking most transcription of the operon (Majors, 1975). Another example is the large family of repressors exemplified by MerR, which in the absence of mercury ions acts as a steric exclusion repressor of transcription initiation (Brown et al., 2003). Repressors such as *E. coli* DksA and the F-plasmid-encoded homolog TraR are unique exceptions, as they bind to the RNAP directly independently of a DNA site to repress initiation by altering the RNAP's conformation (Chen et al., 2019).

We apologize for the lack of clarity in the manuscript on explaining why a ternary complex unlikely exists. The native mass spectrum (now Fig. 5b, top left panel) indeed shows minor peaks corresponding to KorA with the *E. coli* RNAP: σ_{70} holoenzyme in two different stoichiometries (RNAP holo:KorA dimer = 1:2 or 1:4) when 2.5-fold excess KorA was added to the promoter DNA-bound holoenzyme. We have now acquired a native mass spectrum of 2.5-fold excess KorA added to the *E. coli* RNAP: σ_{70} holoenzyme *but without DNA*, and no complex of *E. coli* RNAP: σ_{70} + KorA was observed (Extended Data Fig. 10c). This control suggests that the minor peaks in Fig. 5b are either gas state artifacts in the mass spectrometer and/or the result of non-specific binding. We have now mentioned in the main text that “*Minor $E\sigma^{70}$ -KorA peaks were detected (Fig. 5b), but these are non-specific as an addition of 2.5-fold excess KorA to $E\sigma^{70}$ alone i.e. without promoter DNA resulted in no such complex (Extended Data Fig. 10c).*”

On the point that dissociation of RNAP is more possible on linear DNA scaffolds, we do not see this happening with *E. coli* RNAP, which we and other research groups have used to reconstitute many complexes for single particle analysis by cryo-EM. In truth, before native mass spectrometry experiments, electrophoretic mobility shift assays, *in vitro* transcription assays, and prior literature (Williams et al., 1993) had led us to be convinced of a ternary complex. We prepared samples for cryo-EM of both KorA and KorA-KorB added to the promoter DNA-bound *E. coli* RNAP holoenzyme, resulting in densities with end-bound DNAs, an artefactual complex due to linear DNA systems. This suggests that dissociation is less of a problem and that the RNAP simply cannot bind to the core promoter elements when KorAB factors are present in excess, and this agrees with our native mass spectrometry data in Fig. 5b-c.

Reviewer #3 (Remarks to the Author):

The manuscript (ms) "Molecular switching of a DNA-sliding clamp to a repressor mediates long-range gene silencing in a multi-drug resistance plasmid" deals with long-distance gene regulation – a phenomenon known in eukaryotes but relatively rare in bacteria. It focuses on the DNA binding protein KorB that is necessary for plasmid RK2 inheritance but also for regulation of gene expression. It was known previously that KorB binds a silencer sequence called OB (operator of KorB) and that a small DNA-binding protein KorA is also necessary for gene regulation. The aim of the work was to figure out what was the mechanism of KorB's long-range gene expression regulation with respect to its interplay with KorA.

The ms contains a large amount of data from various experimental approaches, including partial structures of KorB, and KorB in complex with KorA and OA DNA. The main finding is that KorB slides along DNA in a CTP-dependent manner, and upon encountering KorA bound to its own operator (OA) near a promoter, changes conformation and forms a stable KorA-KorB complex that inhibits transcription from the promoter.

Comments:

Overall, the ms contains fundamentally novel insights and will be of interest to a wide range of readers. However, there are several discrepancies between the presented results and the suggested model that should be clarified.

1. It is argued in the ms (line 137) that CTP and OB are required for efficient KorB NTD dimerization. However, (line 161) it is also proposed that binding of KorB at OB is CTP-independent. Clearly, Fig. 2B shows binding of KorB to this sequence. How do you know, however, that CTP does not improve it further? Can you comment on it?

Early in this project, we performed a bio-layer interferometry (BLI) experiment to determine the binding affinity of KorB to a linear 40-bp OB-containing DNA duplex in the presence or absence of CTP (see figure below). Without CTP, KorB binds site-specifically at OB with $K_D \sim 140$ nM. In the presence of 1 mM CTP, the KorB-OB DNA-binding is ~ 400 -fold weaker at $K_D \sim 60,000$ nM. These results are consistent with our findings that (i) site-specific binding of KorB at OB is tight and CTP-independent, (ii) CTP and OB are required for KorB NTD dimerization to form a sliding clamp (Fig. 2a), and (iii) a KorB-CTP sliding clamp escapes a high-affinity OB site to run off an open end of a linear DNA (thus, resulting in a low net BLI signal when CTP was included) (see also Extended Data Fig. 3c). We have now added these data to Extended Data Fig. 3d and expanded the relevant Results section to make it clearer to readers.

Binding affinities of KorB to 40-bp OB DNA in the presence and absence of 1 mM CTP.

2. It is suggested that for the KorB movement on DNA, the presence of CTP but not its hydrolysis was necessary (line 182). Therefore, the results of the optical tweezers experiments (Fig. 2B) should be the same with CTP and its non-hydrolyzable analog (CTP γ S). However, KorB seems more localized/less sliding in the left panel where CTP cannot be hydrolysed.

In addition to KorB diffusion, we indeed occasionally observe KorB-OB stable binding events under both CTP and CTP γ S conditions. We interpret these events as either new KorB proteins being loaded at OB or existing KorB proteins diffusing and occasionally re-binding to OB when they re-encounter the sequence. These KorB localizations are likely influenced by the presence of 8xOB cluster(s), which might prevent diffusion of a newly bound KorB if adjacent KorB proteins are already bound. To clarify, we have now included two additional examples for each condition (+CTP and +CTP γ S) in Fig. 2b, showing that KorB diffuses (and the occasional stable binding at OB) happens in both +CTP and +CTP γ S conditions. We have also updated the relevant Results section to make this clearer to readers. Overall, our conclusion remains that CTP binding, but not hydrolysis, is sufficient for KorB diffusion.

Representative examples of kymographs showing diffusion of KorB in the presence of CTP (left panel) or CTP γ S (right panel).

Please, explain. Moreover, what is the model with respect to the force/energy that allows the KorB sliding on DNA when it does not hydrolyse CTP?

In addition to the numerous examples of active translocation of proteins along DNA via hydrolysis of nucleotides (most often ATP), there are cases of protein translocation on DNA by passive diffusion. The passive translocation of a protein on DNA can occur via hopping or sliding, with the latter likely the case for KorB, as we see continuous movement of the protein along the DNA. Passive sliding relies on transient electrostatic interactions between positively charged residues in the protein and negatively charged phosphates in the DNA backbone. This movement has been well described for other sliding clamp proteins, such as PCNA. The ring-shaped PCNA protein contains a lysine patch that interacts with DNA, allowing the protein to move via consecutive interactions of the lysines in a mechanism described as a ‘cogwheel’ mechanism (De March et al., 2017; Yao and O’Donnell, 2017). The lysine side chains can rapidly switch between adjacent phosphates, generating new electrostatic contacts that result in the advancement of the protein. Interestingly, the flexible linker that constitutes the DNA-storage lumen of KorB (and other ParB protein family members) is enriched in lysine residues. We suspect that KorB might have a similar sliding mechanism to PCNA, but we refrain from speculating a concrete model in this manuscript. We have now acknowledged in the Discussion section that the mechanistic details of how KorB slides on DNA are not yet known but is an important topic for future works.

Additionally, CTP hydrolysis seems to play a role in dimerization of KorB (compare Fig. 2A, lanes 7 of left- and right-hand panels) Please, explain.

The reviewer is correct in identifying differences in crosslinking efficiency between KorB S47C vs. KorB (S47C N146A) in the presence of both OB and CTP (see Fig. 2a below, lanes 7 of the left- and right-hand-side panels). However, the low crosslinking efficiency of KorB S47C N146A variant is due

to N146A being deficient in not only CTP hydrolysis *but also CTP-stimulated NTD dimerization*. This can be concluded from crosslinking experiments where a non-hydrolyzable CTP analog (CTPyS) was used instead of CTP (Fig. 2a, lanes 10 of left and right panels). The use of CTPyS also showed that CTP hydrolysis is not required for KorB NTD dimerization (Fig. 2a, left panel, lane 7 vs. lane 10).

To improve the readability and interpretation of data, we repeated the crosslinking experiments in Fig. 2a but have now included the CTPyS conditions (previously from a supplementary figure) in the same main figure.

SDS-PAGE analysis of BMOE crosslinking products of 8 μ M (dimer concentration) of KorB S47C (and variants) \pm 1 μ M 24 bp OB/scrambled (SCR) OB DNA \pm 1 mM CTP. X indicates a crosslinked form of KorB S47C or S47C R117A or S47C N146A.

3. It is claimed that the KorB-KorA interaction takes place at the OA sequence (Fig. 7). However, the presented crosslinking experiments (Fig 4B) were done with OB and showed that KorA increased KorB NTD engagement outside of OA. In Fig. 7, you show this happening only at OA. Modify/explain better in Discussion.

We thank the reviewer for this suggestion. Please also see our response to reviewer 4 (point 17) on a similar concern, where we also performed crosslinking experiments using DNA containing both OB and OA sites. Our experiments in this manuscript altogether suggest that the most relevant KorAB interaction for transcriptional repression is that of KorB-CTP with KorA at the OA site (hence, the model in the original Fig.7 (now Fig. 6). We acknowledge that we have not investigated whether, inside the cell, free KorA (i.e., DNA unbound KorA) and KorB interact together. We reason that mechanisms that reduce/inhibit interaction between free KorA and KorB must exist (because these interactions might not be productive for transcriptional repression), and we hope to explore this avenue more rigorously for a follow-up manuscript. We have now expanded the Discussion to make these points clearer to readers.

4. Abstract – modify the text so that it is clear that KorA already binds DNA. KorB, then, after sliding along DNA and encountering KorA, changes conformation and the resulting KorA-KorB complex inhibits transcription.

Done. We have now modified the Abstract to state that “DNA-bound KorA thus stimulates repression by stalling KorB sliding at target promoters to occlude RNA polymerase holoenzymes.”

5. Except in the Materials and Methods, it is not mentioned which bacterium was the model system used in the study. It would be useful if the reader did not have to look for this information, it should be mentioned in the Introduction.

We have now stated in the Introduction that *E. coli* was the model bacterium used in this study. Specifically, we wrote “By combining biochemistry, structural biology, single-molecule *in vitro* reconstitution, native mass spectrometry, and *in vivo* gene expression assays in *Escherichia coli*, we reveal that KorB is a DNA-dependent CTPase.”

6. Page 2, first paragraph, last sentence. The long-range effect of topology on gene expression should be mentioned, e. g. PMID 9281422, PMID 34951454, PMID 31539491.

We agree, and we have now added the citations and rewritten the last sentence as follows: “*By contrast, gene regulation over kilobase distances is rare in bacteria, and it remains to be determined whether other mechanisms, beyond DNA looping and DNA supercoiling²⁻⁴, are involved.*”

7. Although the results show that N146 is necessary for CTP hydrolysis, no rationale is provided why/how this mutant was designed – N146 is not even shown in Fig. 1D.

Mutations at the equivalent N146 residue in ParB were previously reported to disrupt both CTP hydrolysis and CTP-stimulated NTD dimerization (Jalal et al., 2021; Soh et al., 2019), thus we were motivated to introduce the same N146A mutation into KorB. As expected, KorB (N146A) is defective in both CTP hydrolysis and CTP-stimulated NTD dimerization as assessed by crosslinking assays (Fig. 2a). Subsequent X-ray crystallography data show that N146 resides on a swinging helix $\alpha 3$ whose conformational changes are important for KorB NTD dimerization, thereby explaining the deficiency of N146A in NTD dimerization and the ensuing CTP hydrolysis.

Based on the X-ray crystallography structure of KorB-CTPyS, N146 did not contact the nucleotide (hence, N146 was not originally shown in Fig. 1d). We have now added N146 into Fig. 1d to show its position on helix $\alpha 3$, and we explained briefly in the legend of Fig. 1d the rationale for mutating N146.

8. Figure 2A lane 2 and 3, panel one (wt) vs panel three (nonhydrolyzing mutant): The X-linking efficiencies between the two protein variants differ. Could you explain?

This is a good point. X-ray crystallography data show that N146 resides on a swinging helix $\alpha 3$ whose conformational changes are important for CTP-stimulated NTD dimerization. Mutation in this helix $\alpha 3$, such as N146A, is therefore expected to cause defects in NTD dimerization. The extent of the defect of N146A is, however, dependent on the presence of CTP or not. In the presence of CTP and OB DNA (Fig. 2a, lane 7, left panel vs right panel), N146A did not NTD dimerize well compared to WT. However, in the absence of CTP (Fig. 2a, lanes 2 and 3, left panel vs right panel), N146A showed an elevated level of NTD dimerization compared to WT.

We have added a sentence in the Results section to make it clearer to readers that KorB (N146A) is defective in not only CTP hydrolysis but also CTP-stimulated NTD dimerization/engagement. We wrote “*Furthermore, KorB (S47C N146A), which can bind but not hydrolyze CTP, did not crosslink beyond ~40% (Fig. 2a), suggesting that the N146A substitution also impairs NTD engagement.*”

9. In Figures 3, 5, it is not clear what the „basal“ values were from which the x-fold of repression was calculated. Could you provide primary data to see the background values? Calculating fold changes from near background values can be misleading.

Reviewer 2 also raised this important point. We agree and have now included supplementary figures (Extended Data Fig. 2c and 9b) to show absolute values from promoter-xyIE reporter assays. See also our reply to reviewer 2 (point 1) for more details.

10. Figure 5 A is somewhat confusing. Why using different relative arrangements of OA, OB sites in the two panels? Can you find better representative images so that the relative positions of the OA, OB sites are identical in experiments where you are also comparing different conditions (presence of KorB, KorA, CTP)? Furthermore, in the same Figure, the right-hand side panel: why did not the upper OB site also bind KorB?

During the ligation step to prepare DNA substrates for optical tweezers experiments, we adjusted the DNA/T4 ligase enzyme ratio to allow for the generation of single- or tandem tethers. Tandem tethers can form in four different arrangements regarding the orientation and positioning of the OA

and *OB* sites. If we define a forward (FWD) orientation as 8x*OB*-2x*OA* and a reverse (REV) orientation as 2x*OA*-8x*OB*, the four possible arrangements are FWD-FWD, FWD-REV, REV-FWD, and REV-REV. The selection of these arrangements is random, but since we know the exact positions of the *OA* and *OB* in base pairs for each DNA, we could easily determine the specific arrangement during data analysis, by examining the KorA/KorB binding positions.

We agree with the reviewer that the previous figure was confusing due to insufficient explanation of the design of our experiments. We have now conducted extensive new measurements, increasing our pool of events to a total of 182 molecules. The revised figure also now shows molecules with the same orientation to facilitate interpretation from the readers (Fig. 4a-b).

Additionally, we engineered a DNA substrate with 2 copies of the *OA* sequence (2x*OA*) to increase the chances of KorA protein binding to this site. The larger pool of events has allowed us to identify four cases regarding KorB localization and simultaneous binding of KorA and KorB. See also our response to Reviewer 4's specific comment related to this point (Reviewer 4-point 4).

In sum, we have now included these additional analyses in the main text, modified figures accordingly, and added a new supplementary figure (Extended Data Fig. 8a) to show several representative images for each of the four cases.

Representative kymographs showing the distribution of KorA and KorB in the presence of CTP in a DNA containing 2x*OA* sites and 8x*OB* sites for the four cases described. Quantification of the frequency of those four cases (45%, 21%, 25%, and 9% of the total $n = 182$ events) is shown in each panel.

Reviewer #4 (Remarks to the Author):

McLean et al. present new insights into the function of KorB, a ParB-like protein involved in the partitioning and gene regulation of the plasmid RK2. KorB can repress genes at a distance and cooperates with its binding partner KorA, however, the underlying mechanisms remained unclear.

The authors show that KorB is a CTPase with characteristics similar to what has been reported for other ParB-like proteins. They present a crystal structure of a KorB fragment comprising the N-

terminal CTP-binding domain (NTD) and the middle DNA-binding domain (DBD), bound to the CTP analogue CTPgS. The structure shows engaged NTDs, consistent with previous observations for other ParB-like CTPases. Cross-linking assays suggest that CTP promotes NTD engagement in the presence of the KorB-binding DNA sequence OB, confirming its ParB-like behavior.

The authors then employ a C-trap single-molecule setup to characterize DNA association of KorB. Recruitment of KorB to OB is independent of CTP. In the presence of CTP or CTPgS, however, KorB appears to release OB sites and diffuse along DNA. Bio-layer interferometry experiments using double-tethered or single-tethered DNA suggest that both nucleotides promote the conversion of KorB to a sliding clamp. In vivo gene expression experiments using KorB mutants defective in CTP binding or hydrolysis suggest that CTPase activity is required for long-distance repression.

Next, the authors show that KorA promotes engagement of the NTD even in the absence of nucleotide. They solve a crystal structure of the KorB NTD DBD fragment in complex with KorA and the KorA-binding DNA sequence OA. The NTDs are engaged in the absence of nucleotide and KorA contacts the bottom of the DBD. The authors then include KorA in their C-trap assay and find that the addition KorB and CTP increases its residence time on OA. Furthermore, KorA seems to act as a diffusion roadblock for KorB. The authors confirm that KorA and KorB have a strong synergistic effect on gene repression, and that this synergy is disrupted by mutations in the KorAB interface.

Finally, the authors propose that KorAB repression works best on promoters with weak sigma70 binding, based on in vitro transcription initiation experiments and displacement studies using native mass-spectrometry. They conclude with a model that aims to explain long-distance repression by KorAB. KorB associates with the RK2 plasmid at OB sites, converts to a sliding clamp upon CTP binding, and diffuses to a target promoter bound by KorA. KorAB then efficiently occludes the sigma factor binding site and represses the promoter.

I found the data and model intriguing, and the manuscript engaging and well-presented. The paper highlights how, in principle, a single binding site for a repressor can be used to regulate many genes by loading a sliding clamp. The paper also points out how this type of regulation can be modulated by additional factors, which may be used to fine-tune its effect on different target genes. I feel the story is ready for publication.

Here are a few comments/questions which I hope are helpful:

1. Any ideas on how, if at all, genes under KorAB repression can be induced? Does this solely depend on KorB levels?

This is an excellent question by the reviewer. We speculate that genes under KorAB repression likely can be induced during plasmid replication, either during normal cell division or before conjugation, whereby rolling circle replication likely strips all the DNA-bound proteins off the plasmid allowing a burst of gene expression to occur from these operons before KorAB repress the genes again. Another possibility is that during cell growth protein levels are reduced/diluted (Klumpp et al., 2009) and this may represent a window of opportunity for genes under KorAB repression to be induced. Since these are speculations with no direct experimental data yet and our manuscript is already well over the word count limit, we feel it is prudent not to include this in the Discussion. However, we are happy to include them if the reviewer and the editor insist.

2. Can we exclude loop formation? Is this notion based on the roadblock experiments from the literature and the magnetic tweezer experiments presented here?

There was no evidence of loop formation in our optical tweezers experiments. Such DNA loops would be indicated by reductions in extensions under force-clamp mode or increases in force under constant-distance mode. As a representative example of the expected outcomes if DNA loops were formed in C-trap experiments, we refer the reviewer to our recent work on the activity of human HELB helicase (Hormeno et al., 2022). Additionally, our magnetic tweezers experiments (see figure

below, now in Extended Data Fig. 4) did not detect any signs of condensation, which could suggest DNA loop formation.

I was wondering if there is any form of condensation in magnetic tweezer experiments when KorA and OA are included in addition to KorB/OB.

The experiments shown in the original Fig. S2C were conducted on DNA that contains a 1xOA sequence. In the original manuscript, we omitted this detail from the cartoon to avoid confusion, but we have now added the label back in the appropriate position for the revised manuscript. Additionally, we performed magnetic tweezers experiment with 1 μ M KorB + 1 μ M KorA + 2 mM CTP to determine whether condensation might occur in the presence of both proteins, the force-extension curves indicated that no condensation was induced by these two proteins. We have now added these latest data to an updated Extended Data Fig. 4 and updated the Results section.

Cartoon showing the magnetic tweezers components and the DNA used for the MT experiments with the location of the 16xOB and the 1xOA sites. Average force-extension curves of bare 16x OB DNA molecules ($n = 56$) and in the presence of different concentrations of KorB + 2 mM CTP (500 nM, $n = 11$, and 1 μ M, $n = 13$) or Bs ParB + 2 mM CTP (500 nM, $n = 11$, 1 μ M, $n = 21$ and 2 μ M, $n = 17$), and in the presence of 1 μ M KorB + 1 μ M KorA + 2 mM CTP ($n = 10$).

3. The background signal in the C-trap experiments is a bit unclear. What does this look like in the absence of OB? Showing this could also confirm that CTP-bound diffusing KorB was recruited via OB.

We had previously performed control experiments with DNA molecules lacking an OB site but did not include them in the original manuscript. Based on the reviewer's suggestion, we have now repeated these experiments and added them to the manuscript as a supplementary figure (Extended Data Fig. 3a). These experiments employed a ~21 kb DNA that contains a single OA site but no OB site. We incubated this DNA with AF488-KorB, and we did not observe any binding event, only a high background from the fluorescently labeled protein (see figure below, now in Extended Data Fig. 3a).

The binding of KorB to OB is indeed sequence-specific because experiments with OB-containing DNA *in the absence of CTP* (Fig. 2b) did not show KorB binding outside the OB cluster. Therefore, we concluded that the presence of KorB-CTP outside the OB requires initial specific binding of KorB to OB, followed by diffusion of CTP-bound KorB.

Representative cartoon showing a DNA containing 1xOA trapped between the two beads. Scans showing the DNA trapped under the presence of 50 nM KorB and 2 mM CTP (upper panel) and the same molecule after one minute of incubation (lower panel).

4. Fig. 5D: KorB accumulates in a somewhat narrow zone close to OA. Why is that? I would expect it to diffuse back and forth over the full length of the DNA, and perhaps form a gradient towards OA. Alternatively, if it forms a stable complex with KorA, I would expect the proteins to co-localize. I think this should be discussed, as I feel it potentially contradicts the proposed model.

We thank the reviewer for pointing this out. We have conducted extensive new measurements, expanding our pool of events to a total of 182 molecules. Additionally, we have engineered a DNA substrate with 2 copies of the OA sequence (2xOA) to increase the chances of KorA protein binding to this site (new Fig. 4b and Extended Data Fig. 8a). Crucially, the larger pool of events allowed us to identify four distinct cases regarding KorB localization and the simultaneous binding of KorA and KorB. N.B. These measurements were performed outside of the protein channel to reduce the fluorescence background.

Case I: KorB localization between the OA and OB sites. This is the most frequent case occurring in 45% of the total cases. We occasionally observe trespassing of KorB beyond the OA site. We interpret this trespassing as either a transient disruption of the KorA-KorB complex or as KorB diffusing beyond the OA before the complex formation.

Case II: KorB localizes between OA and the OB-proximal bead (~21% occurrence). This could alternatively be considered a subtype of Case I where KorB proteins were trapped between the OB site and the proximal bead. Together Case I and II account for 66% of the measured activities. Case II events can be explained as follows: a KorB protein binds to OB and, in the presence of CTP, can freely diffuse in either direction. A subsequent KorB protein binding to OB might act as a roadblock, preventing the first KorB protein from passing through the OB site and thereby restricting KorB proteins to either side of the OB site. The higher occurrence of Case I compared to Case II is likely due to the shorter distance between OB and OA compared to that between OB and the proximal bead, reflecting a higher chance of KorB detachment in the latter case. The absence of a gradient of protein from the OB site suggests a lower number of diffusing KorB proteins, which we estimate to be few. This limited number allowed us to track and obtain diffusion constants.

Case III: stable co-localizations of both proteins at OA (~25% occurrence). This case likely reflects the scenario where a single diffusing KorB protein binds to KorA. Importantly, we never observed KorB localization at OA sites in the absence of KorA protein.

Case IV: mostly static binding of KorA and KorB at OA and OB sites, respectively (least frequent at ~9% occurrence).

Overall, all these different cases can be accommodated in the proposed model (Fig. 6) and reflect different stages of the binding and diffusing activity of the KorAB system. We have now extended the Results section to include these additional analyses, updated Figure 4 to include new data, and added a new supplementary figure (Extended Data Fig. 8) to show more representative binding events.

Representative kymographs showing the distribution of KorA and KorB in the presence of CTP along a DNA containing 2xOA sites and 8xOB sites for the four described cases. Quantification of the frequency of each of the four cases (45%, 21%, 25%, and 9% of the total $n = 182$ events) is shown in each panel.

5. Introduction: “Long-range gene regulation by the DNA-binding protein KorB is essential for stable vertical inheritance and horizontal transmission of the broad-host-range multidrug resistance plasmid RK2, as well as for the fitness of its host bacterium”. I am not sure if the cited publications show that gene regulation is an essential function of KorB. Are there any experiments that uncouple gene regulation from the more traditional ParB-like partitioning activities?

It is not possible to knock out the KorB-encoding gene from RK2/RP4/RP1 plasmid family because the host bacteria would not be viable. N.B. Many canonical plasmidic *parB* can be deleted and the resulting plasmids are unstable i.e., plasmids are lost more frequently from cells, but the host bacteria (now devoid of such plasmids) are still viable. Originally, *korB* was discovered in a screen for factors that suppressed such killing of the host (hence, the namesake *kill override B*), which would otherwise happen due to uncontrolled protein production from plasmid genes (Figurski and Helinski, 1979; Smith and Thomas, 1983).

While it is difficult to cleanly uncouple the roles of KorB in gene regulation from plasmid partitioning due to its essentiality, there are several reasons to suggest that KorB roles in gene regulation is important.

- (i) KorB is required to suppress the killing of the host, not just for the stability of the RK2 plasmid.
- (ii) There are 12 conserved binding sites of KorB (OB) on promoters of RK2 genes that encode functions for stable vertical inheritance and horizontal transmission, among them a single OB site (OB3) is sufficient for plasmid segregation (Williams et al., 1998).

- (iii) The cooperation of KorB and another DNA-binding protein KorA strengthens the transcriptional repression.

Overall, we propose to tone down the sentence in the Introduction to state that gene regulation by KorB is *important* (rather than using the original word *essential*) for stable vertical inheritance and horizontal transmission of RK2, as well as for the fitness of its host bacterium. We also added the two references (Figurski and Helinski, 1979; Smith and Thomas, 1983) on the discovery of KorB from a screen for factors that suppress the killing of the host

6. Introduction: The conserved role of ParABS systems in plasmid partitioning should be mentioned in addition to their role in chromosome segregation.

Done

7. Introduction: The KorA binding site OA should be mentioned.

Done

8. Fig. 1D: Possible to show N146, since targeted by mutagenesis?

We agree and have now shown the position of N146 on helix $\alpha 3$ (near P-motif 3) in Fig. 1d. See also our reply to reviewer 3 (point 7).

9. Cross-linking: The mutants used seem to affect basal cross-linking levels. Why is that?

Reviewer 3 raised a similar concern, please see our reply to reviewer 3 (point 8).

10. Cross-linking: OB DNA was used in sub-stoichiometric amounts (1 μ M DNA and 8 μ M KorB), yet KorB cross-linking is close to 90%. Does this mean that something catalytic is going on? What happens at different concentrations of DNA?

The reviewer is correct that *OB* acts as a catalyst. In the presence of CTP, KorB escapes from the *OB* site to diffuse away and runs off the open end of a linear DNA. The now vacated *OB* site is available for the next KorB-CTP to load on and subsequently diffuse away. Sub-stoichiometric concentrations of *OB* are therefore sufficient to support many cycles of KorB loading-escape-diffusion. To better demonstrate this point, we performed crosslinking experiments with KorB (S47C) + CTP and increasing concentrations of *OB* (from 1/128 to 2/1 *OB*-to-KorB molar ratio) to observe that ~16-fold less *OB* to KorB was sufficient to achieve maximal crosslinking (see figure below). We have now presented these data in a new supplementary figure (Extended Data Fig. 2b) and pointed out that a sub-stoichiometric concentration of *OB* DNA was used for crosslinking experiments in the legend of Fig. 2.

N.B. *parS* DNA in the case of ParB-CTP clamp was also reported previously to serve as a catalyst (Antar et al., 2021; Jalal et al., 2021; Osorio-Valeriano et al., 2021; Soh et al., 2019).

SDS-PAGE analysis of BMOE crosslinking products of 8 μ M KorB S47C with 1 mM CTP and increasing concentrations of 24-bp OB DNA.

11. Cross-linking: WT KorB shows decreased NTD cross-linking in the presence of CTP, but increased cross-linking when CTPgS is added. The protein responds differently to the two nucleotides, possibly due to hydrolysis of CTP. N146A does not hydrolyze CTP, but responds to CTP similarly to WT (decreased cross-linking) and does not respond much to CTPgS or CTP/CTPgS + OB. I expected that N146A would cross-link more when either nucleotide is used, which it doesn't. Does this suggest that the differences between CTP/CTPgS conformations are, at least in part, due to the different nucleotides used rather than nucleotide hydrolysis?

KorB (N146A) is defective in both CTP hydrolysis and also in *CTP-stimulated NTD dimerization* (see our reply to points 7 & 8 from reviewer 3), thus explaining why KorB N146A did not crosslink more when CTP or CTPyS were used. We have now modified the text to make it clear to readers. Regarding the difference in KorB (S47C) crosslinking efficiency between +CTP vs. +CTPyS, it is most easily explained by KorB's inability (or very slowly) to hydrolyze CTPyS, thus preventing KorB NTDs, once dimerized, from being released from each other. However, we also acknowledge that KorB has a higher affinity to CTPyS than CTP (ITC data, Fig. 1b vs. Extended Data Fig. 1b) so the higher crosslinking efficiency in CTPyS condition could be partly due to this. We have now stated this caveat in the revised manuscript. We wrote "*How CTPyS promoted KorB NTD engagement without OB DNA is not yet clear, but could be due to its higher binding affinity to KorB than CTP (Fig. 1b and Extended Fig. 1b)*".

N.B. Other non-hydrolyzable analogs such as CMPPNP or CMPPCP do not bind to ParB/Noc/KorB protein family members (Jalal et al., 2020). Furthermore, during the course of the investigation, we constructed and purified ~16 KorB variants to search for one that was defective in CTP hydrolysis but still competent in clamp closing but was not successful. Overall, despite our extensive effort, we were limited by the availability of suitable reagents.

CTP and CTPgS may not be fully equivalent, and the ITC data presented here perhaps supports this view (much higher affinity for CTPgS). I believe similar observations have been made for other proteins and nucleotide/analogue pairs. The CTPgS bound structure might not fully reflect the CTP bound state, but perhaps relate to a transition state, and one might want to reconsider modelling the nucleotide as CTPgS to avoid confusion (even if the position of the sulphur atom is unknown).

Regarding the crystal structure, given the resolution of the data, we cannot be fully confident in the precise details of bound ligands. However, the electron density appears to be continuous between the beta and gamma phosphates and therefore does not suggest that we are seeing a transition state, although a mixture of states could be a possibility. Nevertheless, given that cleavage of the gamma phosphate is less likely for CTPyS, we would not expect to see a transition state anyway. We could use the B-factors of the oxygens surrounding the gamma phosphate of the modeled CTP to guide the placement of the sulfur of CTPyS, where a lower oxygen B-factor could indicate a higher probability that the atom is actually a sulfur. However, a lower B-factor could simply be the result of

a more ordered oxygen due to stronger interactions with neighboring atoms. Moreover, if this were considered to be a reliable guide and the placement of the sulfur is not random, then we would expect a similar distribution of the oxygen B-factors across the 6 equivalent ligands in the asymmetric unit. However, we do not see a consistent B-factor distribution. We are then left with the two options of leaving the ligands as CTP (i.e. current situation) or rebuilding them as three overlaid CTP γ S molecules differing only in the placement of the sulfur with occupancies set to 0.33. We feel that the latter solution is overly complicated and more likely to confuse PDB users with less structural biology experience, we hope the reviewer agrees.

12. Fig. 2D: Labels are a bit unclear. What are the numbers in the plot and gradient referring to?

After examining all figures in the original manuscript, we assume that the reviewer referred to the original Fig. 2b rather than 2d? We have now updated the relevant figure legend. The scale bar (0 to 1) refers to the fluorescence intensity of the kymographs. The numbers on the x-axis (0 to 0.3) refer to the fluorescence intensity reported in the line graphs.

13. C-trap experiment: “To amplify the signal of KorB DNA binding, eight OB sites were present either in one or two clusters on the DNA (Figure 2B)”. Why one OR two clusters? This needs some explanation. The figure should make this clearer in the cartoon, which currently suggests that two clusters were present.

Due to the low binding affinity of KorB and KorA, DNA binding is not observed in all cases (especially when we had to use a low concentration of fluorescently labeled KorAB to reduce background fluorescence in our C-Trap setup). To address this, we used DNA containing 8xOB to increase the amount of KorB loaded at the OB sites. During the fabrication of the DNA, we could produce either single or tandem tethers (see an explanation in our reply to point 10 in Reviewer 3’s comments). A single tether would contain 8xOB and 1xOA, while a tandem tether would contain two OB clusters, each with 8xOB, and two OA sites.

Measuring with these tandem tethers increases the probability of observations, which is particularly useful for quantification (for example, Fig. 4a-b). We have now better labeled the cartoons and provided a more detailed explanation in the Methods section, as well as in the relevant figure legend.

14. C-trap experiment: What is meant by “to amplify the signal” (see above)? Does clustering of OB lead to stronger fluorescence, an increased probability of observations, or both?

Our aim was to increase the probability of observations because, even with 8xOB, we did not observe binding in all DNA molecules. For the new characterization of the KorB-KorA complex and the description of the four cases (Fig. 4b), we used DNA with 2xOA to also increase the probability of KorA binding. We have now used the phrase “to increase the probability of observation” rather than “to amplify the signal” in the revised manuscript to be more precise.

15. Repression experiments: Do you know whether the reporter plasmid has the same copy number +/- KorA/B? Differences in replication or stability could influence expression levels.

Based on the reviewer’s feedback, we used Illumina whole-genome deep sequencing to determine the copy number (relative to the chromosome) of each of the three plasmids used in the promoter-xy/E reporter assays. We observe no significant difference in plasmid copy number under any tested conditions (one-way ANOVA statistical analysis, see figure below, now in Extended Data Fig. 9a). These data have now been presented in a new supplementary figure (Extended Data Fig. 9a) and the conclusion was mentioned in the Result section.

Copy number of plasmids used in promoter-xylE reporter assays in relation to the chromosome.

16. Fig. 3D: Do all bands come from the same Western blot membrane? Is it possible to show a loading control, e.g. a stained membrane?

All bands were indeed from the same western blot membrane. Loading controls (Coomassie-stained gels) and un-spliced western blots have now been presented in supplementary figures (Extended Data Fig. 2c and Extended Data Fig. 9b). All raw uncropped images of western blots have also been uploaded to a public repository (Mendeley) as recommended by Nature Microbiology.

17. The KD of KorAB formation is rather low (4 μ M). Is binding stronger in the presence of OA? Has KorB cross-linking been tested in the presence of OA?

We thank the reviewer for this point. The K_D of KorA-KorB interaction was measured by ITC. We could not apply such a technique to measure KorA-KorB interaction in the presence of OA DNA because the heat-exchange signal from KorA-OA DNA binding masks that from a weaker KorB-KorA binding. Instead, we have now performed additional crosslinking experiments of KorB + CTP \pm KorA in the presence of a DNA duplex that contains both OB and OA sites (or controls: DNA containing a scrambled OB and OA or DNA containing OB and scrambled OA). These results have now been added to Extended Data Fig. 5a (the same figure is presented below). In essence, the presence of OA did not further improve crosslinking efficiency (lane 2 vs. lane 4, lane 6 vs. lane 8, see figure below). These indirect data suggest that, in these conditions, the presence of OA might not improve the binding affinity between KorB and KorA.

SDS-PAGE analysis of BMOE crosslinking products of KorB S47C ± KorA + 1mM CTP ± 24 bp DNA containing both OB and OA sites (OB_OA) or both scrambled OB site and OA (OB^{SCR}_OA) or both OB and scrambled OA (OB_OA^{SCR}).

18. Notably, for both KorB R117A and N146A variants, which are defective in NTD-engagement, the presence of KorA further increases the efficiency of crosslinking in all tested conditions (Figure S4A vs. Figure 2A) – This would be much more convincing if Figure S4A contained a control without KorA, instead of referring to Figure 2A.

We agree, and we have now repeated the crosslinking experiments in the previous Figure S4A to include a KorA-minus negative control. Figure S4A (now Extended Data Fig. 5b) has been updated in the revised manuscript.

19. Fig. 5C: The residence time of KorA Y84A is compared to KorA WT. I would find this experiment even more convincing if the residence time of KorA WT could be tested in the presence of KorB and KorB F249A, as this would use the same KorA protein as the analyte.

This is an excellent suggestion. We have now performed the recommended experiments with AF647-KorA WT and KorB (F249A). Representative kymographs have been added to Extended Data Fig. 8b and the quantified data to the graph in Fig. 4c (see figure below). In conclusion, KorB F249A did not enhance the residence time of KorA on OA DNA, which is consistent with other findings in our manuscript that KorA-KorB interaction stabilizes KorA on DNA.

Box plot showing the retention times of AF647-KorA in the presence of KorB (2.61 ± 0.24 s, mean \pm SEM, $n = 70$) or KorB-CTP (10.11 ± 0.56 s, mean \pm SEM, $n = 125$), and the retention time of AF647-KorA (Y84A) variant in the presence of KorB-CTP (3.72 ± 0.29 s, mean \pm SEM, $n = 148$), and the retention time of AF647-KorA in the presence of the KorB (F249A) variant and 2 mM CTP (3.07 ± 0.23 s, mean \pm SEM, $n = 126$).

20. In vitro transcription experiment: “Transcription initiation assays confirmed that KorA and KorB (with and without CTP) individually function as repressors (Figure S7A and B), while their combined activity is further augmented, consistent with our in vivo and single-molecule studies.” I am not convinced that this system properly reconstitutes the synergy between KorA and KorB. The reported synergistic effect size, if any, is tiny and nowhere near the massive increase of KorAB-mediated repression in vivo.

We thank the reviewer for bringing up this concern. We have now edited the main text to emphasize that we observed an additive effect instead of a cooperative/synergistic effect at the transcription level in our *in vitro* transcription assay, and removed the previously used phrase “consistent with *in vivo* and single-molecule studies”. *In vitro* transcription assays, due to their reduced nature, cannot recapitulate transcription repression in vivo, as the concentrations of factors and nutrients will be different in our setup versus in cells. Despite this, we believe our *in vitro* transcription assay does correctly report on the biochemistry behind KorA-KorB repression of plasmid RK2 promoters, as the switch with a strong phage λ promoter discriminator not only dramatically alters the promoter complex half-life, it also statistically significantly disables repression by KorA-KorB. Assays such as ours using *E. coli* RNAP and the same conditions of buffers and substrate concentrations have been used successfully in many transcription papers to recapitulate all stages of the transcription cycle (Braffman et al., 2019; Delbeau et al., 2023; Gopalkrishnan et al., 2017; Kang et al., 2018, p. 201; Molodtsov et al., 2023; You et al., 2023).

21. Discussion: “we propose that KorB-CTP loads at the OB site and then switches to a closed-clamp conformation through NTD engagement” and “Once KorB-CTP slides away, another KorB-CTP molecule can bind a now-free OB and repeat the process”. Don’t the structure and single-molecule experiments suggest that nucleotide-free KorB associates with OB, and slides away upon CTP binding, rather than KorB-CTP recognizing OB?

We agree, and we have now modified the Discussion accordingly.

22. Discussion: “and the clamp-trapping ability of CTCF is crucial to switch the function of cohesin between sister chromosome cohesion and gene expression regulation and genome folding”. I’m not sure this statement is precise. The switch to genome folding i.e. loop extrusion does not require CTCF.

We thank the reviewer for pointing this out, and we have now removed this sentence.

23. Is it stated somewhere if protein concentration refers to monomers or dimers?

This information has now been included in the relevant parts in the Methods section and figure legends.

24. I recommend including a strain table. A table of proteins containing the exact constructs used would be an additional plus (including tags, linkers, extra residues etc.). Adding supplemental DNA sequences used in binding experiments and gene expression studies would also be great.

We agree, and we have now included such tables for strains, plasmids, proteins, and oligonucleotides in supplementary tables 1-4. Sequences of DNA used in BLI/C-Trap/*in vitro* transcription experiments and promoter-*xyIE* reporter assays are now listed in the Supplementary Table 6.

References

- Antar H, Soh Y-M, Zamuner S, Bock FP, Anchimiuk A, Rios PDL, Gruber S. 2021. Relief of ParB autoinhibition by parS DNA catalysis and recycling of ParB by CTP hydrolysis promote bacterial centromere assembly. *Sci Adv* **7**:eabj2854. doi:10.1126/sciadv.abj2854
- Braffman NR, Piscotta FJ, Hauver J, Campbell EA, Link AJ, Darst SA. 2019. Structural mechanism of transcription inhibition by lasso peptides microcin J25 and capistruin. *Proc Natl Acad Sci U S A* **116**:1273–1278. doi:10.1073/pnas.1817352116
- Brown NL, Stoyanov JV, Kidd SP, Hobman JL. 2003. The MerR family of transcriptional regulators. *FEMS Microbiol Rev* **27**:145–163. doi:10.1016/S0168-6445(03)00051-2
- Browning DF, Busby SJW. 2016. Local and global regulation of transcription initiation in bacteria. *Nat Rev Microbiol* **14**:638–650. doi:10.1038/nrmicro.2016.103
- Chen J, Gopalkrishnan S, Chiu C, Chen AY, Campbell EA, Gourse RL, Ross W, Darst SA. 2019. E. coli TraR allosterically regulates transcription initiation by altering RNA polymerase conformation. *Elife* **8**:e49375. doi:10.7554/eLife.49375
- De March M, Merino N, Barrera-Vilarmau S, Crehuet R, Onesti S, Blanco FJ, De Biasio A. 2017. Structural basis of human PCNA sliding on DNA. *Nature Communications* **8**:13935. doi:10.1038/ncomms13935
- Delbeau M, Omollo EO, Fromm R, Koh S, Mooney RA, Lilic M, Brewer JJ, Rock J, Darst SA, Campbell EA, Landick R. 2023. Structural and functional basis of the universal transcription factor NusG pro-pausing activity in Mycobacterium tuberculosis. *Mol Cell* **83**:1474-1488.e8. doi:10.1016/j.molcel.2023.04.007
- Figurski DH, Helinski DR. 1979. Replication of an origin-containing derivative of plasmid RK2 dependent on a plasmid function provided in trans. *Proceedings of the National Academy of Sciences* **76**:1648–1652. doi:10.1073/pnas.76.4.1648
- Gopalkrishnan S, Ross W, Chen AY, Gourse RL. 2017. TraR directly regulates transcription initiation by mimicking the combined effects of the global regulators DksA and ppGpp. *Proc Natl Acad Sci U S A* **114**:E5539–E5548. doi:10.1073/pnas.1704105114
- Hormeno S, Wilkinson OJ, Aicart-Ramos C, Kuppa S, Antony E, Dillingham MS, Moreno-Herrero F. 2022. Human HELB is a processive motor protein that catalyzes RPA clearance from single-stranded DNA. *Proc Natl Acad Sci U S A* **119**:e2112376119. doi:10.1073/pnas.2112376119
- Jalal AS, Tran NT, Le TB. 2020. ParB spreading on DNA requires cytidine triphosphate in vitro. *eLife* **9**:e53515. doi:10.7554/eLife.53515
- Jalal AS, Tran NT, Stevenson CE, Chimthanawala A, Badrinarayanan A, Lawson DM, Le TB. 2021. A CTP-dependent gating mechanism enables ParB spreading on DNA. *Elife* **10**:e69676. doi:10.7554/eLife.69676
- Kang JY, Mooney RA, Nediakov Y, Saba J, Mishanina TV, Artsimovitch I, Landick R, Darst SA. 2018. Structural Basis for Transcript Elongation Control by NusG Family Universal Regulators. *Cell* **173**:1650-1662.e14. doi:10.1016/j.cell.2018.05.017
- Klumpp S, Zhang Z, Hwa T. 2009. Growth rate-dependent global effects on gene expression in bacteria. *Cell* **139**:1366–1375. doi:10.1016/j.cell.2009.12.001
- Majors J. 1975. Initiation of in vitro mRNA synthesis from the wild-type lac promoter. *Proc Natl Acad Sci U S A* **72**:4394–4398. doi:10.1073/pnas.72.11.4394

- Molodtsov V, Wang C, Firlar E, Kaelber JT, Ebright RH. 2023. Structural basis of Rho-dependent transcription termination. *Nature* **614**:367–374. doi:10.1038/s41586-022-05658-1
- Osorio-Valeriano M, Altegoer F, Das CK, Steinchen W, Panis G, Connolley L, Giacomelli G, Feddersen H, Corrales-Guerrero L, Giammarinaro PI, Hanßmann J, Bramkamp M, Viollier PH, Murray S, Schäfer LV, Bange G, Thanbichler M. 2021. The CTPase activity of ParB determines the size and dynamics of prokaryotic DNA partition complexes. *Mol Cell* **81**:3992-4007.e10. doi:10.1016/j.molcel.2021.09.004
- Rojo F. 1999. Repression of transcription initiation in bacteria. *J Bacteriol* **181**:2987–2991. doi:10.1128/JB.181.10.2987-2991.1999
- Smith CA, Thomas CM. 1983. Deletion mapping of kil and kor functions in the trfA and trfB regions of broad host range plasmid RK2. *Mol Gen Genet* **190**:245–254. doi:10.1007/BF00330647
- Soh Y-M, Davidson IF, Zamuner S, Basquin J, Bock FP, Taschner M, Veening J-W, De Los Rios P, Peters J-M, Gruber S. 2019. Self-organization of parS centromeres by the ParB CTP hydrolase. *Science* **366**:1129–1133. doi:10.1126/science.aay3965
- Williams DR, Macartney DP, Thomas CM. 1998. The partitioning activity of the RK2 central control region requires only incC, korB and KorB-binding site O(B)3 but other KorB-binding sites form destabilizing complexes in the absence of O(B)3. *Microbiology (Reading)* **144 (Pt 12)**:3369–3378. doi:10.1099/00221287-144-12-3369
- Williams DR, Motallebi-Veshareh M, Thomas CM. 1993. Multifunctional repressor KorB can block transcription by preventing isomerization of RNA polymerase-promoter complexes. *Nucleic Acids Res* **21**:1141–1148. doi:10.1093/nar/21.5.1141
- Yao NY, O'Donnell M. 2017. DNA Replication: How Does a Sliding Clamp Slide? *Curr Biol* **27**:R174–R176. doi:10.1016/j.cub.2017.01.053
- You L, Omollo EO, Yu C, Mooney RA, Shi J, Shen L, Wu X, Wen A, He D, Zeng Y, Feng Y, Landick R, Zhang Y. 2023. Structural basis for intrinsic transcription termination. *Nature* **613**:783–789. doi:10.1038/s41586-022-05604-1

We are very grateful to the editor and all reviewers for their insightful and supportive comments on our manuscript. We have now revised the manuscript accordingly. The manuscript has also been shortened by ~2500 words to fit with the guidelines by *Nature Microbiology*, according to the instructions from the editor. Detailed responses to the specific points from the editor can be found in the cover letter, and an article file with revisions made in Track Changes has also been uploaded. Detailed responses to comments and suggestions from reviewer 4 are given below.

Reviewer #2 (Remarks to the Author):

I have reviewed the revised manuscripts. I am happy with the response of the authors in revising the manuscript. It now presents with strong evidence to show that an enhancer bound protein Slides on DNA to exert its action to repress transcription at the cognate promoter. It is a very novel mechanism of enhancer action. I recommend publication of the manuscript.
Sankar Adhya

Thank you

Reviewer #3 (Remarks to the Author):

The authors have thoroughly answered my questions. Congratulations on an excellent piece of work!

Thank you

Reviewer #4 (Remarks to the Author):

The authors have addressed all my comments and have included additional experimental evidence and controls to support their reasoning. I feel that the manuscript has significantly improved and should be published. I congratulate all authors on this exciting story.

Thank you

I have only a few minor comments that I hope are helpful:

I may have missed this, but is there an explanation for why the background in Fig. 4a (AF-KorB + AF-KorA) is so high? If there is a technical reason, this should be mentioned somewhere.

The imaging part of the C-trap experiments was performed either inside the protein channel (containing fluorescently labeled AF-KorA and AF-KorB, channel 4) or in a buffer-only channel (channel 3) after a 60s pre-incubation in the protein channel, and this is the origin of the higher background fluorescence in Fig. 4a. The experiment shown in the upper panel of Fig. 4a was performed in channel 4, and the lower panel in channel 3. We had to do so because the residence time of KorA alone was so short that its binding to DNA did not survive the transfer from channel 4 to channel 3.

On the contrary, to observe KorB diffusion, we had to lower the fluorescence background (Fig. 4b), thus these experiments were performed in channel 3. To calculate the residence times of KorA, we performed all experiments in the protein channel using the unlabeled version of KorB and reduced the concentration of KorA to 20 nM to minimize background fluorescence.

We originally mentioned this in the "confocal optical tweezers experiments" section of the Methods. We have now added this explanation to the legends of main Figures and Extended Data Figures to make it clearer/more accessible to readers.

p.9 I.329 "A subsequent KorB protein binding to OB might act as a roadblock, preventing the first KorB protein from passing through the OB site, thereby restricting KorB proteins to either side of OB." Case II does not require a roadblock at OB, does it? KorB could just pass through OB and localize over the whole segment between OA and bead.

The reviewer is correct. However, to explain the difference between Case I (KorB accumulation between OB and OA) and Case II (KorB accumulation between the bead and the OB), we

acknowledge that additional KorB binding proteins to *OB* might act as roadblocks confining previously loaded KorB that initially moved towards the bead. A similar reasoning applies to KorB proteins that initially moved towards *OA* and were subsequently roadblocked by a second KorB binding to *OB*. Since this applies to both case I and case II, we have now rewritten the relevant section in the Results section to make it clearer to readers.

p.9 l.331 “The higher occurrence of case I compared to case II is likely due to the shorter distance between *OB* and *OA* compared to one between *OB* and the proximal bead, reflecting a higher chance of KorB detachment in the latter case.” I’m not completely convinced by this explanation for the accumulation of KorB on the *OA* proximal side. Shouldn’t the off-rate for a given KorB be independent of the segment length? I could imagine that transient binding of KorA plays a role in enriching KorB on the *OA* proximal side. In a related matter, is it possible that *OB* alone acts as a roadblock by re-binding the sliding KorB? Another bound KorB might not be needed to block movement.

The reviewer is correct that K_{off} does not depend on the length of the segment, but it gives a frequency of detachment in s^{-1} i.e. how long KorB will remain bound to the DNA. KorB will explore the whole region between the bead and the *OA* but will accumulate more at the *OA* side because it will find KorA within $1/K_{\text{off}}$ time, which notably decreases its off rate (KorB remains bound to KorA). The *OB-OA* distance is shorter than the *OB-bead* distance, thus it is more likely to find proteins on the *OB-OA* side because proteins that moved towards the bead had a higher chance to detach from the DNA (they did not have enough time to find KorA). In this sense, the reviewer is correct in that the transient binding of KorA plays a role in enriching KorB on the *OA* proximal side. We have now modified the relevant sentence in the Results section accordingly (although we had to be brief in the main text due to the strict word limit imposed by the journal).

We also think that the re-binding of a sliding KorB can indeed act as a roadblock, and in this case, there is no need to have additional binding of other KorB to block the movement of other already-bound KorB molecules.

p.10 l.340 “During the investigation, we noted that the AF647-KorA fluorescence signal exhibited an on-off behavior at the *OA* site in the presence of KorB but absence of CTP (Extended Data Fig. 8b). However, the AF647-KorA signal (at *OA*) was more stable over time when KorB and CTP were both included (Extended Data Fig. 8b), suggesting that KorB-CTP might reduce KorA dissociation from *OA*.” The new experiment in Fig. 4c / Extended Data Fig. 8b is nice. For the sake of completeness, the authors might consider also reporting the residence time of KorA alone, as this would show that KorB CTP has indeed a stabilizing effect rather than the other conditions destabilizing KorA/*OA*.

Following the reviewer’s suggestion, we have performed this additional experiment with AF-KorA only and measured its residence time (see below). Residence times of KorA were similar to those measured in experiment with KorA + KorB but without CTP. These data confirm that KorB-CTP has a stabilizing effect on KorA binding. We have now updated Fig. 4c and the accompany Extended Data Figure with these new data.

Response figure 1. (Left) Representative kymogram showing individual binding events of KorA to OA sites. (Right) Distributions of KorA residence times at OA in various conditions.

p.10 I.340 “KorA and KorB were inducibly produced from separate plasmids, and their production did not alter the copy number of the promoter-*xylE* reporter plasmid nor those of the *korAB* expression plasmids (Extended Data Fig. 9a).” A few comments here:

- The p-values or significance level for the ANOVA should be mentioned in the figure or figure legend.

We agree and we have now reported the p-values for the ANOVA test in the figure legend of Extended Data Fig. 9a. We wrote “...one-way ANOVA statistical test with the null hypothesis that there is no difference in plasmid copy number across all four conditions, three replicates, p-values = 0.10 (left panel, not significant, ns), 0.39 (middle panel, ns), and 0.37 (right panel, ns).”

- The statistical test merely indicates that there isn’t sufficient evidence to reject the null hypothesis, namely that the copy numbers are equal. It does not necessarily mean that there is no difference. Differences might become significant if more replicates were to be observed. I would thus recommend softening the statement above.

We agree and we have toned down the corresponding sentence in the main text to “*there was not sufficient evidence that their production altered the copy number of the promoter-*xylE* reporter plasmid nor those of the *korAB* expression plasmids (Extended Data Fig. 9a).*”

- There is a potential threefold reduction of the *xylE* plasmid in the presence of KorB, which is somewhat close to the apparent four- to fivefold repression under the KorB-only condition. However, the new data convincingly show that the strong repressive effects of KorA and KorAB cannot be explained by potential copy number variations.

We appreciate the point from the reviewer and have toned down the sentence as suggested by the reviewer in the above comment.

p.11 I.397 “Transcription initiation assays confirmed that KorA and KorB (with and without CTP) individually function as repressors, while their combined activity is further augmented (Extended Data Fig. 10 a-b). Notably, KorA emerges as the stronger repressor ($p \leq 0.05$; unpaired Welch’s t-test; Extended Data Fig. 10b).” Since a t-test for KorA vs. KorB is included, one might also consider showing this for KorA vs. KorAB.

We agree and have performed the suggested t-test for KorA vs KorAB as well as KorB vs KorAB for completeness. We further toned down the above sentence by removing the original phrase “...while their (*KorAB*) combined activities is further augmented”; we have now rewritten it as follows “Transcription initiation assays confirmed that KorA and KorB (with and without CTP) individually

function as repressors (Extended Data Fig. 9a-b). The combined activity of KorAB was not significantly greater than KorA alone, but under the conditions of our assay the repression activity of KorA alone was quite high so observing a significant increase with additional KorB became difficult (Extended Data Fig. 9a-b)."

p.12 l.426 "Our findings suggest that KorA and KorB exploit RK2 promoter kinetic instabilities (i.e. weak discriminators) to competitively occlude RNAPs from thermodynamically strong promoters." I'm slightly puzzled by this statement. Could you unpack it a bit to make it more accessible? My intuition would be that slower off-rate kinetics also influence the thermodynamic equilibrium (assuming on-rates don't go down as well).

We agree that the above sentence was highly inaccessible to readers, we have now rewritten the sentences as follows: "*The instability of the PkorA RPo complex, coupled with the observations of KorA and KorB binding to DNA but not to E σ 70, suggests the following model. PkorA RPo is inherently unstable, resulting in frequent RNAP dissociation that allows KorA and KorB to bind their respective operator sites and, upon sliding, form a repressome on the promoter that occludes RNAP from the promoter. Our findings suggest that KorAB exploit RK2 promoter kinetic instabilities (i.e. weak discriminators leading to short half-lives) to competitively occlude RNAPs from the DNA."*